# Spuriosity Didn't Kill the Classifier: Using Invariant Predictions to Harness Spurious Features

**Cian Eastwood**[*,1,2] **Shashank Singh**[*,1]

**Andrei L. Nicolicioiu**[1] **Marin Vlastelica**[1] **Julius von Kügelgen**[1,3] **Bernhard Schölkopf**[1]

[1] Max Planck Institute for Intelligent Systems, Tübingen
[2] University of Edinburgh [3] University of Cambridge

## Abstract

To avoid failures on out-of-distribution data, recent works have sought to extract features that have an invariant or *stable* relationship with the label across domains, discarding "spurious" or *unstable* features whose relationship with the label changes across domains. However, unstable features often carry *complementary* information that could boost performance if used correctly in the test domain. In this work, we show how this can be done *without test-domain labels*. In particular, we prove that pseudo-labels based on stable features provide sufficient guidance for doing so, provided that stable and unstable features are conditionally independent given the label. Based on this theoretical insight, we propose Stable Feature Boosting (SFB), an algorithm for: (i) learning a predictor that separates stable and conditionally-independent unstable features; and (ii) using the stable-feature predictions to adapt the unstable-feature predictions in the test domain. Theoretically, we prove that SFB can learn an asymptotically-optimal predictor without test-domain labels. Empirically, we demonstrate the effectiveness of SFB on real and synthetic data.

## 1 Introduction

Machine learning systems can be sensitive to distribution shift [26]. Often, this sensitivity is due to a reliance on "spurious" features whose relationship with the label changes across domains, ultimately leading to degraded performance in the test domain of interest [21]. To avoid this pitfall, recent works on domain or out-of-distribution (OOD) generalization have sought predictors which only make use of features that have a *stable* or invariant relationship with the label across domains, discarding the spurious or *unstable* features [45, 1, 35, 15]. However, despite their instability, spurious features can often provide additional or *complementary* information about the target label. Thus, if a predictor could be adjusted to use spurious features optimally in the test domain, it would boost performance substantially. That is, perhaps we don't need to discard spurious features at all but rather *use them in the right way*.

As a simple but illustrative example, consider the `ColorMNIST` or `CMNIST` dataset [1]. This transforms the original `MNIST` dataset into a binary classification task (digit in 0–4 or 5–9) and then: (i) flips the label with probability 0.25, meaning that, across all 3 domains, digit shape correctly determines the label with probability 0.75; and (ii) colorizes the digit such that digit color (red or green) is a more informative but spurious feature (see Fig. 1a). Prior work focused on learning an invariant predictor that uses only shape and avoids using color—a spurious feature whose relationship with the label changes across domains. However, as shown in Fig. 1b, the invariant predictor is suboptimal test domains where color can be used in a domain-specific manner to improve performance. We thus ask: when and how can such informative but spurious features be safely harnessed *without labels*?

---

*Equal contribution. Correspondence to `c.eastwood@ed.ac.uk` or `shashankssingh44@gmail.com`.

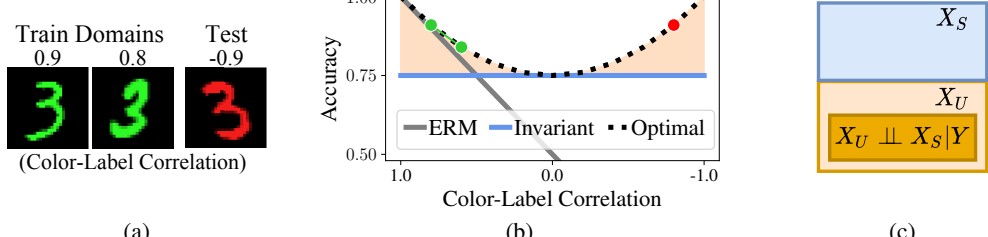

Figure 1: **Invariant (stable) and spurious (unstable) features.** (a) Illustrative images from CMNIST [1]. (b) CMNIST accuracies (y-axis) over test domains of decreasing color-label correlation (x-axis). The 'Oracle' uses both invariant (shape) *and* spurious (color) features optimally in the test domain, boosting performance over an invariant model (orange region). We show how this can be done *without test-domain labels*. (c) Generally, invariant models use only the *stable* component $X_S$ of $X$, discarding the spurious or *unstable* component $X_U$. We prove that predictions based on $X_S$ can be used to safely harness a sub-component of $X_U$ (dark-orange region).

Our main contribution lies in answering this question, showing when and how it is possible to safely harness spurious or *unstable* features without test-domain labels. In particular, we prove that predictions based on stable features provide sufficient guidance for doing so, provided that stable and unstable features are conditionally independent given the label (see Fig. 1c).

**Structure and contributions.** The rest of this paper is organized as follows. We first discuss related work in § 2, providing context and high-level motivation for our approach. In § 3, we then explain how stable and unstable features can be extracted, how unstable features can be harnessed *with* test-domain labels, and the questions/challenges that arise when trying to harness unstable features *without* test-domain labels. In § 4, we present our main theoretical contributions which provide precise answers to these questions, before using these insights to propose the Stable Feature Boosting (SFB) algorithm in § 5. In § 6, we present our experimental results, before ending with a discussion and concluding remarks in § 7. Our contributions can be summarized as follows:

- **Algorithmic:** We propose Stable Feature Boosting (SFB), the first algorithm for using invariant predictions to safely harness spurious features *without test-domain labels*.

- **Theoretical:** SFB is grounded in a novel theoretical result (Thm 4.4) giving sufficient conditions for provable test-domain adaptation without labels. Under these conditions, Thm 4.6 shows that, given enough unlabeled data, SFB learns the Bayes-optimal adapted classifier in the test domain.

- **Experimental:** Our experiments on synthetic and real-world data demonstrate the effectiveness of SFB—even in scenarios where it is unclear if its assumptions are fully satisfied.

## 2 Related Work

**Domain generalization, robustness and invariant prediction.** A fundamental starting point for work in domain generalization is the observation that certain "stable" features, often direct causes of the label, may have an invariant relationship with the label across domains [45, 1, 67, 55, 40, 77, 14]. However, such stable or invariant predictors often discard highly informative but unstable information. Rothenhäusler et al. [51] show that we may need to trade off stability and predictiveness, while Eastwood et al. [15] seek such a trade-off via an interpretable probability-of-generalization parameter. The current work is motivated by the idea that one might avoid such a trade-off by changing how unstable features are used at test time, rather than discarding them at training time.

**Test-domain adaptation *without labels* (unsupervised domain adaptation).** In the source-free and test-time domain adaptation literature, it is common to adapt to new domains using a model's own pseudo-labels [20, 36, 39, 71, 30]—see Rusak et al. [52] for a recent review. In contrast, we: (i) use one (stable) model to provide reliable/robust pseudo-labels and another (unstable) model to adapt domain-specific information; and (ii) propose a bias correction step that provably ensures an accurate, well-calibrated unstable model ($\Pr[Y|X_U]$) as well as an optimal joint/combined model ($\Pr[Y|X_S, X_U]$). Beyond this literature, Bui et al. [12] propose a meta-learning approach for exploiting unstable/domain-specific features. However, they use unstable features *in the same way* in the test domain, which, by definition, is not robust and can degrade performance. Sun et al. [63] share the goal of exploiting unstable features to go "beyond invariance". However, in contrast to our approach, they require labels for the unstable features (rarely available) and only address label shifts.

Table 1: **Comparison with related work.** *QRM [15] uses a hyperparameter $\alpha \in [0, 1]$ to balance the probability of robust generalization and using more information from $X$.

| Method | Components of $X$ Used | | | Robust | No test-domain labels |
|---|---|---|---|---|---|
| | **Stable** | **Complementary** | **All** | | |
| ERM [65] | ✓ | ✓ | ✓ | ✗ | ✓ |
| IRM [1] | ✓ | ✗ | ✗ | ✓ | ✓ |
| QRM [15] | ✓ | ✓* | ✓* | ✓* | ✓ |
| DARE [50] | ✓ | ✓ | ✓ | ✓ | ✗ |
| ACTIR [31] | ✓ | ✓ | ✗ | ✓ | ✗ |
| SFB (**Ours**) | ✓ | ✓ | ✗ | ✓ | ✓ |

**Test-domain adaptation *with labels* (few-shot fine-tuning).** Fine-tuning part of a model using a small number of labeled test-domain examples is a common way to deal with distribution shift [16, 17, 13]. More recently, it has been shown that simply retraining the last layer of an ERM-trained model outperforms more robust feature-learning methods on spurious correlation benchmarks [50, 32, 74]. Similar to our approach, Jiang and Veitch [31] separate stable and conditionally-independent unstable features and then adapt their use of the latter in the test domain. However, in contrast to our approach, theirs requires test-domain labels. In addition, they assume data is drawn from an anti-causal generative model, which is strictly stronger than our "complementarity" assumption (see § 4).

Table 1 summarizes related work while App. H discusses further related work.

## 3 Problem Setup: Extracting and Harnessing Unstable Features

**Setup.** We consider the problem of domain generalization (DG) [8, 42, 24] where predictors are trained on data from multiple training domains and with the goal of performing well on data from unseen test domains. More formally, we consider datasets $D^e = \{(X_i^e, Y_i^e)\}_{i=1}^{n_e}$ collected from $m$ different training domains or *environments* $\mathcal{E}_{\text{tr}} := \{E_1, \ldots, E_m\}$, with each dataset $D^e$ containing data pairs $(X_i^e, Y_i^e)$ sampled i.i.d. from $\mathbb{P}(X^e, Y^e)$.[2] The goal is then to learn a predictor $f(X)$ that performs well on a larger set $\mathcal{E}_{\text{all}} \supset \mathcal{E}_{\text{tr}}$ of possible domains.

**Average performance: use all features.** The first approaches to DG sought predictors that perform well *on average* over domains [8, 42] using empirical risk minimization (ERM, [66]). However, predictors that perform well on average provably lack robustness [43, 49], potentially performing quite poorly on large subsets of $\mathcal{E}_{\text{all}}$. In particular, minimizing the average error leads predictors to make use of any features that are informative about the label (on average), including "spurious" or "shortcut" [21] features whose relationship with the label is subject to change across domains. In test domains where these feature-label relationships change in new or more severe ways than observed during training, this usually leads to significant performance drops or even complete failure [73, 4].

**Worst-case or robust performance: use only stable features.** To improve robustness, subsequent works sought predictors that only use *stable or invariant* features, i.e., those that have a stable or invariant relationship with the label across domains [45, 1, 47, 70, 58]. For example, Arjovsky et al. [1] do so by enforcing that the classifier on top of these features is optimal for all domains simultaneously. We henceforth use *stable features* and $X_S$ to refer to these features, and stable predictors to refer to predictors which use only these features. Analogously, we use *unstable features* and $X_U$ to refer to features with an unstable or "spurious" relationship with the label across domains. Note that $X_S$ and $X_U$ partition the components of $X$ which are informative about $Y$, as depicted in Fig. 1c, and that formal definitions of $X_S$ and $X_U$ are provided in § 4.

### 3.1 Harnessing unstable features *with labels*

A stable predictor $f_S$ is unlikely to be the best predictor in any given domain. As illustrated in Fig. 1b, this is because it excludes unstable features $X_U$ which are informative about $Y$ and can boost performance *if used in an appropriate, domain-specific manner*. Assuming we can indeed learn a stable predictor with prior methods, we start by showing how $X_U$ can be harnessed *with test-domain labels*.

---

[2]We drop the domain superscript $e$ when referring to random variables from any environment.

**Boosting the stable predictor.** We describe boosted joint predictions $f^e(X)$ in domain $e$ as some combination $C$ of stable predictions $f_S(X)$ and domain-specific unstable predictions $f_U^e(X)$, i.e., $f^e(X) = C(f_S(X), f_U^e(X))$. To allow us to adapt only the $X_U$-$Y$ relation, we decompose the stable $f_S(X) = h_S(\Phi_S(X))$ and unstable $f_U^e(X) = h_U^e(\Phi_U(X))$ predictions into feature extractors $\Phi$ and classifiers $h$. $\Phi_S$ extracts stable components $X_S = \Phi_S(X)$ of $X$, $\Phi_U$ extracts unstable components $X_U = \Phi_U(X)$ of $X$, $h_S$ is a classifier learned on top of $\Phi_S$ (shared across domains), and $h_U^e$ is a *domain-specific* unstable classifier learned on top of $\Phi_U$ (one per domain). Putting these together,

$$f^e(X) = C(f_S(X), f_U^e(X)) = C(h_S(\Phi_S(X)), h_U^e(\Phi_U(X))) = C(h_S(X_S), h_U^e(X_U)), \quad (3.1)$$

where $C : [0,1] \times [0,1] \to [0,1]$ is a *combination function* that combines the stable and unstable predictions. For example, Jiang and Veitch [31, Eq. 2.1] add stable $p_S$ and unstable $p_U$ predictions in logit space, i.e., $C(p_S, p_U) = \sigma(\text{logit}(p_S) + \text{logit}(p_U))$. Since it is unclear, *a priori*, how to choose $C$, we will leave it unspecified until Thm. 4.4 in § 4, where we derive a principled choice.

**Adapting with labels.** Given a new domain $e$ and labels $Y^e$, we can boost performance by adapting $h_U^e$. Specifically, letting $\ell : \mathcal{Y} \times \mathcal{Y} \to \mathbb{R}$ be a loss function (e.g., cross-entropy) and $R^e(f) = \mathbb{E}_{(X,Y)}[\ell(Y, f(X))|E = e]$ the risk of predictor $f : \mathcal{X} \to \mathcal{Y}$ in domain $e$, we can adapt $h_U^e$ to solve:

$$\min_{h_U} R^e(C(h_S \circ \Phi_S, h_U \circ \Phi_U)) \tag{3.2}$$

### 3.2 Harnessing unstable features *without labels*

We now consider the main question of this work—can we reliably harness $X_U$ *without* test-domain labels? We could, of course, simply select a *fixed* unstable classifier $h_U^e$ by relying solely on the training domains (e.g., by minimizing average error), and hope that this works for the test-domain $X_U$-$Y$ relation. However, by definition of $X_U$ being unstable, this is clearly not a robust or reliable approach—the focus of our efforts in this work, as illustrated in Table 1. As in § 3.1, we assume that we are able to learn a stable predictor $f_S$ using prior methods, e.g., IRM [1] or QRM [15].

**From stable predictions to robust pseudo-labels.** While we don't have labels in the test domain, we *do* have stable predictions. By definition, these are imperfect (i.e., *noisy*) but robust, and can be used to form *pseudo-labels* $\widehat{Y}_i = \arg\max_j(f_S(X_i))_j$, with $(f_S(X_i))_j \approx \Pr[Y_i = j|X_S]$ denoting the $j^{\text{th}}$ entry of the stable prediction for $X_i$. Can we somehow use these noisy but robust pseudo-labels to guide our updating of $h_U^e$, and, ultimately, our use of $X_U$ in the test domain?

**From joint to unstable-only risk.** If we simply use our robust pseudo-labels as if they were true labels—updating $h_U^e$ to minimize the joint risk as in Eq. (3.2)—we arrive at trivial solutions since $f_S$ already predicts its own pseudo-labels with 100% accuracy. For example, if we follow [31, Eq. 2.1] and use the combination function $C(p_S, p_U) = \sigma(\text{logit}(p_S) + \text{logit}(p_U))$, then the trivial solution $\text{logit}(h_U^e(\cdot)) = 0$ achieves 100% accuracy (and minimizes cross-entropy; see Prop. D.1 of App. D). Thus, we cannot minimize a joint loss involving $f_S$'s predictions when using $f_S$'s pseudo-labels. A sensible alternative is to update $h_U^e$ to minimize the *unstable-only risk* $R^e(h_U^e \circ \Phi_U)$.

**More questions than answers.** While this new procedure *could* work, it raises questions about *when* it will work, or, more precisely, the conditions under which it can be used to safely harness $X_U$. We now summarise these questions before addressing them in § 4:

1. **Does it make sense to minimize the unstable-only risk?** In particular, when can we minimize the unstable-predictor risk *alone* or separately, and then arrive at the optimal joint predictor? This cannot always work; e.g., for independent $X_S, X_U \sim \text{Bernoulli}(1/2)$ and $Y = X_S$ XOR $X_U$, $Y$ is independent of each of $X_S$ and $X_U$ and hence cannot be predicted from either alone.

2. **How should we combine predictions?** Is there a principled choice for the combination function $C$ in Eq. (3.1)? In particular, is there a $C$ that correctly weights stable and unstable predictions in the test domain? As $X_U$ could be very strongly or very weakly predictive of $Y$ in the test domain, this seems a difficult task. Intuitively, correctly weighting stable and unstable predictions requires them to be properly calibrated: do we have any reason to believe that, after training on $f_S$'s pseudo-labels, $h_U^e$ will be properly calibrated in the test domain?

3. **Can the student outperform the teacher?** Stable predictions likely make mistakes—indeed, this is the motivation for trying to improve them. Is it possible to correct these mistakes with $X_U$? Is it

possible to learn an unstable "student" predictor that outperforms its own supervision signal or "teacher"? Perhaps surprisingly, we show that, for certain types of features, the answer is yes. In fact, even a very weak stable predictor, with performance just above chance, can be used to learn an *optimal* unstable classifier in the test domain given enough unlabeled data.

# 4 Theory: When Can We Safely Harness Unstable Features Without Labels?

Suppose we have already identified a stable feature $X_S$ and a potentially unstable feature $X_U$ (we will return to the question of how to learn/extract $X_S$ and $X_U$ themselves in § 5). In this section, we analyze the problem of using $X_S$ to leverage $X_U$ in the test domain without labels. We first reduce this to a special case of the so-called "marginal problem" in probability theory, i.e., the problem of identifying a joint distribution based on information about its marginals. In the special case where two variables are conditionally independent given a third, we show this problem can be solved exactly. This solution, which may be of interest beyond the context of domain generalization/adaptation, motivates our test-domain adaptation algorithm (Alg. 1), and forms the basis of Thm. 4.6 which shows that Alg. 1 converges to the best possible classifier given enough unlabeled data.

We first pose a population-level model of our domain generalization setup. Let $E$ be a random variable denoting the *environment*. Given an environment $E$, we have that the stable feature $X_S$, unstable feature $X_U$ and label $Y$ are distributed according to $P_{X_S,X_U,Y|E}$. We can now formalize the three key assumptions underlying our approach, starting with the notion of a stable feature, motivated in § 3:

**Definition 4.1** (Stable and Unstable Features). $X_S$ is a stable *feature with respect to $Y$ if $P_{Y|X_S}$ does not depend on $E$; equivalently, if $Y$ and $E$ are conditionally independent given $X_S$ ($Y \perp\!\!\!\perp E|X_S$). Conversely, $X_U$ is an unstable feature with respect to $Y$ if $P_{Y|X_U}$ depends on $E$; equivalently, if $Y$ and $E$ are conditionally dependent given $X_U$ ($Y \not\perp\!\!\!\perp E|X_U$).*

Next, we state our complementarity assumption, which we will show justifies the approach of separately learning the relationships $X_S$-$Y$ and $X_U$-$Y$ and then combining them:

**Definition 4.2** (Complementary Features). $X_S$ and $X_U$ are complementary *features with respect to $Y$ if $X_S \perp\!\!\!\perp X_U|(Y,E)$; i.e., if $X_S$ and $X_U$ share no redundant information beyond $Y$ and $E$.*

Finally, to provide a useful signal for test-domain adaptation, the stable feature needs to help predict the label in the test domain. Formally, we assume:

**Definition 4.3** (Informative Feature). $X_S$ is said to be informative *of $Y$ in environment $E$ if $X_S \not\perp\!\!\!\perp Y|E$; i.e., $X_S$ is predictive of $Y$ within the environment $E$.*

We will discuss the roles of these assumptions after stating our main result (Thm. 4.4) that uses them. To keep our results as general as possible, we avoid assuming a particular causal generative model, but the above conditional (in)dependence assumptions can be interpreted as constraints on such a causal model. App. D.2 formally characterizes the set of causal models that are consistent with our assumptions and shows that our setting generalizes those of prior works [49, 68, 31, 69].

**Reduction to the marginal problem with complementary features.** By Defn. 4.1, we have the same stable relationship $P_{Y|X_S,E} = P_{Y|X_S}$ in training and test domains. Now, suppose we have used the training data to learn this stable relationship and thus know $P_{Y|X_S}$. Also suppose that we have enough unlabeled data from test domain $E$ to learn $P_{X_S,X_U|E}$, and recall that our ultimate goal is to predict $Y$ from $(X_S, X_U)$ in test domain $E$. Since the rest of our discussion is conditioned on $E$ being the test domain, we omit $E$ from the notation. Now note that, if we could express $P_{Y|X_S,X_U}$ in terms of $P_{Y|X_S}$ and $P_{X_S,X_U}$, we could then use $P_{Y|X_S,X_U}$ to optimally predict $Y$ from $(X_S, X_U)$. Thus, our task thus becomes to reconstruct $P_{Y|X_S,X_U}$ from $P_{Y|X_S}$ and $P_{X_S,X_U}$. This is an instance of the classical "marginal problem" from probability theory [28, 29, 19], which asks under which conditions we can recover the joint distribution of a set of random variables given information about its marginals. In general, although one can place bounds on the conditional distributions $P_{Y|X_U}$ and $P_{Y|X_S,X_U}$, they cannot be completely inferred from $P_{Y|X_S}$ and $P_{X_S,X_U}$ [19]. However, the following section demonstrates that, *under the additional assumptions that $X_S$ and $X_U$ are complementary and $X_S$ is informative*, we can exactly recover $P_{Y|X_U}$ and $P_{Y|X_S,X_U}$ from $P_{Y|X_S}$ and $P_{X_S,X_U}$.

## 4.1 Solving the marginal problem with complementary features

We now present our main result which shows how to reconstruct $P_{Y|X_S,X_U}$ from $P_{Y|X_S}$ and $P_{X_S,X_U}$ when $X_S$ and $X_U$ are complementary and $X_S$ is informative. To simplify notation, we assume the label $Y$ is binary and defer the multi-class extension to App. C.

**Theorem 4.4** (Solution to the marginal problem with binary labels and complementary features)**.** *Consider three random variables $X_S$, $X_U$, and $Y$, where (i) $Y$ is binary ($\{0,1\}$-valued), (ii) $X_S$ and $X_U$ are complementary features for $Y$ (i.e., $X_S \perp\!\!\!\perp X_U | Y$), and (iii) $X_S$ is informative of $Y$ ($X_S \not\perp\!\!\!\perp Y$). Then, the joint distribution of $(X_S, X_U, Y)$ can be written in terms of the joint distributions of $(X_S, Y)$ and $(X_S, X_U)$. Specifically, if $\widehat{Y}|X_S \sim \text{Bernoulli}(\Pr[Y=1|X_S])$ is a pseudo-label[3] and*

$$\epsilon_0 := \Pr[\widehat{Y}=0|Y=0] \quad and \quad \epsilon_1 := \Pr[\widehat{Y}=1|Y=1] \tag{4.1}$$

*are the accuracies of the pseudo-labels on classes $0$ and $1$, respectively. Then, we have:*

$$\epsilon_0 + \epsilon_1 > 1, \tag{4.2}$$

$$\Pr[Y=1|X_U] = \frac{\Pr[\widehat{Y}=1|X_U] + \epsilon_0 - 1}{\epsilon_0 + \epsilon_1 - 1}, \quad and \tag{4.3}$$

$$\Pr[Y=1|X_S,X_U] = \sigma\left(\text{logit}(\Pr[Y=1|X_S]) + \text{logit}(\Pr[Y=1|X_U]) - \text{logit}(\Pr[Y=1])\right). \tag{4.4}$$

Intuitively, suppose we generate pseudo-labels $\widehat{Y}$ based on feature $X_S$ and train a model to predict $\widehat{Y}$ using feature $X_U$. For complementary $X_S$ and $X_U$, Eq. (4.3) shows how to transform this into a prediction of the *true* label $Y$, correcting for differences between $\widehat{Y}$ and $Y$. Crucially, given the conditional distribution $P_{Y|X_S}$ and observations of $X_S$, we can estimate class-wise pseudo-label accuracies $\epsilon_0$ and $\epsilon_1$ in Eq. (4.3) even without new labels $Y$ (see App. A.1, Eq. (A.2)). Finally, Eq. (4.4) shows how to weight predictions based on $X_S$ and $X_U$, justifying the combination function

$$C_p(p_S, p_U) = \sigma(\text{logit}(p_S) + \text{logit}(p_U) - \text{logit}(p)) \tag{4.5}$$

in Eq. (3.1), where $p = \Pr[Y=1]$ is a constant independent of $x_S$ and $x_U$. We now sketch the proof of Thm. 4.4, elucidating the roles of informativeness and complementarity (full proof in App. A.1).

*Proof Sketch of Thm. 4.4.* We prove Eq. (4.2), Eq. (4.3), and Eq. (4.4) in order.

**Proof of Eq. (4.2):** The informativeness condition (iii) is equivalent to the pseudo-labels having predictive accuracy above random chance; formally, App. A.1 shows:

**Lemma 4.5.** $\epsilon_0 + \epsilon_1 > 1$ *if and only if $X_S$ is informative of $Y$ (i.e., $X_S \not\perp\!\!\!\perp Y$).*

Together with Eq. (4.3), it follows that *any* dependence between $X_S$ and $Y$ allows us to fully learn the relationship between $X_U$ and $Y$, affirmatively answering our question from § 3: *Can the student outperform the teacher?* While a stronger relationship between $X_S$ and $Y$ is still helpful, it only improves the (unlabeled) *sample complexity* of learning $P_{Y|X_U}$ and not *consistency* (Thm. 4.6 below), mirroring related results in the literature on learning from noisy labels [44, 7]. In particular, a weak relationship corresponds to $\epsilon_0 + \epsilon_1 \approx 1$, increasing the variance of the bias-correction in Eq. (4.3). With a bit more work, one can formalize this intuition to show that our approach has a relative statistical efficiency of $\epsilon_0 + \epsilon_1 - 1 \in [0,1]$, compared to using true labels $Y$.

**Proof of Eq. (4.3):** The key observation behind the bias-correction (Eq. (4.3)) is that, due to complementarity ($X_S \perp\!\!\!\perp X_U | Y$) and the fact that the pseudo-label $\widehat{Y}$ depends only on $X_S$, $\widehat{Y}$ is conditionally independent of $X_U$ given the true label $Y$ ($\widehat{Y} \perp\!\!\!\perp X_U | Y$); formally:

$$\begin{aligned}
\Pr[\widehat{Y}=1|X_U] &= \Pr[\widehat{Y}=1|Y=0,X_U]\Pr[Y=0|X_U] \\
&\quad + \Pr[\widehat{Y}=1|Y=1,X_U]\Pr[Y=1|X_U] \quad &\text{(Law of Total Probability)} \\
&= \Pr[\widehat{Y}=1|Y=0]\Pr[Y=0|X_U] \\
&\quad + \Pr[\widehat{Y}=1|Y=1]\Pr[Y=1|X_U] \quad &\text{(Complementarity)} \\
&= (\epsilon_0 + \epsilon_1 - 1)\Pr[Y=1|X_U] + 1 - \epsilon_0. \quad &\text{(Definitions of } \epsilon_0 \text{ and } \epsilon_1)
\end{aligned}$$

---

[3]Our *stochastic* pseudo-labels differ from hard ($\widehat{Y} = 1\{\Pr[Y=1|X_S] > 1/2\}$) pseudo-labels often used in practice [20, 36, 52]. By capturing irreducible error in $Y$, stochastic pseudo-labels ensure $\Pr[Y|X_U]$ is well-calibrated, allowing us to combine $\Pr[Y|X_S]$ and $\Pr[Y|X_U]$ in Eq. (4.4).

---

**Algorithm 1:** Bias-corrected adaptation procedure. Multi-class version given by Algorithm 2.

**Input:** Calibrated stable classifier $f_S(x_S) = \Pr[Y = 1 | X_S = x_S]$, unlabelled data $\{(X_{S,i}, X_{U,i})\}_{i=1}^n$

**Output:** Joint classifier $\widehat{f}(x_S, x_U)$ estimating $\Pr[Y = 1 | X_S = x_S, X_U = x_U]$

1  Compute soft pseudo-labels (PLs) $\{\widehat{Y}_i\}_{i=1}^n$ with $\widehat{Y}_i = f_S(X_{S,i})$

2  Compute soft class-1 count $n_1 = \sum_{i=1}^n \widehat{Y}_i$

3  Estimate PL accuracies $(\widehat{\epsilon}_0, \widehat{\epsilon}_1) = \left(\frac{1}{n-n_1} \sum_{i=1}^n (1 - \widehat{Y}_i)(1 - f_S(X_{S,i})), \frac{1}{n_1} \sum_{i=1}^n \widehat{Y}_i f_S(X_{S,i})\right)$    `// Eq.(4.1)`

4  Fit unstable classifier $\widetilde{f}_U(x_U)$ to pseudo-labelled data $\{(X_{U,i}, \widehat{Y}_i)\}_{i=1}^n$    `//` $\approx \Pr[\widehat{Y} = 1 | X_U = x_U]$

5  Bias-correct $\widehat{f}_U(x_U) \mapsto \max\left\{0, \min\left\{1, \frac{\widetilde{f}_U(x_U) + \widehat{\epsilon}_0 - 1}{\widehat{\epsilon}_0 + \widehat{\epsilon}_1 - 1}\right\}\right\}$    `// Eq.(4.3),` $\approx \Pr[Y = 1 | X_U = x_U]$

6  **return** $\widehat{f}(x_S, x_U) \mapsto C_{\frac{n_1}{n}}(f_S(x_S), \widehat{f}_U(x_U))$    `// Eq.(4.4)/(4.5),` $\approx \Pr[Y = 1 | X_S = x_S, X_U = x_U]$

---

Here, complementarity allowed us to approximate the unknown $\Pr[\widehat{Y} = 1 | Y = 0, X_U]$ by its average $\Pr[\widehat{Y} = 1 | Y = 0] = \mathbb{E}_{X_U}[\Pr[\widehat{Y} = 1 | Y = 0, X_U]]$, which depends only on the known distribution $P_{X_S, Y}$. By informativeness, Lemma 4.5 allows us to divide by $\epsilon_0 + \epsilon_1 - 1$, giving Eq. (4.3).

**Proof of Eq. (4.4):**    While the exact proof of Eq. (4.4) is a bit more algebraically involved, the key idea is simply that complementarity allows us to decompose $\Pr[Y | X_S, X_U]$ into separately-estimable terms $\Pr[Y | X_S]$ and $\Pr[Y | X_U]$: for any $y \in \mathcal{Y}$,

$$\Pr[Y = y | X_S, X_U] \propto_{X_S, X_U} \Pr[X_S, X_U | Y = y] \Pr[Y = y] \qquad \text{(Bayes' Rule)}$$

$$= \Pr[X_S | Y = y] \Pr[X_U | Y = 1] \Pr[Y = y] \qquad \text{(Complementarity)}$$

$$\propto_{X_S, X_U} \frac{\Pr[Y = y | X_S] \Pr[Y = 1 | X_U]}{\Pr[Y = 1]}, \qquad \text{(Bayes' Rule)}$$

where, $\propto_{X_S, X_U}$ denotes proportionality with a constant depending only on $X_S$ and $X_U$, not on $y$. Directly estimating these constants involves estimating the density of $(X_S, X_U)$, which may be intractable without further assumptions. However, in the binary case, since $1 - \Pr[Y = 1 | X_S, X_U] = \Pr[Y = 0 | X_S, X_U]$, these proportionality constants conveniently cancel out when the above relationship is written in logit-space, as in Eq. (4.4). In the multi-class case, App. C shows how to use the constraint $\sum_{y \in \mathcal{Y}} \Pr[Y = y | X_S, X_U] = 1$ to avoid computing the proportionality constants.    $\square$

## 4.2  A provably consistent algorithm for unsupervised test-domain adaptation

Having learned $P_{Y | X_S}$ from the training domain(s), Thm. 4.4 implies we can learn $P_{Y | X_S, X_U}$ in the test domain by learning $P_{X_S, X_U}$—the latter only requiring *unlabeled* test-domain data. This motivates our Alg. 1 for test-domain adaptation, which is a finite-sample version of the bias-correction and combination equations (Eqs. (4.3) and (4.4)) in Thm. 4.4. Alg. 1 comes with the following guarantee:

**Theorem 4.6** (Consistency Guarantee, Informal)**.** *Assume (i) $X_S$ is stable, (ii) $X_S$ and $X_U$ are complementary, and (iii) $X_S$ is informative of $Y$ in the test domain. As $n \to \infty$, if $\widetilde{f}_U \to \Pr[\widehat{Y} = 1 | X_U]$ then $\widehat{f} \to \Pr[Y = 1 | X_S, X_U]$.*

In words, as the amount of unlabeled data from the test domain increases, if the unstable classifier on Line 4 of Alg. 1 learns to predict the pseudo-label $\widehat{Y}$, then the joint classifier output by Alg. 1 learns to predict the true label $Y$. Convergence in Thm. 4.6 occurs $P_{X_S, X_U}$-a.e., both weakly (in prob.) and strongly (a.s.), depending on the convergence of $\widetilde{f}_U$. Formal statements and proofs are in Appendix B.

## 5  Algorithm: Stable Feature Boosting (SFB)

Using theoretical insights from § 4, we now propose Stable Feature Boosting (SFB): an algorithm for safely harnessing unstable features without test-domain labels. We first describe learning a stable predictor and extracting complementary unstable features from the training domains. We then describe how to use these with Alg. 1, adapting our use of the unstable features to the test domain.

**Training domains: Learning stable and complementary features.** Using the notation of Eq. (3.1), our goal on the training domains is to learn stable and unstable features $\Phi_S$ and $\Phi_U$, a stable predictor $f_S$, and domain-specific unstable predictors $f_U^e$ such that:

1. $f_S$ is stable, informative, and calibrated (i.e., $f_S(x_S) = \Pr[Y = 1 | X_S = x_S]$).

2. In domain $e$, $f_U^e$ boosts $f_s$'s performance with complementary $\Phi_U(X^e) \perp\!\!\!\perp \Phi_S(X^e) | Y^e$.

To achieve these learning goals, we propose the following objective:

$$\min_{\Phi_S, \Phi_U, h_S, h_U^e} \sum_{e \in \mathcal{E}_{\text{tr}}} R^e(h_S \circ \Phi_S) + R^e(C(h_S \circ \Phi_S, h_U^e \circ \Phi_U)) \tag{5.1}$$
$$+ \lambda_S \cdot P_{\text{Stability}}(\Phi_S, h_S, R^e) + \lambda_C \cdot P_{\text{CondIndep}}(\Phi_S(X^e), \Phi_U(X^e), Y^e)$$

The first term encourages good stable predictions $f_S(X) = h_S(\Phi_S(X))$ while the second encourages improved domain-specific joint predictions $f^e(X^e) = C(h_S(\Phi_S(X^e)), h_U^e(\Phi_U(X^e)))$ via a domain-specific use $h_U^e$ of the unstable features $\Phi_U(X^e)$. For binary $Y$, the combination function $C$ takes the simplified form of Eq. (4.5). Otherwise, $C$ takes the more general form of Eq. (C.1). $P_{\text{Stability}}$ is a penalty encouraging stability while $P_{\text{CondIndep}}$ is a penalty encouraging complementarity or conditional independence, i.e., $\Phi_U(X^e) \perp\!\!\!\perp \Phi_S(X^e) | Y^e$. Several approaches exist for enforcing stability [1, 35, 58, 47, 15, 67, 40, 77] (e.g., IRM [1]) and conditional independence (e.g., conditional HSIC [22]). $\lambda_S \in [0, \infty)$ and $\lambda_C \in [0, \infty)$ are regularization hyperparameters. While another hyperparameter $\gamma \in [0, 1]$ could control the relative weighting of stable and joint risks, i.e., $\gamma R^e(h_S \circ \Phi_S)$ and $(1 - \gamma) R^e(C(h_S \circ \Phi_S, h_U^e \circ \Phi_U))$, we found this unnecessary in practice. Finally, note that, in principle, $h_U^e$ could take any form and we could learn completely separate $\Phi_S, \Phi_U$. In practice, we simply take $h_U^e$ to be a linear classifier and split the output of a shared $\Phi(X) = (\Phi_S(X), \Phi_U(X))$.

**Post-hoc calibration.** As noted in § 4.2, the stable predictor $f_S$ must be properly calibrated to (i) form unbiased unstable predictions (Line 5 of Alg. 1) and (ii) correctly combine the stable and unstable predictions (Line 6 of Alg. 1). Thus, after optimizing the objective (5.1), we apply a post-processing step (e.g., temperature scaling [25]) to calibrate $f_S$.

**Test-domain adaptation without labels.** Given a stable predictor $f_S = h_S \circ \Phi_S$ and complementary features $\Phi_U(X)$, we now adapt the unstable classifier $h_U^e$ in the test domain to safely harness (or make optimal use of) $\Phi_U(X)$. To do so, we use the bias-corrected adaptation algorithm of Alg. 1 (or Alg. 2 for the multi-class case) which takes as input the stable classifier $h_S$[4] and unlabelled test-domain data $\{\Phi_S(x_i), \Phi_U(x_i)\}_{i=1}^{n_e}$, outputting a joint classifier adapted to the test domain.

## 6 Experiments

We now evaluate the performance of our algorithm on synthetic and real-world datasets requiring out-of-distribution generalization. App. E contains full details on these datasets and a depiction of their samples (see Fig. 4). In the experiments below, SFB uses IRM [1] for $P_{\text{Stability}}$ and the conditional-independence proxy of Jiang and Veitch [31, §3.1] for $P_{\text{CondIndep}}$, with App. F.1.2 giving results with other stability penalties. App. F contains further results, including ablation studies (F.1.1) and results on additional datasets (F.2). In particular, App. F.2 contains results on the `Camelyon17` medical dataset [3] from the WILDS package [33], where we find that all methods perform similarly *when properly tuned* (see discussion in App. F.2). Code is available at: https://github.com/cianeastwood/sfb.

**Synthetic data.** We consider two synthetic datasets: anti-causal (AC) data and cause-effect data with direct $X_S$-$X_U$ dependence (CE-DD). AC data satisfies the structural equations

$$Y \leftarrow \text{Rad}(0.5);$$
$$X_S \leftarrow Y \cdot \text{Rad}(0.75);$$
$$X_U \leftarrow Y \cdot \text{Rad}(\beta_e),$$

where the input $X = (X_S, X_U)$ and $\text{Rad}(\beta)$ denotes a Rademacher random variable that

---

[4]Note: while Sections 3 and 5 use $h$ for the classifier and $f = h \circ \Phi$ for the classifier-representation composition, Section 4 and Alg. 1 use $f$ for the classifier, since no representation $\Phi$ is being learned.

Table 2: Synthetic & PACS test-domain accuracies over 100 & 5 seeds each.

| | Synthetic | | PACS | | | |
|---|---|---|---|---|---|---|
| Algorithm | AC | CE-DD | P | A | C | S |
| ERM | $9.9 \pm 0.1$ | $11.6 \pm 0.7$ | $93.0 \pm 0.7$ | $79.3 \pm 0.5$ | $74.3 \pm 0.7$ | $65.4 \pm 1.5$ |
| ERM + PL | $9.9 \pm 0.1$ | $11.6 \pm 0.7$ | $93.7 \pm 0.4$ | $79.6 \pm 1.5$ | $74.1 \pm 1.2$ | $63.1 \pm 3.1$ |
| IRM [1] | $74.9 \pm 0.1$ | $69.6 \pm 1.3$ | $93.3 \pm 0.3$ | $78.7 \pm 0.7$ | $75.4 \pm 1.5$ | $65.6 \pm 2.5$ |
| IRM + PL | $74.9 \pm 0.1$ | $69.6 \pm 1.3$ | $94.1 \pm 0.7$ | $78.9 \pm 2.9$ | $75.1 \pm 4.6$ | $62.9 \pm 4.9$ |
| ACTIR [31] | $74.8 \pm 0.4$ | $43.5 \pm 2.6$ | $94.8 \pm 0.1$ | $\mathbf{82.5 \pm 0.4}$ | $\mathbf{76.6 \pm 0.6}$ | $62.1 \pm 1.3$ |
| SFB no adpt. | $74.7 \pm 1.2$ | $74.9 \pm 3.6$ | $93.7 \pm 0.6$ | $78.1 \pm 1.1$ | $73.7 \pm 0.6$ | $69.7 \pm 2.3$ |
| SFB | $\mathbf{89.2 \pm 2.9}$ | $\mathbf{88.6 \pm 1.4}$ | $\mathbf{95.8 \pm 0.6}$ | $80.4 \pm 1.3$ | $\mathbf{76.6 \pm 0.6}$ | $\mathbf{71.8 \pm 2.0}$ |

Table 3: CMNIST test accuracies over 10 seeds.

| Algorithm | Test Acc. |
|---|---|
| ERM | $27.9 \pm 1.5$ |
| IRM [1] | $69.7 \pm 0.9$ |
| SFB no adpt. | $70.6 \pm 1.8$ |
| SFB | $\mathbf{88.1 \pm 1.8}$ |
| Oracle no adpt. | $72.1 \pm 0.7$ |
| Oracle | $89.9 \pm 0.1$ |

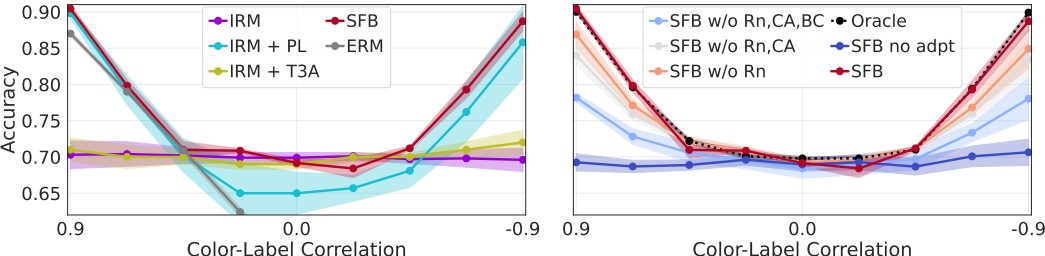

Figure 2: CMNIST accuracies (y-axis) over test domains of decreasing color-label correlation (x-axis). Empirical versions of Fig. 1b. *Left:* SFB vs. baseline methods. *Right:* Ablations showing SFB without (w/o) bias correction (BC), calibration (CA) and multiple pseudo-labeling rounds (Rn). Numerical results in Table 7 of App. F.1.3.

is $-1$ with probability $1 - \beta$ and $+1$ with probability $\beta$. Following [31, §6.1], we create two training domains with $\beta_e \in \{0.95, 0.7\}$, one validation domain with $\beta_e = 0.6$ and one test domain with $\beta_e = 0.1$. CE-DD data is generated according to the structural equations

$$
\begin{aligned}
X_S &\leftarrow \text{Bern}(0.5); \\
Y &\leftarrow \text{XOR}(X_S, \text{Bern}(0.75)); \\
X_U &\leftarrow \text{XOR}(\text{XOR}(Y, \text{Bern}(\beta_e)), X_S),
\end{aligned}
$$

where $\text{Bern}(\beta)$ denotes a Bernoulli random variable that is 1 with probability $\beta$ and 0 with probability $1 - \beta$. Note that $X_S \not\!\perp\!\!\!\perp X_U | Y$, since $X_S$ directly influences $X_U$. Following [31, App. B], we create two training domains with $\beta_e \in \{0.95, 0.8\}$, one validation domain with $\beta_e = 0.2$, and one test domain with $\beta_e = 0.1$. For both datasets, the idea is that, during training, prediction based on the stable $X_S$ results in lower accuracy (75%) than prediction based on the unstable $X_U$. Thus, models optimizing for prediction accuracy only—and not stability—will use $X_U$ and ultimately end up with only 10% in the test domain. Importantly, while the stable predictor achieves 75% accuracy in the test domain, this can be improved to 90% if $X_U$ is used correctly. Following [31], we use a simple 3-layer network for both datasets and choose hyperparameters using the validation-domain performance: see App. G.2 for further implementation details.

On the AC dataset, Table 2 shows that ERM performs poorly as it misuses $X_U$, while IRM, ACTIR, and SFB-no-adpt. do well by using only $X_S$. Critically, only SFB (with adaptation) is able to harness $X_U$ in the test domain *without labels*, leading to a near-optimal performance boost.

On the CE-DD dataset, Table 2 again shows that ERM performs poorly while IRM and SFB-no-adpt. do well by using only the stable $X_S$. However, we now see that ACTIR performs poorly since its assumption of anti-causal structure no longer holds. This highlights another key advantage of SFB over ACTIR: any stability penalty can be used, including those with weaker assumptions than ACTIR's anti-causal structure (e.g., IRM). Perhaps more surprisingly, SFB (with adaptation) performs well despite the complementarity assumption $X_S \perp\!\!\!\perp X_U | Y$ being violated. One explanation for this is that complementarity is only weakly violated in the test domain. Another is that complementarity is not *necessary* for SFB, with some weaker, yet-to-be-determined condition(s) sufficing. In App. I, we provide a more detailed explanation and discussion of this observation.

**ColorMNIST.** We now consider the ColorMNIST dataset [1], described in § 1 and Fig. 1a. We follow the experimental setup of Eastwood et al. [15, §6.1]; see App. G.3 for details. Table 3 shows that: (i) SFB learns a stable predictor ("no adpt.") with performance comparable to other stable/invariant

methods like IRM [1]; and (ii) only SFB (with adaptation) is capable of harnessing the spurious color feature in the test domain *without labels*, leading to a near-optimal boost in performance. Note that "Oracle no adpt." refers to an ERM model trained on grayscale images, while "Oracle" refers to an ERM model trained on labeled test-domain data. Table 6 of App. F.1.3 compares to additional baseline methods, including V-REx [35], EQRM [15], Fishr [48] and more. Fig. 2 gives more insight by showing performance across test domains of varying color-label correlation. On the left, we see that SFB outperforms ERM and IRM, as well as additional adaptive baseline methods in IRM + pseudo-labeling (PL, [36]) and IRM + T3A [30] (see App. G.1 for details). On the right, ablations show that: (i) bias-correction (BC), post-hoc calibration (CA), and multiple rounds of pseudo-labeling (Rn) improve adaptation performance; and (ii) without labels, SFB harnesses the spurious color feature near-optimally in test domains of varying color-label correlation—the original goal we set out to achieve in Fig. 1b. Further results and ablations are provided in App. F.1.

**PACS.** Table 2 shows that SFB's stable ("no adpt.") performance is comparable to that of the other stable/invariant methods (IRM, ACTIR). One exception is the sketch domain (S)—the most severe shift based on performance drop—where SFB's stable predictor performs best. Another is on domains A and C, where ACTIR performs better than SFB's stable predictor. Most notable, however, is: (i) the consistent performance boost that SFB gets from unsupervised adaptation; and (ii) SFB performing best or joint-best on 3 of the 4 domains. These results suggest SFB can be useful on real-world datasets where it is unclear if complementarity holds. In App. I, we discuss why this may be the case.

## 7 Conclusion & Future Work

This work demonstrated, both theoretically and practically, how to adapt our usage of spurious features to new test domains using only a stable, complementary training signal. By using invariant predictions to safely harness complementary spurious features, our proposed Stable Feature Boosting algorithm can provide significant performance gains compared to only using invariant/stable features or using unadapted spurious features—without requiring any true labels in the test domain.

**Stable and calibrated predictors.** Perhaps the greatest challenge in applying SFB in practice is the need for a stable and calibrated predictor. While stable features may be directly observable in some cases (e.g., using prior knowledge of causal relationships between the domain, features, and label, as in Prop. D.2), they often need to be extracted from high-dimensional observations (e.g., images). Several methods for stable-feature extraction have recently been proposed [1, 35, 58, 70, 15], with future improvements likely to benefit SFB. Calibrating complex predictors like deep neural networks is also an active area of research [18, 25, 72, 59], with future improvements likely to benefit SFB.

**Weakening the complementarity condition.** SFB also assumes that stable and unstable features are complementarity, i.e., conditionally independent given the label. This assumption is implicit in the causal generative models assumed by prior work [49, 68, 31], and future work may look to weaken it. However, our experimental results suggest that SFB may be robust to violations of complementarity in practice: on our synthetic data where complementarity does not hold (CE–DD) and real data where we have no reason to believe it holds (PACS), SFB still outperformed baseline methods. We discuss potential reasons for this in App. I and hope that future work can identify weaker sufficient conditions.

**Exploiting newly-available test-domain features without labels.** While we focused on domain generalization (DG) and the goal of (re)learning how to use the same spurious features (e.g., color) in a new way, our solution to the "marginal problem" in § 4.1 can be used to exploit a completely new set of (complementary) features in the test domain that weren't available in the training domains. For example, given a stable predictor of diabetes based on causal features (e.g., age, genetics), SFB could exploit new unlabeled data containing previously-unseen effect features (e.g., glucose levels). We hope future work can explore such uses of SFB.

## Acknowledgments and Disclosure of Funding

The authors thank Chris Williams and Ian Mason for providing feedback on an earlier draft, as well as the MPI Tübingen causality group for helpful discussions and comments. This work was supported by the Tübingen AI Center (FKZ: 01IS18039B) and by the Deutsche Forschungsgemeinschaft (DFG, German Research Foundation) under Germany's Excellence Strategy – EXC number 2064/1 – Project number 390727645. The authors declare no competing interests.

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

# Appendices

## Table of Contents

# A Proof and Further Discussion of Theorem 4.4

## A.1 Proof of Theorem 4.4

In this section, we prove our main results regarding the marginal generalization problem presented in Section 4, namely Thm. 4.4. For the reader's convenience, we restate Thm. 4.4 here:

**Theorem 4.4** (Marginal generalization with for binary labels and complementary features). *Consider three random variables $X_S$, $X_U$, and $Y$, where*

1. *$Y$ is binary ($\{0,1\}$-valued),*
2. *$X_S$ and $X_U$ are complementary features for $Y$ (i.e., $X_S \perp\!\!\!\perp X_U | Y$), and*
3. *$X_S$ is informative of $Y$ ($X_S \not\perp\!\!\!\perp Y$).*

*Then, the joint distribution of $(X_S, X_U, Y)$ can be written in terms of the joint distributions of $(X_S, Y)$ and $(X_S, X_U)$. Specifically, if $\widehat{Y}|X_S \sim \mathrm{Bernoulli}(\Pr[Y=1|X_S])$ is pseudo-label and*

$$\epsilon_0 := \Pr[\widehat{Y} = 0|Y = 0] \quad and \quad \epsilon_1 := \Pr[\widehat{Y} = 1|Y = 1] \tag{A.1}$$

*are the conditional probabilities that $\widehat{Y}$ and $Y$ agree, given $Y = 0$ and $Y = 1$, respectively, then,*

1. *$\epsilon_0 + \epsilon_1 > 1$,*
2. *$\Pr[Y = 1|X_U] = \dfrac{\Pr[\widehat{Y} = 1|X_U] + \epsilon_0 - 1}{\epsilon_0 + \epsilon_1 - 1}$, and*
3. *$\Pr[Y = 1|X_S, X_U] = \sigma\left(\mathrm{logit}(\Pr[Y = 1|X_S]) + \mathrm{logit}(\Pr[Y = 1|X_U]) - \mathrm{logit}(\Pr[Y = 1])\right)$.*

Before proving Thm. 4.4, we provide some examples demonstrating that the complementarity and informativeness assumptions in Thm. 4.4 cannot be dropped.

**Example A.1.** Suppose $X_S$ and $X_U$ have independent $\mathrm{Bernoulli}(1/2)$ distributions. Then, $X_S$ is informative of both of the binary variables $Y_1 = X_S X_U$ and $Y_2 = X_S(1 - X_U)$ and both have identical conditional distributions given $X_S$, but $Y_1$ and $Y_2$ have different conditional distributions given $X_U$:
$$\Pr[Y_1 = 1|X_U = 0] = 0 \neq 1/2 = \Pr[Y_2 = 1|X_U = 0].$$
Thus, the complementarity condition cannot be omitted.

On the other hand, $X_S$ and $X_U$ are complementary for both $Y_3 = X_U$ and an independent $Y_4 \sim \mathrm{Bernoulli}(1/2)$ and both $Y_3$ and $Y_4$ both have identical conditional distributions given $X_S$, but $Y_1$ and $Y_2$ have different conditional distributions given $X_U$:
$$\Pr[Y_3 = 1|X_U = 1] = 1/2 \neq 1 = \Pr[Y_4 = 1|X_U = 1].$$
Thus, the informativeness condition cannot be omitted.

Before proving Thm. 4.4, we prove Lemma 4.5, which allows us to safely divide by the quantity $\epsilon_0 + \epsilon_1 - 1$ in the formula for $\Pr[Y = 1|X_U]$, under the condition that $X_S$ is informative of $Y$.

**Lemma 4.5.** *In the setting of Thm. 4.4, let $\epsilon_0$ and $\epsilon_1$ be the class-wise pseudo-label accuracies defined in as in Eq. (A.1). Then, $\epsilon_0 + \epsilon_1 = 1$ if and only if $X_S$ and $Y$ are independent.*

Note that the entire result also holds, with almost identical proof, in the multi-environment setting of Sections 3 and 5, conditioned on a particular environment $E$.

*Proof.* We first prove the forward implication. Suppose $\epsilon_0 + \epsilon_1 = 1$. If $\Pr[Y = 1] \in \{0, 1\}$, then $X_S$ and $Y$ are trivially independent, so we may assume $\Pr[Y = 1] \in (0, 1)$. Then,

$$\begin{aligned}
\mathbb{E}[\widehat{Y}] &= \epsilon_1 \Pr[Y = 1] + (1 - \epsilon_0)(1 - \Pr[Y = 1]) && \text{(Law of Total Expectation)} \\
&= (\epsilon_0 + \epsilon_1 - 1)\Pr[Y = 1] + 1 - \epsilon_0 \\
&= 1 - \epsilon_0 && (\epsilon_0 + \epsilon_1 = 1) \\
&= \mathbb{E}[\widehat{Y}|Y = 0]. && \text{(Definition of } \epsilon_0)
\end{aligned}$$

Since $Y$ is binary and $\Pr[Y=1] \in (0,1)$, it follows that $\mathbb{E}[\widehat{Y}] = \mathbb{E}[\widehat{Y}|Y=0] = \mathbb{E}[\widehat{Y}|Y=1]$; i.e., $\mathbb{E}[\widehat{Y}|Y] \perp\!\!\!\perp Y$. Since $\widehat{Y}$ is binary, its distribution is specified entirely by its mean, and so $\widehat{Y} \perp\!\!\!\perp Y$. It follows that the covariance between $\widehat{Y}$ and $Y$ is 0:

$$
\begin{aligned}
0 &= \mathbb{E}[(Y - \mathbb{E}[Y])(\widehat{Y} - \mathbb{E}[\widehat{Y}])] \\
&= \mathbb{E}[\mathbb{E}[(Y - \mathbb{E}[Y])(\widehat{Y} - \mathbb{E}[\widehat{Y}])|X_S]] && \text{(Law of Total Expectation)} \\
&= \mathbb{E}[\mathbb{E}[Y - \mathbb{E}[Y]|X_S]\,\mathbb{E}[\widehat{Y} - \mathbb{E}[\widehat{Y}]|X_S]] && (Y \perp\!\!\!\perp \widehat{Y}|X_S) \\
&= \mathbb{E}[(\mathbb{E}[Y - \mathbb{E}[Y]|X_S])^2],
\end{aligned}
$$

where the final equality holds because $\widehat{Y}$ and $Y$ have identical conditional distributions given $X_S$. Since the $\mathcal{L}_2$ norm of a random variable is 0 if and only if the variable is 0 almost surely, it follows that, $P_{X_S}$-almost surely,

$$0 = \mathbb{E}[Y - \mathbb{E}[Y]|X_S] = \mathbb{E}[Y|X_S] - \mathbb{E}[Y],$$

so that $\mathbb{E}[Y|X_S] \perp\!\!\!\perp X_S$. Since $Y$ is binary, its distribution is specified entirely by its mean, and so $Y \perp\!\!\!\perp X_S$, proving the forward implication.

To prove the reverse implication, suppose $X_S$ and $Y$ are independent. Then $\widehat{Y}$ and $Y$ are also independent. Hence,

$$\epsilon_1 = \mathbb{E}[\widehat{Y}|Y=1] = \mathbb{E}[\widehat{Y}|Y=0] = 1 - \epsilon_0,$$

so that $\epsilon_0 + \epsilon_1 = 1$. $\qquad\square$

We now use Lemma 4.5 to prove Thm. 4.4:

*Proof.* To begin, note that $\widehat{Y}$ has the same conditional distribution given $X_S$ as $Y$ (i.e., $P_{\widehat{Y}|X_S} = P_{Y|X_S}$ and that $\widehat{Y}$ is conditionally independent of $Y$ given $X_S$ ($\widehat{Y} \perp\!\!\!\perp Y|X_S$). Then, since

$$\Pr[\widehat{Y}=1] = \mathbb{E}[\Pr[Y=1|X_S]] = \Pr[Y=1], \tag{A.2}$$

we have

$$
\begin{aligned}
\epsilon_1 = \Pr[\widehat{Y}=1|Y=1] &= \frac{\Pr\left[Y=1, \widehat{Y}=1\right]}{\Pr[Y=1]} && \text{(Definition of } \epsilon_1) \\[2mm]
&= \frac{\Pr\left[Y=1, \widehat{Y}=1\right]}{\Pr[\widehat{Y}=1]} && \text{(Eq. (A.2))} \\[2mm]
&= \frac{\mathbb{E}_{X_S}[\Pr\left[Y=1, \widehat{Y}=1|X_S\right]]}{\mathbb{E}_{X_S}[\Pr[\widehat{Y}=1|X_S]]} && \text{(Law of Total Expectation)} \\[2mm]
&= \frac{\mathbb{E}_{X_S}[\Pr[Y=1|X_S]\,\Pr[\widehat{Y}=1|X_S]]}{\mathbb{E}_{X_S}[\Pr[\widehat{Y}=1|X_S]]} && (\widehat{Y} \perp\!\!\!\perp Y|X_S) \\[2mm]
&= \frac{\mathbb{E}_{X_S}\left[(\Pr[Y=1|X_S])^2\right]}{\mathbb{E}_{X_S}[\Pr[Y=1|X_S]]} && (P_{\widehat{Y}|X_S} = P_{Y|X_S})
\end{aligned}
$$

entirely in terms of the conditional distribution $P_{Y|X_S}$ and the marginal distribution $P_{X_S}$. Similarly, $\epsilon_0$ can be written as $\epsilon_0 = \frac{\mathbb{E}_{X_S}\left[(\Pr[Y=0|X_S])^2\right]}{\mathbb{E}_{X_S}[\Pr[Y=0|X_S]]}$. Meanwhile, by the law of total expectation, and the assumption that $X_S$ (and hence $\widehat{Y}$) is conditionally independent of $X_U$ given $Y$, the conditional distribution $P_{\widehat{Y}|X_U}$ of $\widehat{Y}$ given $X_U$ can be written as

$$
\begin{aligned}
\Pr[\widehat{Y}&=1|X_U] \\
&= \Pr[\widehat{Y}=1|Y=0, X_U]\,\Pr[Y=0|X_U] + \Pr[\widehat{Y}=1|Y=1, X_U]\,\Pr[Y=1|X_U] \\
&= \Pr[\widehat{Y}=1|Y=0]\,\Pr[Y=0|X_U] + \Pr[\widehat{Y}=1|Y=1]\,\Pr[Y=1|X_U] \\
&= (1 - \epsilon_0)(1 - \Pr[Y=1|X_U]) + \epsilon_1\,\Pr[Y=1|X_U] \\
&= (\epsilon_0 + \epsilon_1 - 1)\,\Pr[Y=1|X_U] + 1 - \epsilon_0.
\end{aligned}
$$

By Lemma 4.5, the assumption $X_S \not\perp\!\!\!\perp Y$ implies $\epsilon_0 + \epsilon_1 \neq 1$. Hence, re-arranging the above equality gives us the conditional distribution $P_{Y|X_U}$ of $Y$ given $X_U$ purely in terms of the conditional $P_{Y|X_S}$ and $P_{X_S,X_U}$:

$$\Pr[Y = 1|X_U = X_U] = \frac{\Pr[\widehat{Y} = 1|X_U = X_U] + \epsilon_0 - 1}{\epsilon_0 + \epsilon_1 - 1}.$$

It remains now to write the conditional distribution $P_{Y|X_S,X_U}$ in terms of the conditional distributions $P_{Y|X_S}$ and $P_{Y|X_U}$ and the marginal $P_Y$. Note that

$$\begin{aligned}
\frac{\Pr[Y = 1|X_S, X_U]}{\Pr[Y = 0|X_S, X_U]} &= \frac{\Pr[X_S, X_U|Y = 1]\Pr[Y = 1]}{\Pr[X_S, X_U|Y = 0]\Pr[Y = 0]} && \text{(Bayes' Rule)} \\
&= \frac{\Pr[X_S|Y = 1]\Pr[X_U|Y = 1]\Pr[Y = 1]}{\Pr[X_S|Y = 0]\Pr[X_U|Y = 0]\Pr[Y = 0]} && \text{(Complementarity)} \\
&= \frac{\Pr[Y = 1|X_S]\Pr[Y = 1|X_U]\Pr[Y = 0]}{\Pr[Y = 0|X_S]\Pr[Y = 0|X_U]\Pr[Y = 1]}. && \text{(Bayes' Rule)}
\end{aligned}$$

It follows that the logit of $\Pr[Y = 1|X_S, X_U]$ can be written as the sum of a term depending only on $X_S$, a term depending only on $X_U$, and a constant term:

$$\begin{aligned}
\text{logit}\left(\Pr[Y = 1|X_S, X_U]\right) &= \log \frac{\Pr[Y = 1|X_S, X_U]}{1 - \Pr[Y = 1|X_S, X_U]} \\
&= \log \frac{\Pr[Y = 1|X_S, X_U]}{\Pr[Y = 0|X_S, X_U]} \\
&= \log \frac{\Pr[Y = 1|X_S]}{\Pr[Y = 0|X_S]} + \log \frac{\Pr[Y = 1|X_U]}{\Pr[Y = 0|X_U]} - \log \frac{\Pr[Y = 1]}{\Pr[Y = 0]} \\
&= \text{logit}\left(\Pr[Y = 1|X_S]\right) + \text{logit}\left(\Pr[Y = 1|X_U]\right) - \text{logit}\left(\Pr[Y = 1]\right).
\end{aligned}$$

Since the sigmoid $\sigma$ is the inverse of logit,

$$\Pr[Y = 1|X_S, X_U] = \sigma\left(\text{logit}\left(\Pr[Y = 1|X_S]\right) + \text{logit}\left(\Pr[Y = 1|X_U]\right) - \text{logit}\left(\Pr[Y = 1]\right)\right),$$

which, by Eq. (4.3), can be written in terms of the conditional distribution $P_{Y|X_S}$ and the joint distribution $P_{X_S,X_U}$. $\qquad\square$

### A.2 Further discussion of Theorem 4.4

**Connections to learning from noisy labels.** Thm. 4.4 leverages two theoretical insights about the special structure of pseudo-labels that complement results in the literature on learning from noisy labels. First, Blanchard et al. [7] showed that learning from noisy labels is possible if and only if the total noise level is below the critical threshold $\epsilon_0 + \epsilon_1 > 1$; in the case of learning from pseudo-labels, we show (see Lemma 4.5 in Appendix A.1) that this is satisfied if and only if $X_S$ is informative of $Y$ (i.e., $Y \not\perp\!\!\!\perp X_S$). Second, methods for learning under label noise commonly assume knowledge of $\epsilon_0$ and $\epsilon_1$ [44], which may be unrealistic in applications where we have absolutely no true (i.e., test-domain) labels; however, for pseudo-labels sampled from a known conditional probability distribution $P_{Y|X_S}$, one can express these noise levels in terms of $P_{Y|X_S}$ and $P_{X_S}$ and thereby estimate them *without any true labels*, as on line 3 of Alg. 1.

**Possible applications of Thm. 4.4 beyond domain adaptation** The reason we wrote Thm. 4.4 in the more general setting of the marginal problem rather than in the specific context of domain adaptation is that we envision possible applications to a number of problems besides domain adaptation. For example, suppose that, after learning a calibrated machine learning model $M_1$ using a feature $X_S$, we observe an additional feature $X_U$. In the case that $X_S$ and $X_U$ are complementary, Thm. 4.4 justifies using the student-teacher paradigm [11, 2, 27] to train a model for predicting $Y$ from $X_U$ (or from $(X_S, X_U)$ jointly) based on predictions from $M_1$. This could be useful if we don't have access to labeled pairs $(X_U, Y)$, or if retraining a model using $X_S$ would require substantial computational resources or access to sensitive or private data. Exploring such approaches could be a fruitful direction for future work

# B  Proof of Theorem 4.6

This appendix provides a proof of Thm. 4.6, which provides conditions under which our proposed domain adaptation procedure (Alg. 1) is consistent.

We state a formal version of Thm. 4.6:

**Theorem 4.6** (Consistency of the bias-corrected classifier). *Assume*

1. $X_S$ *is stable,*

2. $X_S$ *and $X_U$ are complementary, and*

3. $X_S$ *is informative of $Y$ (i.e., $X_S \not\perp Y$).*

*Let $\widehat{\eta}_n : \mathcal{X}_S \times \mathcal{X}_U \to [0,1]$ given by*

$$\widehat{\eta}_n(x_S, x_U) = \sigma\left(f_S(x_S) + \text{logit}\left(\frac{\widehat{\eta}_{U,n}(x_U) + \widehat{\epsilon}_{0,n} - 1}{\widehat{\epsilon}_{0,n} + \widehat{\epsilon}_{1,n} - 1}\right) - \beta_1\right), \quad \text{for all } (x_S, x_U) \in \mathcal{X}_S \times \mathcal{X}_U,$$

*denote the bias-corrected regression function estimate proposed in Alg. 1, and let $\widehat{h}_n : \mathcal{X}_S \times \mathcal{X}_U \to \{0,1\}$ given by*

$$\widehat{h}_n(x_S, x_U) = 1\{\widehat{\eta}(x_S, x_U) > 1/2\}, \quad \text{for all } (x_S, x_U) \in \mathcal{X}_S \times \mathcal{X}_U,$$

*denote the corresponding hard classifier. Let $\eta_U : \mathcal{X}_U \to [0,1]$, given by $\eta_U(x_U) = \Pr[Y = 1|X_U = x_U, E = 1]$ for all $x_U \in \mathcal{X}_U$, denote the true regression function over $X_U$, and let $\widehat{\eta}_{U,n}$ denote its estimate as assumed in Line 4 of Alg. 1. Then, as $n \to \infty$,*

(a) *if, for $P_{X_U}$-almost all $x_U \in \mathcal{X}_U$, $\widehat{\eta}_{U,n}(x_U)) \to \eta_U(x_U)$ in probability, then $\widehat{\eta}_n$ and $\widehat{h}_n$ are weakly consistent (i.e., $\widehat{\eta}_n(x_S, x_U) \to \eta(x_S, x_U)$ $P_{X_S, X_U}$-almost surely and $R(\widehat{h}_n) \to R(h^*)$ in probability).*

(b) *if, for $P_{X_U}$-almost all $x_U \in \mathcal{X}_U$, $\widehat{\eta}_{U,n}(x_U)) \to \eta_U(x_U)$ almost surely, then $\widehat{\eta}_n$ and $\widehat{h}_n$ are strongly consistent (i.e., $\widehat{\eta}_n(x_S, x_U) \to \eta(x_S, x_U)$ $P_{X_S, X_U}$-almost surely and $R(\widehat{h}_n) \to R(h^*)$ a.s.).*

Before proving Thm. 4.6, we provide a few technical lemmas. The first shows that almost-everywhere convergence of regression functions implies convergence of the corresponding classifiers in classification risk:

**Lemma B.1.** *Consider a sequence of regression functions $\eta, \eta_1, \eta_2, \ldots : \mathcal{X} \to [0,1]$. Let $h, h_1, h_2, \ldots : \mathcal{X} \to \{0,1\}$ denote the corresponding classifiers*

$$h(x) = 1\{\eta(x) > 1/2\} \quad \text{and} \quad h_i(x) = 1\{\eta_i(x) > 1/2\}, \quad \text{for all } i \in \mathbb{N}, x \in \mathcal{X}.$$

(a) *If $\eta_n(x) \to \eta(x)$ for $P_X$-almost all $x \in \mathcal{X}$ in probability, then $R(h_n) \to R(h^*)$ in probability.*

(b) *If $\eta_n(x) \to \eta(x)$ for $P_X$-almost all $x \in \mathcal{X}$ almost surely as $n \to \infty$, then $R(h_n) \to R(h)$ almost surely.*

*Proof.* Note that, since $h_n(x) \neq h(x)$ implies $|\eta_n(x) - \eta(x)| \geq |\eta(x) - 1/2|$,

$$1\{h_n(x) \neq h(x)\} \leq 1\{|\eta_n(x) - \eta(x)| \geq |\eta(x) - 1/2|\}. \tag{B.1}$$

We utilize this observation to prove both (a) and (b).

**Proof of (a)**  Let $\delta > 0$. By Inequality (B.1) and partitioning $\mathcal{X}$ based on whether $|2\eta(X) - 1| \leq \delta/2$,

$$\mathbb{E}_X\left[|2\eta(X) - 1|1\{h_n(X) \neq h(X)\}\right]$$
$$\leq \mathbb{E}_X\left[|2\eta(X) - 1|1\{|\eta_n(X) - \eta(X)| \geq |\eta(X) - 1/2|\}\right]$$
$$= \mathbb{E}_X\left[|2\eta(X) - 1|1\{|\eta_n(X) - \eta(X)| \geq |\eta(X) - 1/2|\}1\{|2\eta(X) - 1| > \delta/2\}\right]$$
$$\quad + \mathbb{E}_X\left[|2\eta(X) - 1|1\{|\eta_n(X) - \eta(X)| \geq |\eta(X) - 1/2|\}1\{|2\eta(X) - 1| \leq \delta/2\}\right]$$
$$\leq \mathbb{E}_X\left[1\{|\eta_n(X) - \eta(X)| > \delta/2\}\right] + \delta/2.$$

Hence,

$$\lim_{n\to\infty} \Pr_{\eta_n}\left[\mathbb{E}_X\left[|2\eta(X)-1|1\{h_n(X)\neq h(X)\}\right] > \delta\right]$$

$$\leq \lim_{n\to\infty} \Pr_{\eta_n}\left[\mathbb{E}_X\left[1\{|\eta_n(X)-\eta(X)| > \delta/2\}\right] > \delta/2\right]$$

$$\leq \lim_{n\to\infty} \frac{2}{\delta}\mathbb{E}_{\eta_n}\left[\mathbb{E}_X\left[1\{|\eta_n(X)-\eta(X)| > \delta/2\}\right]\right] \qquad \text{(Markov's Inequality)}$$

$$= \lim_{n\to\infty} \frac{2}{\delta}\mathbb{E}_X\left[\mathbb{E}_{\eta_n}\left[1\{|\eta_n(X)-\eta(X)| > \delta/2\}\right]\right] \qquad \text{(Fubini's Theorem)}$$

$$= \frac{2}{\delta}\mathbb{E}_X\left[\lim_{n\to\infty}\Pr_{\eta_n}\left[|\eta_n(X)-\eta(X)| > \delta/2\right]\right] \qquad \text{(Dominated Convergence Theorem)}$$

$$= 0. \qquad (\eta_n(X)\to\eta(X), P_X\text{-a.s., in probability})$$

**Proof of (b)** For any $x \in \mathcal{X}$ with $\eta(x) \neq 1/2$, if $\eta_n(x) \to \eta(x)$ then $1\{|\eta_n(x)-\eta(x)| \geq |\eta(x)-1/2|\} \to 0$. Hence, by Inequality (B.1), the dominated convergence theorem (with $|2\eta(x)-1|1\{|\eta_n(x)-\eta(x)| \geq |\eta(x)-1/2|\} \leq 1$), and the assumption that $\eta_n(x) \to \eta(x)$ for $P_X$-almost all $x \in \mathcal{X}$ almost surely,

$$\lim_{n\to\infty}\mathbb{E}_X\left[|2\eta(X)-1|1\{h_n(X)\neq h(X)\}\right]$$

$$\leq \lim_{n\to\infty}\mathbb{E}_X\left[|2\eta(X)-1|1\{|\eta_n(X)-\eta(X)| \geq |\eta(X)-1/2|\}\right]$$

$$= \mathbb{E}_X\left[\lim_{n\to\infty}|2\eta(X)-1|1\{|\eta_n(x)-\eta(x)| \geq |\eta(x)-1/2|\}\right]$$

$$= 0, \quad \text{almost surely.}$$

$\square$

Our next lemma concerns an edge case in which the features $X_S$ and $X_U$ provide perfect but contradictory information about $Y$, leading to Equation (4.4) being ill-defined. We show that this can happen only with probability 0 over $(X_S, X_U) \sim P_{X_S,X_U}$ can thus be safely ignored:

**Lemma B.2.** *Consider two predictors $X_S$ and $X_Y$ of a binary label $Y$. Then,*

$$\Pr_{X_S,X_U}\left[\mathbb{E}[Y|X_S] = 1 \text{ and } \mathbb{E}[Y|X_U] = 0\right] = \Pr_{X_S,X_U}\left[\mathbb{E}[Y|X_S] = 0 \text{ and } \mathbb{E}[Y|X_U] = 1\right] = 0.$$

*Proof.* Suppose, for sake of contradiction, that the event

$$A := \{(x_S, x_U) : \mathbb{E}[Y|X_S = x_S] = 1 \text{ and } \mathbb{E}[Y|X_U = x_U] = 0\}$$

has positive probability. Then, the conditional expectation $\mathbb{E}[Y|A]$ is well-defined, giving the contradiction

$$1 = \mathbb{E}_{X_S}[\mathbb{E}[Y|E, X_S]] = \mathbb{E}[Y|A] = \mathbb{E}_{X_U}[\mathbb{E}[Y|E, X_U]] = 0.$$

The case $\mathbb{E}[Y|X_S] = 0$ and $\mathbb{E}[Y|X_U] = 1$ is similar. $\square$

We now utilize Lemmas B.1 and B.2 to prove Thm. 4.6.

*Proof.* By Lemma B.1, it suffices to prove that $\widehat{\eta}(x_S, x_U) \to \eta(x_S, x_U)$, for $P_{X_S,X_U}$-almost all $(x_S, x_U) \in \mathcal{X}_S \times \mathcal{X}_U$, in probability (to prove (a)) and almost surely (to prove (b)).

**Finite case** We first consider the case when both $\Pr[Y|X_S = x_S], \Pr[Y|X_U = x_U] \in (0,1)$, so that $f_S(x_S)$ and $\text{logit}\left(\frac{\widetilde{\eta}(x_U)+\epsilon_0-1}{\epsilon_0+\epsilon_1-1}\right)$ are both finite. Since

$$\widehat{\eta}_{S,U}(x_S, x_U) - \eta_{S,U}(x_S, x_U)$$

$$= \sigma\left(f_S(x_S) + \text{logit}\left(\frac{\widehat{\eta}_{U,1}(x_U)+\widehat{\epsilon}_0-1}{\widehat{\epsilon}_0+\widehat{\epsilon}_1-1}\right) - \widehat{\beta}_{1,n}\right) - \sigma\left(f_S(x_S) + \text{logit}\left(\frac{\widetilde{\eta}(x_U)+\epsilon_0-1}{\epsilon_0+\epsilon_1-1}\right) - \beta_1\right),$$

where the sigmoid $\sigma : \mathbb{R} \to [0,1]$ is continuous, by the continuous mapping theorem and the assumption that $\widehat{\eta}_{U,1}(x_U) \to \widetilde{\eta}(x_U)$, to prove both of these, it suffices to show:

(i) $\widehat{\epsilon}_0 \to \epsilon_0$ and $\widehat{\epsilon}_1 \to \epsilon_1$ almost surely as $n \to \infty$.

(ii) $\widehat{\beta}_{1,n} \to \beta_1 \in (-\infty, \infty)$ almost surely as $n \to \infty$.

(iii) The mapping $(a, b, c) \mapsto \mathrm{logit}\left(\frac{a+b-1}{b+c-1}\right)$ is continuous at $(\widetilde{\eta}(x_U), \epsilon_0, \epsilon_1)$.

We now prove each of these in turn.

**Proof of (i)**   Since $\widehat{Y}_i \perp\!\!\!\perp Y_i | X_S$ and $0 < \Pr[\widehat{Y} = 1]$, by the strong law of large numbers and the continuous mapping theorem,

$$\widehat{\epsilon}_1 = \frac{1}{n_1}\sum_{i=1}^{n}\widehat{Y}_i\sigma(f_S(X_i)) = \frac{\frac{1}{n}\sum_{i=1}^{n}\widehat{Y}_i\sigma(f_S(X_i))}{\frac{1}{n}\sum_{i=1}^{n}\widehat{Y}_i} \to \frac{\mathbb{E}[\sigma(f_S(X))\mathbf{1}\{\widehat{Y}=1\}]}{\Pr[\widehat{Y}=1]} = \mathbb{E}[\sigma(f_S(X))|\widehat{Y}=1] = \epsilon_1,$$

almost surely as $n \to \infty$. Similarly, since $\Pr[\widehat{Y} = 0] = 1 - \Pr[\widehat{Y} = 1] > 0$, $\widehat{\epsilon}_0 \to \epsilon_0$ almost surely.

**Proof of (ii)**   Recall that

$$\widehat{\beta}_{1,n} = \mathrm{logit}\left(\frac{1}{n}\sum_{i=1}^{n}\widehat{Y}_i\right).$$

By the strong law of large numbers, $\frac{1}{n}\sum_{i=1}^{n}\widehat{Y}_i \to \Pr[\widehat{Y} = 1|E = 1] = \Pr[Y = 1|E = 1]$. Since we assumed $\Pr[Y = 1|E = 1] \in (0, 1)$, it follows that the mapping $a \mapsto \mathrm{logit}(a)$ is continuous at $a = \Pr[Y = 1|E = 1]$. Hence, by the continuous mapping theorem, $\widehat{\beta}_{1,n} \to \mathrm{logit}\left(\Pr[Y = 1|E = 1]\right) = \beta_1$ almost surely.

**Proof of (iii)**   Since the logit function is continuous on the open interval $(0, 1)$ and we assumed $\epsilon_0 + \epsilon_1 > 1$, it suffices to show that $0 < \widetilde{\eta}(x_U) + \epsilon_0 - 1 < \epsilon_0 + \epsilon_1 - 1$. Since, according to Thm. 4.4,

$$\widetilde{\eta}(x_U) = (\epsilon_0 + \epsilon_1 - 1)\eta^*(x_U)) + 1 - \epsilon_0,$$

this holds as long as $0 < \eta^*(x_U) < 1$, as we assumed for $P_{X_U}$-almost all $x_U \in \mathcal{X}_U$.

**Infinite case**   We now address the case where either $\Pr[Y|X_S = x_S] \in \{0, 1\}$ or $\Pr[Y|X_U = x_U] \in \{0, 1\}$. By Lemma B.2, only one of these can happen at once, $P_{X_S, X_U}$-almost surely. Hence, since $\lim_{n\to\infty}\widehat{\beta}_{1,n}$ is also finite almost surely, if $\Pr[Y|X_S = x_S] \in \{0, 1\}$, then $\widehat{\eta}(x_S, x_U) = \sigma(\mathrm{logit}(\Pr[Y|X_S = x_S])) = \eta(x_S, x_U)$, while, if $\Pr[Y|X_U = x_U] \in \{0, 1\}$, then $\widehat{\eta}(x_S, x_U) \to \sigma\left(\mathrm{logit}(\Pr[Y|X_U = x_U])\right) = \eta(x_S, x_U)$, in probability or almost surely, as appropriate. $\square$

## C   Multiclass Case

In the main paper, to simplify notation, we presented our unsupervised test-domain adaptation method in the case of binary labels $Y$. However, in many cases, including several of our experiments in Section 6, the label $Y$ can take more than 2 distinct values. Hence, in this section, we show how to generalize our method to the multiclass setting and then present the exact procedure (Alg. 2) used in our multiclass experiments in Section 6.

Suppose we have $K \geq 2$ classes. We "one-hot encode" these classes, so that $Y$ takes values in the set

$$\mathcal{Y} = \{(1, 0, ..., 0), (0, 1, 0, ..., 0), ..., (0, ..., 0, 1)\} \subseteq \{0, 1\}^K.$$

Let $\epsilon \in [0, 1]^{\mathcal{Y} \times \mathcal{Y}}$ with

$$\epsilon_{y,y'} = \Pr[\widehat{Y} = y | Y = y']$$

denote the class-conditional confusion matrix of the pseudo-labels. Then, we have

$$\mathbb{E}[\widehat{Y}|X_U] = \sum_{y\in\mathcal{Y}} \mathbb{E}[\widehat{Y}|Y = y, X_U]\Pr[Y = y|X_U] \qquad \text{(Law of Total Expectation)}$$

$$= \sum_{y\in\mathcal{Y}} \mathbb{E}[\widehat{Y}|Y = y]\Pr[Y = y|X_U] \qquad \text{(Complementary)}$$

$$= \epsilon\,\mathbb{E}[Y|X_U]. \qquad \text{(Definition of } \epsilon\text{)}$$

When $\epsilon$ is non-singular, this has the unique solution $\mathbb{E}[Y|X_U] = \epsilon^{-1} \mathbb{E}[\widehat{Y}|X_U]$, giving a multiclass equivalent of Eq. (4.3) in Thm. 4.4. In practice, however, it is numerically more stable to estimate $\mathbb{E}[Y|X_U]$ by the least-squares solution

$$\arg\min_{p \in \Delta^{\mathcal{Y}}} \left\| \epsilon p - \mathbb{E}[\widehat{Y}|X_U] \right\|_2,$$

which is what we will do in Algorithm 2. To estimate $\epsilon$ without observing the label $Y$ in the test domain, note that

$$\begin{aligned}
\epsilon_{y,y'} = \Pr[\widehat{Y} = y | Y = y'] &= \frac{\Pr[\widehat{Y} = y, Y = y']}{\Pr[Y = y']} \\
&= \frac{\mathbb{E}\left[\Pr[\widehat{Y} = y, Y = y'|X_S]\right]}{\mathbb{E}\left[\Pr[Y = y'|X_S]\right]} \\
&= \frac{\mathbb{E}\left[\Pr[\widehat{Y} = y|X_S]\Pr[Y = y'|X_S]\right]}{\mathbb{E}\left[\Pr[Y = y'|X_S]\right]} \\
&= \frac{\mathbb{E}\left[f_{1,y}(X_S)f_{1,y'}(X_S)\right]}{\mathbb{E}\left[f_{1,y'}(X_S)\right]}.
\end{aligned}$$

This suggests the estimate

$$\widehat{\epsilon}_{y,y'} = \frac{\sum_{i=1}^{n} \widehat{f}_{S,y}(X_{S,i})\widehat{f}_{S,y'}(X_{S,i})}{\sum_{i=1}^{n} \widehat{f}_{S,y'}(X_{S,i})} = \sum_{i=1}^{n} \widehat{f}_{S,y}(X_{S,i}) \frac{\widehat{f}_{S,y'}(X_{S,i})}{\sum_{i=1}^{n} \widehat{f}_{S,y'}(X_{S,i})}$$

of each $\epsilon_{y,y'}$, or, in matrix notation,

$$\widehat{\epsilon} = f_S^{\mathsf{T}}(X_S) \operatorname{Normalize}(f_S(X_S)),$$

where $\operatorname{Normalize}(X)$ scales each column of $X$ to sum to 1. This gives us a multiclass equivalent of Line 3 in Alg. 1.

The multiclass versions of Eq. (4.4) and Line 6 of Alg. 1 are slightly less straightforward. Specifically, whereas, in the binary case, we used the fact that $\Pr[X_S, X_U | Y \neq 1] = \Pr[X_S, X_U | Y = 0] = \Pr[X_S | Y = 0] \Pr[X_U | Y = 0] = \Pr[X_S | Y \neq 1] \Pr[X_U | Y \neq 1]$ (by complementarity), in the multiclass case, we do not have $\Pr[X_S, X_U | Y \neq 1] = \Pr[X_S | Y \neq 1] \Pr[X_U | Y \neq 1]$. However, following similar reasoning as in the proof of Thm. 4.4, we have

$$\begin{aligned}
\frac{\Pr[Y = y | X_S, X_U, E]}{\Pr[Y \neq y | X_S, X_U, E]} &= \frac{\Pr[Y = y | X_S, X_U, E]}{\sum_{y' \neq y} \Pr[Y = y' | X_S, X_U, E]} \\
&= \frac{\Pr[X_S, X_U | Y = y, E] \Pr[Y = y | E]}{\sum_{y' \neq y} \Pr[Y \neq y | X_S, X_U, E] \Pr[Y = y' | E]} && \text{(Bayes' Rule)} \\
&= \frac{\Pr[X_S | Y = y, E] \Pr[X_U | Y = y, E] \Pr[Y = y | E]}{\sum_{y' \neq y} \Pr[X_S | Y = y', E] \Pr[X_U | Y = y', E] \Pr[Y = y' | E]} && (X_S \perp\!\!\!\perp X_U | Y) \\
&= \frac{\Pr[Y = y | X_S, E] \Pr[Y = y | X_U, E]}{\sum_{y' \neq y} \Pr[Y = y' | X_S, E] \Pr[Y = y' | X_U, E] \cdot \frac{\Pr[Y=y|E]}{\Pr[Y=y'|E]}}. && \text{(Bayes' Rule)}
\end{aligned}$$

Hence,

$$\begin{aligned}
\operatorname{logit}(\Pr[Y = y | X_S, X_U, E]) &= \log\left( \frac{\Pr[Y = y | X_S, E] \Pr[Y = y | X_U, E]}{\sum_{y' \neq y} \Pr[Y = y' | X_S, E] \Pr[Y = y' | X_U, E] \cdot \frac{\Pr[Y=y|E]}{\Pr[Y=y'|E]}} \right) \\
&= \log\left( \frac{Q_y}{\sum_{y' \neq y} Q_{y'}} \right) = \log\left( \frac{\frac{Q_y}{\|Q\|_1}}{\sum_{y' \neq y} \frac{Q_{y'}}{\|Q\|_1}} \right) = \operatorname{logit}\left( \frac{Q_y}{\|Q\|_1} \right),
\end{aligned}$$

for $Q \in \mathbb{R}^{\mathcal{Y}}$ defined by

$$Q_y = \frac{f_{S,y}(X_S) f_{U,y}(X_U)}{\Pr[Y = y]} \quad \text{for each } y \in \mathcal{Y}.$$

In particular, applying the sigmoid function to each side, we have

$$\Pr[Y|X_S, X_U] = \frac{Q}{\|Q\|_1}.$$

We can estimate $Q_y$ by

$$\widehat{Q}_y = \frac{f_{S,y}(X_S) f_{U,y}(X_U)}{\frac{1}{n} \sum_{i=1}^{n} f_{S,y}(X_{S,i})}.$$

In matrix notation, this is

$$\widehat{Q} = \frac{f_S(X_S) \circ f_U(X_U)}{\frac{1}{n} \sum_{i=1}^{n} f_S(X_{S,i})},$$

where $\circ$ denotes element-wise multiplication. It follows that, for $p \in \Delta^{\mathcal{Y}}$ (we will use $p_y = \Pr[Y = y]$), we can use the multiclass combination function $C : \Delta^{\mathcal{Y}} \times \Delta^{\mathcal{Y}} \to \Delta^{\mathcal{Y}}$ with

$$C_p(p_S, p_U) = \text{Normalize}\left(\frac{p_S p_U}{p}\right), \tag{C.1}$$

where the multiplication and division are performed element-wise and $\text{Normalize}(x) = \frac{x}{\|x\|_1}$, to generalize Eq. (4.5). Putting these derivations together gives us our multiclass version of Alg. 1, presented in Alg. 2, where $\Delta^{\mathcal{Y}} = \{z \in [0,1]^K : \sum_{y \in \mathcal{Y}} z_y = 1\}$ denotes the standard probability simplex over $\mathcal{Y}$.

---

**Algorithm 2:** Multiclass bias-corrected adaptation procedure.

---

**Input:** Calibrated stable classifier $f_S : \mathcal{X} \to \Delta^{\mathcal{Y}}$ with $f_{S,y}(x_S) = \Pr[Y = y|X_S = x_S]$, $n$ unlabeled samples $\{(X_{S,i}, X_{U,i})\}_{i=1}^{n}$

**Output:** Joint classifier $\widehat{f} : \mathcal{X}_S \times \mathcal{X}_U \to \Delta^{\mathcal{Y}}$ estimating $\Pr[Y = y|X_S = x_S, X_U = x_U]$

1 Compute soft pseudo-labels $\{\widehat{Y}_i\}_{i=1}^{n}$ with $\widehat{Y}_i = f_S(X_{S,i})$
2 Compute soft class counts $\widehat{n} = \sum_{i=1}^{n} \widehat{Y}_i$
3 Estimate class-conditional pseudo-label confusion matrix $\widehat{\epsilon} \leftarrow f_S^{\mathsf{T}}(X_S) \text{Normalize}(f_S^{\mathsf{T}}(X_S))$
4 Fit unstable classifier $\widetilde{f}_U(x_U)$ to pseudo-labelled data $\{(X_{U,i}, \widehat{Y}_i)\}_{i=1}^{n}$     // $\approx \Pr[\widehat{Y} = y|X_U]$
5 Bias-correction $\widehat{f}_U(x_U) \mapsto \arg\min_{p \in \Delta^{\mathcal{Y}}} \|\epsilon p - \widetilde{f}_U(x_U)\|_2$     // $\approx \Pr[Y = y|X_U]$
6 **return** $\widehat{f}(x_S, x_U) \mapsto C_{\widehat{n}/n}(f_S(x_S), \widehat{f}_U(x_U))$     // Eq. (C.1), $\approx \Pr[Y = y|X_S, X_U]$

---

# D    Supplementary Results

## D.1    Trivial solution to joint-risk minimization

In Prop. D.1 below, we assume that the stable $f_S(X)$ and unstable $f_U(X)$ predictors output *logits*. In contrast, throughout the rest of the paper, we assume that $f_S(X)$ and $f_U(X)$ output *probabilities* in $[0, 1]$.

**Proposition D.1.** *Suppose* $\widehat{Y}|f_S(X) \sim \text{Bernoulli}(\sigma(f_S(X)))$, *such that* $\widehat{Y} \perp\!\!\!\perp f_U(X)|f_S(X)$. *Then,*

$$0 \in \arg\min_{f_U : \mathcal{X} \to \mathbb{R}} \mathbb{E}[\ell(\widehat{Y}, \sigma(f_S(X) + f_U(X)))],$$

*where* $\ell(x, y) = -x \log y - (1 - x) \log(1 - y)$ *denotes the cross-entropy loss.*

*Proof.* Suppose $\widehat{Y}|f_S(X) \sim \text{Bernoulli}(\sigma(f_S(X)))$, such that $\widehat{Y} \perp\!\!\!\perp f_U(X)|f_S(X)$. Then,

$$
\begin{aligned}
&- \mathbb{E}[\ell(\widehat{Y}, \sigma(f_S(X) + f_U(X)))] \\
&= \mathbb{E}[\mathbb{E}[\ell(\widehat{Y}, \sigma(f_S(X) + f_U(X)))]] \qquad\qquad\qquad \text{(Law of Total Expectation)} \\
&= \mathbb{E}[\mathbb{E}[\widehat{Y} \log \sigma(f_S(X) + f_U(X)) \\
&\qquad + (1 - Y) \log(1 - \sigma(f_S(X) + f_U(X)))|f_S(X)]] \\
&= \mathbb{E}[\mathbb{E}[\widehat{Y}|f_S(X_S)]\mathbb{E}[\log \sigma(f_S(X) + f_U(X))|f_S(X_S)] \\
&\qquad + \mathbb{E}[(1 - \widehat{Y})|f_S(X_S)]\mathbb{E}[\log(1 - \sigma(f_S(X) + f_U(X)))|f_S(X)]] \qquad (\widehat{Y} \perp\!\!\!\perp f_U(X)|f_S(X)) \\
&= \mathbb{E}[\sigma(f_S(X)) \log \sigma(f_S(X) + f_U(X)) \\
&\qquad + (1 - \sigma(f_S(X))) \log(1 - \sigma(f_S(X) + f_U(X)))]. \qquad (\widehat{Y}|f_S(X) \sim \text{Bernoulli}(\sigma(f_S(X)))).
\end{aligned}
$$

Since the cross-entropy loss is differentiable and convex, any $f_U(X)$ satisfying $0 = \frac{d}{df_U(X)}\mathbb{E}[\ell(\widehat{Y}, f_S(X) + f_U(X))]$ is a minimizer. Indeed, under the mild assumption that the expectation and derivative commute, for $f_U(X) = 0$,

$$
\begin{aligned}
\frac{d}{df_U(X)}\mathbb{E}[\ell(\widehat{Y}, \sigma(f_S(X) + f_U(X)))] &= -\mathbb{E}\left[\frac{\sigma(f_S(X))}{\sigma(f_S(X) + f_U(X))} + \frac{1 - \sigma(f_S(X))}{1 - \sigma(f_S(X) + f_U(X))}\right] \\
&= -\mathbb{E}\left[\frac{\sigma(f_S(X))}{\sigma(f_S(X))} + \frac{1 - \sigma(f_S(X))}{1 - \sigma(f_S(X))}\right] = 0.
\end{aligned}
$$

$\square$

## D.2 Causal perspectives

The stability, complementarity, and informativeness assumptions in Thm. 4.4 can be interpreted as constraints on the causal relationships between the variables $X_S$, $X_U$, $Y$, and $E$. We conclude this section with a result with a characterization of causal, directed acyclic graphs (DAGs) that are consistent with these assumptions. In particular, this result shows that our assumptions are satisfied in the "anti-causal" and "cause-effect" settings assumed in prior work [49, 68, 31], as well as work assuming only covariate shift (i.e., changes in the distribution of $X$ without changes in the conditional $P_{Y|X}$).

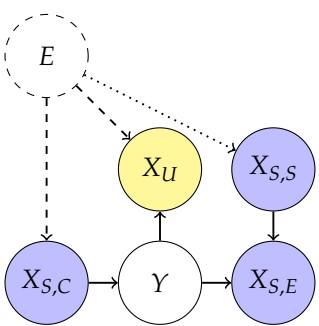

Figure 3: Causal DAGs over the environment $E$, three types of stable features (causes $X_{S,C}$, effects $X_{S,E}$, and spouses $X_{S,S}$), unstable features $X_U$, and label $Y$, under conditions 1)-6). At least one, and possibly both, of the dashed edges $E \to X_{S,C}$ and $E \to X_U$ must be included. The dotted edge $E \to X_{S,S}$ may or may not be included.

**Proposition D.2** (Possible Causal DAGs). *Consider an environment variable $E$, two covariates $X_U$ and $X_S$, and a label $Y$. Assume there are no other hidden confounders (i.e., causal sufficiency). First, assume:*

1) *$E$ is a root (i.e., none of $X_U$, $X_S$, and $Y$ is an ancestor of $E$).*
2) *$X_S$ is informative of $Y$ (i.e., $X_S \not\perp\!\!\!\perp Y|E$).*
3) *$X_S$ and $X_U$ are complementary predictors of $Y$; i.e., $X_S \perp\!\!\!\perp X_U|(Y, E)$.*
4) *$X_S$ is stable (i.e., $E \perp\!\!\!\perp Y|X_S$).*

*These are the four structural assumptions under which Theorems 4.4 and 4.6 show that the SFB algorithm learns the conditional distribution $P_{Y|X_S,X_U}$ in the test domain. Additionally, suppose*

5) *$X_U$ is unstable (i.e., $E \not\perp\!\!\!\perp Y|X_U$), This is the case in which empirical risk minimization [ERM; 65] may suffer bias due to distribution shift, and hence when SFB may outperform ERM.*
6) *$X_U$ contains some information about $Y$ that is not included in $X_S$ (i.e., $X_U \not\perp\!\!\!\perp Y|X_S$). This is information we expect invariant risk minimization [IRM; 1] is unable to learn, and hence when we expect SFB to outperform IRM.*

*Then, $X_U$ consists of causal descendants ("effects") of $Y$, while three types of stable features are possible:*

1. *causal ancestors $X_{S,C}$ of $Y$,*

2. *causal descendants $X_{S,E}$ of $Y$ that are not also descendants of $E$,*

3. *causal spouses $X_{S,S}$ of $Y$ (i.e., causal ancestors of $X_{S,E}$).*

Notable special cases of the DAG in Figure 3 include:

1. the "cause-effect" settings, studied by Rojas-Carulla et al. [49], von Kügelgen et al. [68, 69], where $X_S$ is a cause of $Y$, $X_U$ is an effect of $Y$, and $E$ may affect both $X_S$ and $X_U$ but may affect $Y$ only indirectly through $X_S$. Note that this generalizes the commonly used "covariate shift" assumption, as not only the covariate distribution $P_{X_S,X_U}$ but also the conditional distribution $P_{Y|X_U}$ can change between environments.

2. the "anti-causal" setting, studied by Jiang and Veitch [31], where $X_S$ and $X_U$ are both effects of $Y$, but $X_S$ is unaffected by $E$.

3. the widely studied "covariate shift" setting [62, 23, 6, 61], which corresponds (see Sections 3 and 5 of Schölkopf [55]) to a causal factorization $P(X,Y) = P(X)P(Y|X)$ (i.e., in which the only stable components $X_S$ are causes $X_{S,C}$) of $Y$ or unconditionally independent (e.g., causal spouses $X_{S,S}$)) of $Y$.

However, this model is more general than these special cases. Also, for sake of simplicity, we assumed causal sufficiency here; however, in the presence of unobserved confounders, other types of stable features are also possible; for example, if we consider the possibility of unobserved confounders $U$ influencing $Y$ that are independent of $E$ (i.e., invariant across domains), then our method can also utilize stable features that are descendants of $U$ (i.e., "siblings" of $Y$).

## E   Datasets

In our experiments, we consider five datasets: two (synthetic) numerical datasets and three image datasets. We now describe each dataset.

**Synthetic: Anti-causal (AC).**   We consider an anti-causal synthetic dataset based on that of Jiang and Veitch [31, §6.1] where data is generated according to the following structural equations:

$$Y \leftarrow \text{Rad}(0.5);$$
$$X_S \leftarrow Y \cdot \text{Rad}(0.75);$$
$$X_U \leftarrow Y \cdot \text{Rad}(\beta^e),$$

where the input $X = (X_S, X_U)$ and $\text{Rad}(\beta)$ denotes a Rademacher random variable that is $-1$ with probability $1 - \beta$ and $+1$ with probability $\beta$. Following Jiang and Veitch [31, §6.1], we create two training domains with $\beta_e \in \{0.95, 0.7\}$, one validation domain with $\beta_e = 0.6$ and one test domain with $\beta_e = 0.1$.

**Synthetic: Cause-effect with a direct $X_S$-$X_U$ dependence (CE-DD).**   We also consider a synthetic cause-effect dataset in which there is a direct dependence between $X_S$ and $X_U$. In particular, following Jiang and Veitch [31, App. B], data is generated according to the following structural equations:

$$X_S \leftarrow \text{Bern}(0.5);$$
$$Y \leftarrow \text{XOR}(X_S, \text{Bern}(0.75));$$
$$X_U \leftarrow \text{XOR}(\text{XOR}(Y, \text{Bern}(\beta_e)), X_S),$$

where the input $X = (X_S, X_U)$ and $\text{Bern}(\beta)$ denotes a Bernoulli random variable that is 1 with probability $\beta$ and 0 with probability $1 - \beta$. Note that $X_S \not\perp\!\!\!\perp X_U | Y$, since $X_S$ directly influences $X_U$. Following Jiang and Veitch [31, App. B], we create two training domains with $\beta_e \in \{0.95, 0.8\}$, one validation domain with $\beta_e = 0.2$, and one test domain with $\beta_e = 0.1$.

**ColorMNIST.** We next consider the `ColorMNIST` dataset [1]. This transforms the original `MNIST` dataset into a binary classification task (digit in 0–4 or 5–9) and then: (i) flips the label with probability 0.25, meaning that, across all 3 domains, digit shape correctly determines the label with probability 0.75; and (ii) colorizes the digit such that digit color (red or green) is a more informative but spurious feature (see Fig. 4).

**PACS.** We next consider the `PACS` dataset [37]—a 7-class image-classification dataset consisting of 4 domains: photos (P), art (A), cartoons (C) and sketches (S), with examples shown in Fig. 4. Model performances are reported for each domain after training on the other 3 domains.

**Camelyon17.** Finally, in the additional experiments of App. F.2, we consider the `Camelyon17` [3] dataset from the WILDS benchmark [33]: a medical dataset with histopathology images from 5 hospitals which use different staining and imaging techniques (see Fig. 4). The goal is to determine whether or not a given image contains tumor tissue, making it a binary classification task across 5 domains (3 training, 1 validation, 1 test).

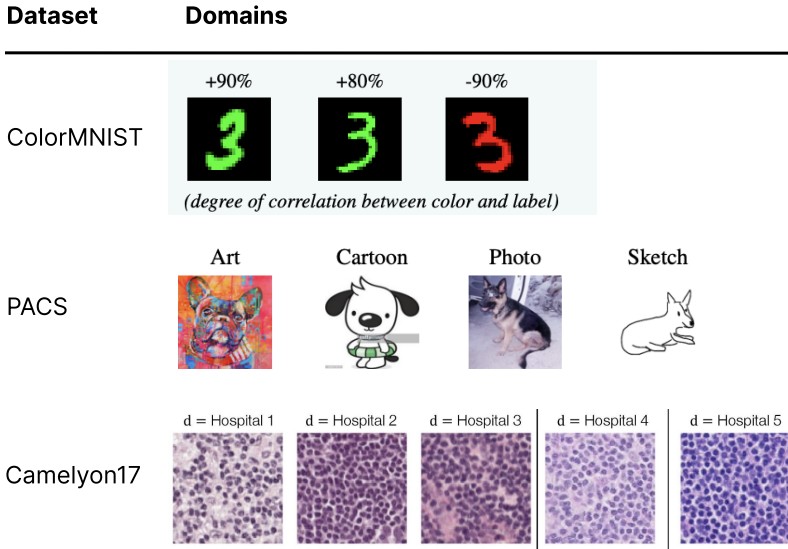

Figure 4: Examples from `ColorMNIST` [1], `PACS` [37] and `Camelyon17` [3]. Figure and examples based on Gulrajani and Lopez-Paz [24, Table 3] and Koh et al. [33, Figure 4]. For `ColorMNIST`, we follow the standard approach [1] and use the first two domains for training and the third for testing. For `PACS` [37], we follow the standard approach [37, 24] and use each domain in turn for testing, using the remaining three domains for training. For `Camelyon17` [3], we follow WILDS [33] and use the first three domains for training, the fourth for validation, and the fifth for testing.

# F  Further Experiments

This appendix provides further experiments which supplement those in the main text. In particular, it provides: (i) an ablation on the `ColorMNIST` dataset showing the effects of bias correction, post-hoc calibration and multiple rounds of pseudo-labelling on SFB's performance (F.1.1); (ii) the

performance of SFB on the `ColorMNIST` dataset when using different stability penalties (F.1.2); and (iii) results on a real-world medical dataset, `Camelyon17` [3], where we find that all methods perform similarly *when properly tuned* (F.2).

## F.1 ColorMNIST

We now provide ablations on the `ColorMNIST` dataset to illustrate the effectiveness of the different components of SFB. In particular, we focus on bias correction and calibration, while also showing how multiple rounds of pseudo-labeling can improve performance in practice.

### F.1.1 Ablations

**Bias correction.** To adapt the unstable classifier in the test domain, SFB employs the bias-corrected adaptation algorithm of Alg. 1 (or Alg. 2 for the multi-class case) which corrects for biases caused by possible disagreements between the stable-predictor pseudo-labels $\widehat{Y}$ and the true label $Y$. In this (sub)section, we investigate the performance of SFB with and without bias correction (BC).

**Calibration.** As discussed in § 4.2, correctly combining the stable and unstable predictions post-adaptation requires them to be properly calibrated. In particular, it requires the stable predictor $f_S$ to be calibrated with respect to the true labels $Y$ and the unstable predictor $f_U$ to be calibrated with respect to the pseudo-labels $\widehat{Y}$. In this (sub)section, we investigate the performance of SFB with and without post-hoc calibration (in particular, simple temperature scaling [25]). More specifically, we investigate the effect of calibrating the stable predictor (CS) and calibrating the unstable predictor (CU).

**Multiple rounds of pseudo-labeling.** While SFB learns the optimal unstable classifier $h_U^e$ in the test domain *given enough unlabelled data*, § 4.1 discussed how more accurate pseudo-labels $\widehat{Y}$ improve the sample efficiency of SFB. In particular, in a restricted-sample setting, more accurate pseudo-labels result in an unstable classifier $h_U^e$ which better harnesses $X_U$ in the test domain. With this in mind, note that, after adapting, we expect the joint predictions of SFB to be more accurate than its stable-only predictions. This raises the question: can we use these improved predictions to form more accurate pseudo-labels, and, in turn, an unstable classifier $h_U^e$ that leads to even better performance? Furthermore, can we repeat this process, using multiple rounds of pseudo-labelling to refine our pseudo-labels and ultimately $h_U^e$? While this multi-round approach loses the asymptotic guarantees of § 4.2, we found it to work quite well in practice. In this (sub)section, we thus investigate the performance of SFB with and without multiple rounds of pseudo-labeling (PL rounds).

Table 4: SFB ablations on `CMNIST`. Means and standard errors are over 3 random seeds. *BC:* bias correction. *CS:* post-hoc calibration of the stable classifier. *CU:* post-hoc calibration of the unstable classifier. *PL Rounds:* Number of pseudo-labeling rounds used. *GT adpt:* "ground-truth" adaptation using true labels in the test domain.

| Algorithm | Bias Correction | Calibration Stable | Calibration Unstable | PL Rounds | Test Acc. |
|---|---|---|---|---|---|
| SFB no adpt. | | | | 1 | $70.6 \pm 1.8$ |
| SFB | | | | 1 | $78.0 \pm 2.9$ |
| +BC | ✓ | | | 1 | $83.4 \pm 2.8$ |
| +CS | | ✓ | | 1 | $80.6 \pm 3.4$ |
| +CU | | | ✓ | 1 | $76.6 \pm 2.4$ |
| +BC+CS+CU | ✓ | ✓ | ✓ | 1 | $84.4 \pm 2.2$ |
| +BC+CS | ✓ | ✓ | | 1 | $84.9 \pm 2.6$ |
| +BC+CS | ✓ | ✓ | | 2 | $87.4 \pm 1.9$ |
| +BC+CS | ✓ | ✓ | | 3 | $88.1 \pm 1.8$ |
| +BC+CS | ✓ | ✓ | | 4 | $88.6 \pm 1.3$ |
| +BC+CS | ✓ | ✓ | | 5 | $88.7 \pm 1.3$ |
| SFB GT adpt. | ✓ | ✓ | | 1 | $89.0 \pm 0.3$ |

**Results.** Table 4 reports the ablations of SFB on `ColorMNIST`. Here we see that: (i) bias correction significantly boosts performance (+BC); (ii) calibrating the stable predictor also boosts performance without (+CS) and with (+BC+CS) bias correction, with the latter leading to the best performance; (iii) calibrating the unstable predictor (with respect to the pseudo-labels) slightly hurts performance without (+CU) and with (+BC+CS+CU) bias correction and stable-predictor calibration; (iv) multiple rounds of pseudo-labeling boosts performance, while also reducing the performance variation across random seeds; (v) using bias correction, stable-predictor calibration and 5 rounds of pseudo-labeling results in near-optimal adaptation performance, as indicated by the similar performance of SFB when using true labels $Y$ to adapt $h_U^e$ (denoted "SFB GT adpt." in Table 4).

### F.1.2 Different stability penalties

In our experiments of § 6, we used IRM for the stability term of our SFB method, given in Eq. (5.1). However, as discussed in § 5, many other approaches exist for enforcing stability [35, 58, 47, 15, 67, 40, 77], and, in principle, any of these could be used. To illustrate this point, we now evaluate the performance of SFB when using different stability penalties, namely IRM [1], VREx [35], EQRM [15] and CLOvE [70]. For all penalties, we use SFB with bias correction, post-hoc calibration of the stable predictor, and 5 rounds of pseudo-labeling (see the ablation study of App. F.1.1).

Table 5: `CMNIST` test-domain accuracies for SFB with different stability penalties. Shown are the mean and standard error over 10 seeds.

| Algorithm | Without Adaptation | With Adaptation |
|---|---|---|
| SFB w. IRM | $70.6 \pm 1.8$ | $88.7 \pm 1.3$ |
| SFB w. VREx | $72.5 \pm 1.0$ | $88.7 \pm 1.5$ |
| SFB w. EQRM | $69.0 \pm 2.8$ | $88.2 \pm 2.5$ |
| SFB w. CLOvE | $67.0 \pm 3.7$ | $77.0 \pm 6.6$ |

### F.1.3 Full results

We now provide extended/full results of those provided in the main text. In particular, Table 6 represents an extended version of Table 3 in the main text, comparing against many more baseline methods. In addition, Table 7 provides the full numerical results for all adaptive baseline methods (described in App. G.1), which correspond to the plots of Fig. 2 in the main text.

Table 6: `CMNIST` test-domain accuracies. Mean and standard error are over 10 seeds. Extended/full version of Table 3 in the main text.

| Algorithm | Test Acc. |
|---|---|
| ERM | $27.9 \pm 1.5$ |
| GroupDRO [53] | $29.0 \pm 1.1$ |
| IRM [1] | $69.7 \pm 0.9$ |
| SD [46] | $70.3 \pm 0.6$ |
| IGA [57] | $57.7 \pm 3.3$ |
| Fishr [48] | $70.1 \pm 0.7$ |
| V-REx [35] | $71.6 \pm 0.5$ |
| EQRM [15] | $71.4 \pm 0.4$ |
| SFB no adpt. | $70.6 \pm 1.8$ |
| **SFB** | $\mathbf{88.1 \pm 1.8}$ |
| Oracle no adpt. | $72.1 \pm 0.7$ |
| Oracle | $89.9 \pm 0.1$ |

### F.2 Camelyon17

We now provide results on the `Camelyon17` [3] dataset. See App. E for a description of the dataset, and App. G.5 for implementation details.

| Algorithm | Domain (Color-Label Correlation) | | | | | | | | | | |
|---|---|---|---|---|---|---|---|---|---|---|---|
| | 1.0 | 0.9 | 0.8 | 0.7 | 0.6 | 0.5 | -0.6 | -0.7 | -0.8 | -0.9 | -1.0 |
| ERM | 97.5 ± 0.3 | 88.5 ± 0.4 | 79.7 ± 0.4 | 70.6 ± 0.5 | 61.4 ± 0.7 | 52.5 ± 0.4 | 43.5 ± 0.7 | 34.7 ± 0.7 | 25.1 ± 0.5 | 16.4 ± 0.4 | 7.6 ± 0.6 |
| ERM+T3A | 98.1 ± 0.2 | 88.9 ± 0.4 | 79.8 ± 0.4 | 70.4 ± 0.5 | 61.0 ± 0.8 | 51.7 ± 0.4 | 42.3 ± 0.7 | 33.0 ± 0.6 | 23.1 ± 0.4 | 13.8 ± 0.5 | 4.5 ± 0.5 |
| ERM+PL (last) | 97.6 ± 0.2 | 88.6 ± 0.3 | 79.7 ± 0.4 | 70.6 ± 0.5 | 61.4 ± 0.8 | 52.5 ± 0.4 | 43.4 ± 0.7 | 34.6 ± 0.7 | 25.0 ± 0.5 | 16.2 ± 0.3 | 7.4 ± 0.6 |
| ERM+PL (all) | **100.0 ± 0.0** | 90.0 ± 0.4 | **80.2 ± 0.4** | 70.1 ± 0.4 | 59.9 ± 0.8 | 50.1 ± 0.4 | 40.0 ± 0.5 | 30.1 ± 0.6 | 19.6 ± 0.3 | 9.9 ± 0.3 | 0.0 ± 0.1 |
| IRM | 70.6 ± 2.1 | 70.3 ± 1.9 | 70.4 ± 1.7 | 70.2 ± 1.1 | 69.9 ± 0.7 | **69.9 ± 0.7** | 70.1 ± 0.5 | 69.7 ± 0.6 | 69.8 ± 1.2 | 69.6 ± 1.6 | 69.4 ± 1.7 |
| IRM+T3A | 72.3 ± 1.7 | 71.2 ± 1.6 | 70.7 ± 1.6 | 70.2 ± 1.0 | 69.8 ± 0.7 | **69.9 ± 0.7** | **70.3 ± 0.6** | 70.6 ± 0.5 | 71.6 ± 1.1 | 72.4 ± 1.7 | 73.4 ± 1.9 |
| IRM+PL (last) | 70.8 ± 2.2 | 70.5 ± 1.9 | 70.5 ± 1.7 | 70.2 ± 1.1 | 69.9 ± 0.7 | **69.9 ± 0.6** | 70.0 ± 0.6 | 69.7 ± 0.6 | 69.9 ± 1.2 | 69.8 ± 1.6 | 69.7 ± 1.7 |
| IRM+PL (all) | 99.6 ± 1.2 | 89.4 ± 1.2 | 79.5 ± 1.3 | 68.7 ± 3.8 | 63.3 ± 4.7 | 63.5 ± 5.3 | 63.8 ± 4.5 | 67.5 ± 2.8 | 76.4 ± 4.2 | 87.2 ± 5.0 | 98.2 ± 3.0 |
| SFB | **100.0 ± 0.1** | **90.5 ± 0.5** | 79.8 ± 0.8 | **71.0 ± 1.1** | **70.9 ± 0.3** | 69.2 ± 0.4 | 68.4 ± 1.3 | **71.2 ± 0.3** | **79.3 ± 1.2** | **88.7 ± 1.3** | **98.9 ± 1.5** |

Table 7: CMNIST comparison with other test-time/source-free unsupervised domain adaptation methods. Means and standard errors are over 10 seeds. The largest mean per column/domain is in bold. "last": only last-layer updated. "all": all layers updated. Fig. 2 gives the corresponding plot.

Table 2 shows that ERM, IRM and SFB perform similarly on Camelyon17. In line with [24], we found that a properly-tuned ERM model can be difficult to beat on real-world datasets, particularly when the model is pre-trained on ImageNet and the dataset doesn't contain severe distribution shift. While we conducted this proper tuning for ERM, IRM, and SFB (see App. G.5), doing so for ACTIR was non-trivial. We thus report the result from their paper [31, Table 1], which is likely lower due to sub-optimal hyperparameters. In particular, we found that, for ERM and IRM, using a lower learning rate (1e-5 vs 1e-4) and early stopping (1 vs 25 epochs) improved performance by 20 percentage points, from around 70% [31, Table 1] to around 90% (Table 8 below). It remains to be seen whether or not ACTIR can improve over a properly-tuned ERM model on Camelyon17.

While it may seem disappointing that SFB does not outperform the simpler methods of IRM and ERM on Camelyon17, we note that SFB can only be expected to do well when there is some gain in out-of-distribution performance from enforcing stability, e.g., when IRM outperforms ERM. The identical performances of IRM and ERM in Table 8 indicate that, with ImageNet pre-training and proper hyperparameter tuning, this is not the case for Camelyon17. Finally, despite the similar performances, we note that adapting SFB on Camelyon17 still gives a small performance boost and reduces the variance across random seeds.

Table 8: Camelyon17 test-domain accuracies. Mean and standard errors are over 5 random seeds. [†]: Result taken from [31, Tab. 1] and is likely lower due to sub-optimal hyperparameters (they report ≈70% for ERM and IRM).

| Algorithm | Accuracy |
|---|---|
| ERM | 90.2 ± 1.1 |
| IRM | 90.2 ± 1.1 |
| ACTIR | 77.7 ± 1.7[†] |
| SFB no adpt. | 89.8 ± 1.2 |
| SFB | **90.3 ± 0.7** |

# G   Implementation Details

Below we provide further implementation details for the experiments of this work. Code is available at: https://github.com/cianeastwood/sfb.

## G.1   Adaptive baselines

For both the synthetic and CMNIST datasets, we compare against adaptive baseline methods by using pseudo-labeling (PL, [36]) and test-time classifier adjustment (T3A, [30]) on top of both ERM and IRM, choosing all adaptation hyperparameters using leave-one-domain-out cross-validation:

- *ERM/IRM + PL (last):* After training with ERM/IRM, we update the last layer using the model's own pseudo-labels [36].

- *ERM/IRM + PL (all):* After training with ERM/IRM, we update all layers using the model's own pseudo-labels [36].

- *ERM/IRM + T3A:* After training with ERM/IRM, we replace the classifier (final layer) with the template-based classifier of T3A [30]. This means: (i) computing template representations for each class using pseudo-labeled test-domain data; and (ii) classifying each example based on its distance to these templates.

## G.2 Synthetic experiments

Following Jiang and Veitch [31], we use a simple three-layer network with 8 units in each hidden layer and the Adam optimizer, choosing hyperparameters using the validation domain.

For SFB, we sweep over $\lambda_S$ in $\{0.01, 0.1, 1, 5, 10, 20\}$ and $\lambda_C$ in $\{0.01, 0.1, 1\}$. For SFB's unsupervised adaptation, we employ the bias correction of Alg. 1 and calibrate the stable predictor using post-hoc temperature scaling, choosing the temperature to minimize the expected calibration error (ECE, [25]) on the validation domain. In addition, we use the Adam optimizer with an adaptation learning rate of 0.01, choosing the number of adaptation steps in $[1, 20]$ (via early stopping) using the validation domain. Finally, we report the mean and standard error over 100 random seeds.

## G.3 ColorMNIST experiments

**Training details.** We follow the setup of Eastwood et al. [15, §6.1] and build on their open-source code[5]. In particular, we use the original `MNIST` training set to create training and validation sets for each domain, and the original `MNIST` test set for the test sets of each domain. For all methods, we use a 2-hidden-layer MLP with 390 hidden units, the Adam optimizer, a learning rate of 0.0001 with cosine scheduling, and dropout with $p = 0.2$. In addition, we use full batches (size 25000), 400 steps for ERM pre-training (which directly corresponds to the delicate penalty "annealing" or warm-up periods used by penalty-based methods on `ColorMNIST` [1, 35, 15, 74]), and 600 total steps. We sweep over stability-penalty weights in $\{50, 100, 500, 1000, 5000\}$ for IRM, VREx and SFB and $\alpha$'s in $1 - \{e^{-100}, e^{-250}, e^{-500}, e^{-750}, e^{-1000}\}$ for EQRM. As the stable (shape) and unstable (color) features are conditionally independent given the label, we fix SFB's conditional-independence penalty weight $\lambda_C = 0$. As is the standard for `ColorMNIST`, we use a test-domain validation set to select the best settings (after the total number of steps), and then report the mean and standard error over 10 random seeds on a test-domain test set. As in previous works, the hyperparameter ranges of all methods are selected by peeking at test-domain performance. While far from ideal, this is quite difficult to avoid with `ColorMNIST` and highlights a core problem with hyperparameter selection in DG—as discussed by many previous works [1, 35, 24, 74, 15].

**SFB adaptation details.** For SFB's unsupervised adaptation in the test domain, we use a batch size of 2048 and employ the bias correction of Alg. 1. In addition, we calibrate the stable predictor using post-hoc temperature scaling, choosing the temperature to minimize the expected calibration error (ECE, [25]) across the two training domains. Again using the two training domains for hyperparameter selection, we sweep over adaptation learning rates in $\{0.1, 0.01\}$, choose the best adaptation step in $[5, 20]$ (via early stopping), and sweep over the number of pseudo-labeling rounds in $[1, 5]$. Finally, we report the mean and standard error over 3 random seeds for adaptation.

## G.4 PACS experiments

We follow the setup of Jiang and Veitch [31, § 6.4] and build on their open-source code[6]. This means using an ImageNet-pretrained ResNet-18, the Adam optimizer with a learning rate of $10^{-4}$, and choosing hyperparameters using leave-one-domain-out cross-validation (akin to K-fold cross-validation, except with domains). In particular, for each held-out test domain, we train 3 models—each time leaving out 1 of the 3 training domains for validation—and then select hyperparameters based on the best average performance across the held-out validation domains. Finally, we use the selected hyperparameters to retrain the model using all 3 training domains.

For SFB, we sweep over $\lambda_S$ in $\{0.01, 0.1, 1, 5, 10, 20\}$, $\lambda_C$ in $\{0.01, 0.1, 1\}$, and learning rates in $\{10^{-4}, 50^{-4}\}$. For SFB's unsupervised adaptation, we employ the multi-class bias correction of Alg. 2 and calibrate the stable predictor using post-hoc temperature scaling, choosing the temperature

---

[5]https://github.com/cianeastwood/qrm/tree/main/CMNIST
[6]https://github.com/ybjiaang/ACTIR.

to minimize the expected calibration error (ECE, [25]) across the three training domains. In addition, we use the Adam optimizer with an adaptation learning rate of 0.01, choosing the number of adaptation steps in $[1, 20]$ (via early stopping) using the training domains. Finally, we report the mean and standard error over 5 random seeds.

### G.5 Camelyon17 experiments

We follow the setup of Jiang and Veitch [31, § 6.3] and build on their open-source code[7]. This means using an ImageNet-pretrained ResNet-18, the Adam optimizer, and, following [33], choosing hyperparameters using the validation domain (hospital 4). In contrast to [31], we use a learning rate of $10^{-5}$ for all methods, rather than $10^{-4}$, and employ early stopping using the validation domain. We found this to significantly improve all methods. E.g., the baselines of ERM and IRM improve by approximately 20 percentage points, jumping from $\approx 70\%$ to $\approx 90\%$.

For SFB, we sweep over $\lambda_S$ in $\{0.01, 0.1, 1, 5, 10, 20\}$ and $\lambda_C$ in $\{0.01, 0.1, 1\}$. For SFB's unsupervised adaptation, we employ the bias correction of Alg. 1 and calibrate the stable predictor using post-hoc temperature scaling, choosing the temperature to minimize the expected calibration error (ECE, [25]) on the validation domain. In addition, we use the Adam optimizer with an adaptation learning rate of 0.01, choosing the number of adaptation steps in $[1, 20]$ (via early stopping) using the validation domain. Finally, we report the mean and standard error over 5 random seeds.

## H    Further Related Work

**Learning with noisy labels.**  An intermediate goal in our work, namely learning a model to predict $Y$ from $X_U$ using pseudo-labels based on $X_S$, is an instance of *learning with noisy labels*, a widely studied problem [56, 44, 7, 60, 38, 64]. Specifically, under the complementarity assumption ($X_S \perp\!\!\!\perp X_U | Y$), the accuracy of the pseudo-labels on each class is independent of $X_U$, placing us in the so-called *class-conditional random noise model* [56, 44, 7]. As we discuss in Section 4, our theoretical insights about the special structure of pseudo-labels complement existing results on learning under this model. Our bias-correction (Eq. (4.3)) for $P_{Y|X_U}$ is also closely related to the "method of unbiased estimators" [44]. However, rather than correcting the loss used in ERM, our post-hoc bias correction applies to any calibrated classifier. Moreover, our ultimate goal, learning a predictor of $Y$ *jointly* using $X_S$ and $X_U$, is not captured by learning with noisy labels.

**Co-training.**  Our use of stable-feature pseudo-labels to train a classifier based on a disjoint subset of (unstable) features is reminiscent of co-training [10]. Both methods benefit from conditional independence of the two feature subsets given the label to ensure that they provide complementary information.[8] The key difference is that while co-training requires (a small number of) labeled samples from the *same distribution as the test data*, our method instead uses labeled data from *a different distribution* (training domains), along with the assumption of a stable feature. Additionally, while co-training iteratively refines two pre-trained classifiers symmetrically based on each other's predictions, our method only trains the unstable classifier, in a single iteration, using the stable classifier's predictions.

**Boosting.**   Our method of building a strong (albeit unstable) classifier using a weak (but stable) one is reminiscent of boosting, in which one ensembles weak classifiers to create a single strong classifier [54] and which inspires the name of our approach, "stable feature boosting (SFB)". However, whereas traditional boosting improves weak classifiers by examining how their predictions differ from true labels, our adaptation method utilizes only pseudo-labels and needs no true labels from the test domain. For example, while traditional boosting only refines functions of existing features, SFB can utilize new features that are only available in the test domain.

**Learning theory for domain generalization.**   In addition to often assuming particular kinds of distribution shifts (e.g., covariate shift), existing error bounds for domain generalization often depend on some notion of distance between training and test domains (which does not vanish as more data is collected within domains) [9, 5, 76, 75] or assume that the test domain has a particular structural relationship with the training domains (e.g., is a convex combination of training domains [41]).

---

[7]See Footnote 6.
[8]See Krogel and Scheffer [34], Blum and Mitchell [10, Theorem 1] for discussion of this assumption.

In contrast, under the structure of invariant and complementary features, we show that consistent generalization (i.e., with generalization error vanishing as more data is collected within domains) is possible in *any* test domain. Additionally, whereas these prior works derive uniform convergence bounds (implying good generalization for ERM), our results demonstrate the benefit of an additional bias-correction step after training. We also note that, in much of this literature, "invariance" refers to invariance of the covariate marginal distribution $P_X$ across domains; in contrast, our notion of stable features (Defn. 4.1) refers to invariance of the conditional $P_{Y|X}$.

## I    Performance When Complementarity is Violated

Thm. 4.4 justifies the bias correction of Eq. (4.3) under the assumption that stable $X_S$ and unstable $X_U$ features are complementary, i.e., conditionally independent given the label $Y$. In this section, we discuss what happens if this assumption is relaxed and provide some intuition for why the bias correction appears to help even when complementarity is violated (as we observed in some of our experiments). In particular, we provide an argument that, in most cases, the bias correction should improve the accuracy of a naive classifier by making it agree more often with the Bayes-optimal classifier. While not a rigorous proof, we believe that this argument provides some insight into SFB's strong performance even when complementarity is violated.

In the absence of complementarity, the quantity $\Pr[\widehat{Y} = 1|Y = 1, X_u = x_U]$ no longer reduces to the class-wise accuracy $\Pr[\widehat{Y} = 1|Y = 1]$; thus we write more generally $\epsilon_1(x_U) = \Pr[\widehat{Y} = 1|Y = 1, X_u = x_U]$, and we write $\overline{\epsilon_1} = \mathbb{E}_{X_U}[\epsilon_1(X_U)] = \Pr[\widehat{Y} = 1|Y = 1]$ instead of simply $\epsilon_1$ for the accuracy on class 1. Similarly, we write $\epsilon_0(x_U) = \Pr[\widehat{Y} = 0|Y = 0, X_u = x_U]$, and we write $\overline{\epsilon_0} = \mathbb{E}_{X_U}[\epsilon_0(X_U)] = \Pr[\widehat{Y} = 0|Y = 0]$ instead of simply $\epsilon_0$ for the accuracy on class 0.

Let $f_*(x_U) = \Pr[Y = 1|X_U = x_U]$ denote the true regression function, and let $h_*(x_U) = 1\{f_*(x_U) > 0.5\}$ denote the Bayes-optimal classifier. It is well known that the Bayes-optimal classifier $h_*$ has the maximum possible accuracy out of all classifiers. Thus, the sub-optimality of a classifier $h$ can be measured by the probability $S(h) = \Pr_{X_U}[h(X_U) \neq h_*(X_U)]$ that it disagrees with the Bayes-optimal classifier. Our next result expresses $S(h)$ in terms of the true regression function $f_*$, the functions $\epsilon_0$ and $\epsilon_1$, and the distribution of $X_U$, when $h$ is the bias-corrected classifier

$$h_{BC}(x_U) := 1\left\{\frac{\Pr[\widehat{Y} = 1|X_U = x_U] + \overline{\epsilon}_0 - \overline{\epsilon}_1}{\overline{\epsilon}_0 + \overline{\epsilon}_1 - 1} > 0.5\right\}$$

from Thm. 4.4 or when $h$ is the "naive" classifier

$$h_{Naive}(x_U) := 1\left\{\Pr[\widehat{Y} = 1|X_U = x_U] > 0.5\right\}$$

that simply treats the pseudo-labels as true labels.

**Proposition I.1.**

$$S(h_{BC}) = \Pr_{X_U}\left[|f_*(X_U) - 0.5| \leq \frac{|\epsilon_0(X_U) - \epsilon_1(X_U) - \mathbb{E}_{X_U}[\epsilon_0(X_U) - \epsilon_1(X_U)]|}{2(\epsilon_0(X_U) + \epsilon_1(X_U) - 1)}\right],$$

*and*

$$S(h_{Naive}) = \Pr_{X_U}\left[|f_*(X_U) - 0.5| \leq \frac{|\epsilon_0(X_U) - \epsilon_1(X_U)|}{2(\epsilon_0(X_U) + \epsilon_1(X_U) - 1)}\right].$$

These two formulae for $S(h_{BC})$ and $S(h_{Naive})$ differ only in the numerator of the right-hand side; letting $Z := \epsilon_0(X_U) - \epsilon_1(X_U)$, the sub-optimality of $h_{BC}$ scales with $|Z - \mathbb{E}[Z]|$, whereas the sub-optimality of $h_{Naive}$ scales with $|Z|$. Intuitively, for all except very pathological random variables $Z$, $|Z - \mathbb{E}[Z]|$ is typically smaller than $|Z|$. Although not a rigorous proof that the bias correction is always better than the naive classifier, this analysis provides an argument that, in most cases, the bias correction should improve on the accuracy of the naive classifier, by making it agree more often with the Bayes-optimal classifier.

We conclude by sketching the proof of Proposition I.1:

*Proof.* By construction, a thresholding classifier $h(x) = 1\{f(x) > 0.5\}$ disagress with the Bayes-optimal classifier if and only if

$$f(x) \le 0.5 < f_*(x) \quad \text{or} \quad f_*(x) \le 0.5 < f(x).$$

Expanding these inequalities in the cases $f(x) = \frac{\Pr[\widehat{Y}=1|X_U=x]+\bar{\epsilon}_0-\bar{\epsilon}_1}{\bar{\epsilon}_0+\bar{\epsilon}_1-1}$ and $f(x) = \Pr[\widehat{Y} = 1|X_U = x]$ and solving for the quantity $f_*(x) - 0.5$ in each case gives Proposition I.1. $\qquad\square$

