# Supplementary Material for:
# Spuriosity Didn't Kill the Classifier: Using Invariant Predictions to Harness Spurious Features

## Notes

- For convenience, we include both the main paper and supplement here.
- The main paper is identical to the one uploaded originally.

# Spuriosity Didn't Kill the Classifier: Using Invariant Predictions to Harness Spurious Features

## Abstract

Machine learning models often fail on out-of-distribution data. To avoid this, many works have sought to extract features with a *stable* or invariant relationship with the label across domains, improving robustness by discarding the "spurious" or *unstable* features whose relationship with the label may change across domains. However, the discarded unstable features often carry *complementary* information about the label that could boost performance if used correctly in the test domain. Our main contribution is to show that it is possible to learn how to use these unstable features in the test domain *without labels*. In particular, we prove that *pseudo-labels* based on stable features provide sufficient guidance for doing so, provided that stable and unstable features are conditionally independent given the label. Along the way, we present a solution to the so-called "marginal problem" from probability theory, in the special case of conditionally-independent features, which may be of independent interest. Based on this theoretical insight, we propose Stable Feature Boosting (SFB), an algorithm for: (i) learning a predictor that separates stable and conditionally-independent unstable features; and (ii) using the stable-feature predictions to adapt the unstable-feature predictions in the test domain. Theoretically, we prove that SFB can learn an asymptotically-optimal predictor in the test domain without using any test-domain labels, while, empirically, we demonstrate the effectiveness of SFB on real and synthetic datasets.

## 1 Introduction

Machine learning systems can be sensitive to distribution shift [25]. Often, this sensitivity is due to a reliance on "spurious" features whose relationship with the label changes across domains, ultimately leading to degraded performance in the test domain of interest [20]. To avoid this pitfall, recent works on out-of-distribution (OOD) generalization have sought predictors which do not rely on these *spurious* or *unstable* relationships, but instead leverage relationships which are *invariant* or *stable* across multiple domains [43, 2, 34, 14]. However, despite their instability, spurious features can often provide additional or *complementary* information about the target label. Thus, if a predictor could be adjusted to use spurious features optimally in the test domain, it would boost performance substantially. That is, perhaps we don't need to discard spurious features at all, but rather use them in the right way.

As a very simple but illustrative example, consider the `ColorMNIST` dataset [2]. This takes the original `MNIST` dataset and first turns it into a binary classification task (digit in 0–4 or 5–9), and then colorizes it such that digit color (red or green) is a highly-informative but spurious feature. In particular, as depicted in Fig. 1, the two training domains are constructed such that green digits generally belong to class 0, while the test domain is constructed such that they generally belong to class 1. Finally, some label noise is added so that, across all 3 domains, digit shape correctly determines the label with probability 0.75. In previous works, the goal is to learn an invariant predictor which uses only shape and avoids using color—a spurious or unstable feature whose relationship with the label varies across domains. In this work, however, we ask the question: when and how can these such informative but spurious features be safely harnessed *without labels?* As shown in Fig. 1, this question is motivated by the fact that the invariant predictor is not Bayes-optimal in many test domains, since color information can be used to improve predictions in a domain-specific manner.

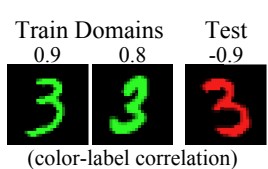
Train Domains    Test
0.9      0.8     -0.9

(color-label correlation)

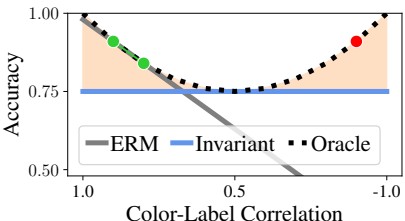

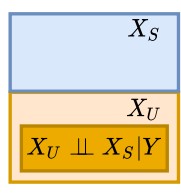

Figure 1: **Invariant (stable) and spurious (unstable) features.** (Left) Illustrative images from the `ColorMNIST` dataset. (Center) Performance across `ColorMNIST` test domains of decreasing color-label correlation for: an ERM model; an invariant model; and an oracle model using both the invariant shape *and* spurious color features optimally in the test domain. The shaded region depicts the performance boost from using the spurious feature correctly in the test domain, alongside the invariant feature. Our main contribution shows how this can be done *without labels*. (Right) Generally, invariant models use only the *stable* component $X_S$ of $X$, discarding the spurious or *unstable* component $X_U$. We prove that predictions based on $X_S$ can be used to harness a sub-component of $X_U$, highlighted in darkened orange, to reliably improve test-domain performance.

**Structure and contributions.** To answer this question, the remainder of this paper is organised as follows. We first discuss related work in § 2, providing context and high-level motivation for our proposed approach. In § 3, we then formalise the notion of stable and unstable features, showing how unstable features can be harnessed *with* test labels, and end with a number of challenges in doing so *without labels*. Next, in the theory of § 4, we provide concrete answers to these questions, before using our theoretical insights to propose a Stable Feature Boosting (SFB) algorithm with guarantees in § 5. Finally, § 6 presents our experimental results. Our main contributions can be summarised as:

- **Algorithmic:** We propose the Stable Feature Boosting (SFB) algorithm for using stable/invariant predictions to reliably harness unstable/spurious features *without test-domain labels*. To the best of our knowledge, SFB is the first method to do so.

- **Theoretical:** SFB is grounded in a novel theoretical result (Thm 4.4) giving sufficient conditions under which test-domain adaptation is provably possible without labels. Under these conditions, Thm 4.5 shows that, given enough unlabeled data, SFB learns the optimal adapted classifier.

- **Experimental:** Our experiments on synthetic and real-world data demonstrate the effectiveness of SFB, even in practical scenarios where it is unclear if its assumptions are satisfied.

## 2   Related Work

**Domain generalization, robustness and invariant prediction.** A fundamental starting point for work in domain generalization and robustness is the observation that certain "stable" features, often direct causes of the label, may have an invariant relationship with the label across domains [43, 2, 58, 49, 39, 65]. However, such stable or causal predictors often discard highly-informative but unstable information about the label. Rothenhäusler et al. [47] show that we may need to trade-off stability and predictiveness, with the causal predictor often too conservative. Eastwood et al. [14] seek such a trade-off via an interpretable probability-of-generalization parameter. The current work is motivated by the idea that one might avoid such a trade-off by changing how spurious features are used at test time, rather than discarding them are training time.

**Test-domain adaptation with labels.** Fine-tuning part of a model using a small number of labelled test-domain examples is a common way to deal with distribution shift [16, 17, 13]. More recently, it has been shown that simply retraining the last layer of an ERM-trained model outperforms more robust feature-learning methods on spurious correlation benchmarks [46, 31, 64]. In particular, Jiang and Veitch [30] do so when using a conditional-independence assumption not too dissimilar to ours. However, all of these works require labels in the test domain, while we seek to adapt *without labels*.

**Learning with noisy labels.** An intermediate goal in our work, namely learning a model to predict $Y$ from $X_U$ using pseudo-labels based on $X_S$, is an instance of *learning with noisy labels*, a widely studied problem [50, 42, 8, 51, 37, 55]. Specifically, under the complementarity assumption ($X_S \perp\!\!\!\perp X_U | Y$), the accuracy of the pseudo-labels on each class is independent of $X_U$, placing us in the so-called *class-conditional random noise model* [50, 42, 8]. As we discuss in Section 4, our theoretical insights about the special structure of pseudo-labels complement existing results on learning under this model. Our bias-correction (Eq. (4.1)) for $P_{Y|X_U}$ is also closely related to the "method of unbiased estimators" [42]. However, rather than correcting the loss used in ERM, our post-hoc bias correction applies to any calibrated classifier. Moreover, our ultimate goal, learning a predictor of $Y$ *jointly* using $X_S$ and $X_U$, is not captured by learning with noisy labels.

Table 1: **Related work.** [*]QRM [14] includes a continuous hyperparameter $\alpha \in [0,1]$ trading off between robustness and using more information from $X$.

| Method | Components of $X$ Used | | | Robust | No test-domain labels |
| | Stable | Complementary | All | | |
| --- | --- | --- | --- | --- | --- |
| ERM [56] | ✓ | ✓ | ✓ | ✗ | ✓ |
| DARE [46] | ✓ | ✓ | ✓ | ✓ | ✗ |
| IRM [2] | ✓ | ✗ | ✗ | ✓ | ✓ |
| ACTIR [30] | ✓ | ✓ | ✗ | ✓ | ✗ |
| QRM [14] | ✓ | ✓[*] | ✓[*] | ✓[*] | ✓ |
| SFB (Ours) | ✓ | ✓ | ✗ | ✓ | ✓ |

**Co-training.** Our use of stable-feature pseudo-labels to train a classifier based on a disjoint subset of (unstable) features is reminiscent of co-training [10]. Both methods benefit from conditional independence of the two feature subsets given the label to ensure that they provide complementary information.[1] The key difference is that while co-training requires (a small number of) labeled samples from the *same distribution as the test data*, our method instead uses labeled data from *a different distribution* (training domains), along with the assumption of a stable feature. Further related work is discussed in Appendix H.

# 3 Stable and Unstable Features

**Setup.** We consider the problem of domain generalization (DG) [9, 40, 23] where predictors are trained on data from multiple training domains and with the goal of performing well on data from unseen test domains. For example, in the `Camelyon17` dataset, the task is to predict if a given image of cells contains tumor tissue, and domains correspond to the different hospitals in which the images were captured ([5], see Fig. 4 of Appendix E). More formally, we consider datasets $D^e = \{(X_i^e, Y_i^e)\}_{i=1}^{n_e}$ collected from $m$ different training domains or *environments* $\mathcal{E}_{\text{tr}} := \{E_1, \ldots, E_m\}$, with each dataset $D^e$ containing data pairs $(X_i^e, Y_i^e)$ sampled i.i.d. from $\mathbb{P}(X^e, Y^e)$.[2] The goal is then to learn a predictor $f(X)$ that performs well on data from a larger set of all possible domains $\mathcal{E}_{\text{all}} \supset \mathcal{E}_{\text{tr}}$.

**Average performance: use all features.** The first approaches to DG sought predictors that perform well *on average* over domains [9, 40] using empirical risk minimization (ERM, Vapnik [57]). However, predictors that perform well on average provably lack robustness [41], potentially performing quite poorly on large subsets of $\mathcal{E}_{\text{all}}$. In particular, minimizing the average error leads predictors to make use of any features which are informative about the label (on average), including "spurious" or "shortcut" [20] features whose relationship with the label is subject to change across domains. In test domains where these feature-label relationships change in new or more severe ways than observed during training, this usually leads to significant performance drops or even complete failure [63, 6].

**Worst-case or robust performance: use only stable features.** To mitigate this lack of robustness, subsequent works have sought predictors that only use *stable or invariant* features, i.e., those which have a stable or invariant relationship with the label across domains [43, 2]. In particular, Arjovsky et al. [2] learn features which have an invariant functional relationship with the label by enforcing that the classifier on top of these features is optimal for all domains simultaneously. We henceforth use *stable features* and $X_S$ to refer to these features, and stable predictors to refer to predictors which use only these features. Analogously, we use *unstable features* $X_U$ to refer to features with an unstable or changing relationship with the label across domains. Finally, note that $X_S$ and $X_U$ form a partition of the components of $X$ *which are informative about* $Y$, as depicted in Fig. 1.

## 3.1 Harnessing unstable features *with labels*

A stable predictor $f_S(X)$ is unlikely to be the best predictor in any given domain. As depicted by the orange regions of Fig. 1, this is because it excludes unstable features $X_U$ which are informative about $Y$ and can boost performance *if used correctly*. The main question we will address in the present work is how we can harness $X_U$ to reliably boost the performance of $f_S(X)$ in a new domain $e$. To explore this question, we assume that we are indeed able to learn a stable predictor using prior methods, e.g., IRM [2], and, for now, that we have access to labelled examples in this new domain which can be used to update or re-learn the domain-specific relation between $X_U$ and $Y$.

**Boosting the stable predictor.** To begin, note that we need only update the $X_U$-$Y$ relation since, by definition, the $X_S$-$Y$ relation is stable across domains. We will thus seek a feature space which

---

[1] See Krogel and Scheffer [33] and Theorem 1 of Blum and Mitchell [10] for discussion of this assumption.
[2] We drop the domain superscript $e$ when referring to random variables from any environment.

separates $X_S$ and $X_U$, allowing only the unstable $X_U$-$Y$ relation to be updated. To do so, let us first decompose a predictor $f$ into a feature representation $\Phi$ and classifier $h$, with $f = h \circ \Phi$, and then describe the boosted joint predictor $f^e(X)$ in domain $e$ as:

$$f^e(X) = f_S(X) + f^e(X) = h_S(\Phi_S(X)) + h^e(\Phi_U(X)) \tag{3.1}$$

$$= h_S(X_S) + h^e(X_U). \tag{3.2}$$

Here, both $f_S(X)$ and $f^e(X)$ produce logits, meaning that the unstable predictor $f^e(X)$ essentially adds a domain-specific adjustment to the stable predictor $f^s(X)$ in logit space. As illustrated by Eqs. (3.1) and (3.2), the role of $\Phi_S$ and $\Phi_U$ is to extract $X_S$ and $X_U$, respectively, from the observed features $X$. Note that the stable predictor $f_S$ and classifier $h_S$, as well as the feature extractors $\Phi_S$ and $\Phi_U$ are shared across domains $e$, whereas the unstable classifier $h_U^e$ is not. In principle, $h_U^e$ could take any form, so long as we have enough labelled examples to learn it. In practice, however, we generally take $h_U^e$ to be a linear classifier for sample efficiency.

**Adapting $h_U^e$ with labels.** Given a new domain $e$ with labelled examples, we can boost the performance of our stable predictor by adapting $h_U^e$ to minimize the joint-predictor loss. Specifically, letting $\ell : \mathcal{Y} \times \mathcal{Y} \to \mathbb{R}$ be a loss function (e.g., cross-entropy) and $R_e(f) = \mathbb{E}_{(X,Y)}\left[\ell(Y, f(X))|E = e\right]$ the statistical risk of a predictor $f : \mathcal{X} \to \mathcal{Y}$ in domain $e$, we can adapt $h_U^e$ to solve:

$$\min_{h_U} \sum_{e \in \mathcal{E}_{\text{tr}}} R_e(\sigma \circ ((h_S \circ \Phi_S) + (h_U \circ \Phi_U))) \tag{3.3}$$

Note that Jiang and Veitch [30, Eq. 2.1] proposed a similar joint predictor for using *labelled* test-domain examples to update a domain-specific component. However, they do not explicitly separate stable and unstable features $X_S$ and $X_U$, which will later prove crucial for our approach *without labels*.

### 3.2 Harnessing unstable features *without labels*

The previous section made clear how we can safely harness $X_U$ when we have test-domain labels. We now consider the main question of this work—*can we safely harness $X_U$ without test-domain labels*? More specifically, how can we update the unstable classifier $h_U^e$ to capture the new $X_U$-$Y$ relation given only unlabelled test-domain examples $\{X_i^e\}_{i=1}^{n_e}$? We could, of course, simply select a *fixed* unstable classifier $h_U^e$ by relying solely on the training domains (e.g., by minimizing average error), and hope that this works for the test-domain $X_U$-$Y$ relation. However, by definition of $X_U$ being unstable, this is clearly not a robust or reliable approach—the focus of our efforts, as illustrated in Table 1. As in § 3.1, we assume that we are able to learn a stable predictor $f_S$ using prior methods [2].

**From stable predictions to robust pseudo-labels.** While we do not have labels in the test domain, we *do* have stable predictions. By definition, such predictions are imperfect (i.e., *noisy*) but robust, and can be used to form *pseudo-labels* $\hat{Y}_i = \arg\max_j f_S(X_i)_j$, with $f_S(X_i)_j$ denoting the $j^{\text{th}}$ logit of the stable prediction for $X_i$. Can we somehow use these noisy but robust pseudo-labels to guide our updating of $h_U^e$, and, ultimately, our use of $X_U$ in the test domain?

**From joint to unstable-only risk.** Unfortunately, if we try to use our robust pseudo-labels as if they were true labels—updating $h_U^e$ to minimize the joint risk as in Eq. (3.3)—we get a trivial solution of $h_U^e(\cdot) = 0$. If our loss $\ell$ is accuracy, this trivial solution is clear since $h_U^e(\cdot) = 0$ achieves 100% accuracy. For cross-entropy, the same trivial solution exists, as we show in Prop. D.1 of Appendix D. Thus, we cannot minimize a joint loss involving $f_S$'s predictions when using $f_S$'s pseudo-labels. Instead, we must consider updating $h_U^e$ to minimize the unstable-only risk $R_e(\sigma \circ h_S \circ \Phi_S)$.

**More questions than answers.** While this new procedure *could* work, it raises many questions about *when* it will work, or, more precisely, the conditions under which it can be used to safely harness $X_U$. We now summarise these questions before addressing them in the next section (§ 4):

1. **Does it make sense to minimize the unstable-only risk?** In particular, when can we minimize the unstable-predictor risk *alone* or separately, and then arrive at the optimal joint predictor? This cannot always work; e.g., for independent $X_S, X_U \sim \text{Bernoulli}(1/2)$ and $Y = X_S$ XOR $X_U$, $Y$ is independent of each of $X_S$ and $X_U$ and hence cannot be predicted from either alone.

2. **Can we just add the logits as before?** Building on question 1, if we separately optimize the predictions of the unstable classifier $h_U^e$ using the pseudo-labels $\hat{Y}$, does it make sense to simply add the logits afterwards as in Eq. (3.2)? Intuitively, simply adding the stable and unstable logits as before would require them both to be "of the same scale", or, more precisely, properly calibrated. Do we have any reason to believe that, after training on $h_S$'s pseudo-labels, $h_U^e$ will properly calibrated and thus can be integrated with $h_S$ as in Eq. (3.2)?

3. **Can the student outperform the teacher?** Stable predictions likely make mistakes—indeed, this is the motivation for trying to improve them. Is it possible to correct these mistakes with unstable features, thus improving performance? In particular, is it possible to learn an unstable "student" predictor that outperforms its own supervision signal or teacher? Perhaps surprisingly, we show that, for certain types of features, the answer is yes; in fact, even a very weak stable predictor, with performance just above chance, can be used to learn an *optimal* unstable classifier in the test domain given enough unlabeled data.

# 4  Theory: When can we safely harness unstable features without labels?

Suppose we have already identified a stable feature $X_S$ and a potentially unstable feature $X_U$ (we will return to the question of how to learn $X_S$ and $X_U$ themselves in Section 5, after identifying the additional conditions we would like $X_S$ and $X_U$ to satisfy). In this section, we analyze the problem of using $X_S$ to leverage $X_U$ without labels in the test domain. We first reduce this to a special case of the so-called "marginal problem" in probability theory, i.e., the problem of identifying a joint distribution based on information about its marginals. In the special case where two variables are conditionally independent given a third, we show this problem can be solved exactly; this solution, which may be of interest beyond the context of domain adaptation, motivates our test-domain adaptation algorithm (Algorithm 1), presented in Section 5, and forms the basis of Theorem 4.5 showing that Algorithm 1 converges to the best possible classifier given enough unlabeled data.

To formalize our assumptions, we first pose a population-level model of our domain generalization setup. Let $E$ be a random variable denoting the *environment*. Given an environment $E$, the stable feature $X_S$, the unstable feature $X_U$, and the label $Y$ are distributed according to $P_{X_S,X_U,Y|E}$. Given, this we can formalize the three key assumptions underlying our approach.

We first formalize the notion of a stable feature, motivated in the previous section:

**Definition 4.1** (Stable and Unstable Predictors). *$X_S$ is a* stable *predictor of $Y$ if $P_{Y|X_S}$ does not depend on $E$; equivalently, if $Y$ and $E$ are conditionally independent given $X_S$ ($Y \perp\!\!\!\perp E|X_S$). Conversely, $X_U$ is an* unstable *predictor of $Y$ if $P_{Y|X_U}$ depends on $E$; equivalently, if $Y$ and $E$ are conditionally dependent given $X_U$ ($Y \not\perp\!\!\!\perp E|X_U$).*

Next, we state our complementarity assumption, which we will show is key to justifying the approach of separately learning the relationships $X_S$-$Y$ and $X_U$-$Y$ and then combining them:

**Definition 4.2** (Complementary Features). *$X_S$ and $X_U$ are* complementary *predictors of $Y$ if $X_S \perp\!\!\!\perp X_U|(Y,E)$; i.e., if $X_S$ and $X_U$ contain no redundant information beyond that contained in $Y$ and $E$.*

Finally, it is fairly intuitive that, to provide a useful signal for test-domain adaptation, the stable feature needs to be predictive of the label in the test domain. Formally, we assume

**Definition 4.3** (Informative Stable Predictor). *$X_S$ is said to be* informative *of $Y$ in environment $E$ if $X \not\perp\!\!\!\perp Y|E$ (i.e., $X_S$ is predictive of $Y$ within the environment $E$).*

We will discuss the roles of these assumptions, and how they relate to the motivating questions at the end of Section 3.2, in greater detail after stating the main result (Theorem 4.4) that utilizes them. Note that, to keep our results as general as possible, we avoid assuming a particular causal generative model underlying data. However, the conditional (in)dependence assumptions above can be interpreted as constraints on such a causal model, and, in Appendix D.1, we formally characterize the set of causal generative models that are consistent with our assumptions. Notably, we show that our setting generalizes those of several existing works assuming specific causal generative models or constraints on possible distribution shifts [45, 59, 30].

**Reduction to the marginal problem with complementary features.** Since we assumed the feature $X_S$ is stable, $P_{Y|X_S,E} = P_{Y|X_S}$ in the test domain is the same as in the training domains. Hence, let us suppose we have used the training data to learn this relationship, and hence know $P_{Y|X_S}$. Suppose also that we have enough unlabeled test-domain data to learn $P_{X_S,X_U|E}$ in the test environment $E$.

Recall that our goal in test-domain adaptation is to predict $Y$ from $(X_S, X_U)$ in the test domain $E$. The remainder of our discussion will take place entirely conditioned on $E$ being the test domain, and hence we will omit this dependence from the notation. If we could express $P_{Y|X_S,X_U}$ in terms of $P_{Y|X_S}$ and $P_{X_S,X_U}$, we could then use $P_{Y|X_S,X_U}$ to optimally predict $Y$ from $(X_S, X_U)$. Thus, our task

thus becomes to reconstruct $P_{Y|X_S,X_U}$ from $P_{Y|X_S}$ and $P_{X_S,X_U}$. This is an instance of the classical "marginal problem" from probability theory [27, 28, 18], which asks under which conditions we can recover the joint distribution of a set of random variables given information about its marginals. In general, although one can place bounds on the conditional distribution $P_{Y|X_U}$, it cannot be completely inferred from $P_{Y|X_S}$ and $P_{X_S,X_U}$ [18]. However, the following section demonstrates that, *under the additional assumptions that $X_S$ and $X_U$ are complementary and $X_S$ is informative*, we can exactly recover $P_{Y|X_S,X_U}$ from $P_{Y|X_S}$ and $P_{X_S,X_U}$.

### 4.1 Solving the marginal problem with complementary features

To simplify notation, suppose the label $Y$ is binary, taking values in $\{0, 1\}$; the multiclass extension is detailed in Appendix C. The following result then shows how to reconstruct $P_{Y|X_S,X_U}$ from $P_{Y|X_S}$ and $P_{X_S,X_U}$ when $X_S$ and $X_U$ are complementary and $X_S$ is informative.

**Theorem 4.4** (Solution to the marginal problem with binary labels and complementary features)**.**
*Consider three random variables $X_1$, $X_2$, and $Y$, where (i) $Y$ is binary ($\{0,1\}$-valued), (ii) $X_1$ and $X_2$ are complementary features for $Y$ (i.e., $X_1 \perp\!\!\!\perp X_2|Y$), and (iii) $X_1$ is informative of $Y$ ($X_1 \not\perp\!\!\!\perp Y$). Then, the joint distribution of $(X_1, X_2, Y)$ can be written in terms of the joint distributions of $(X_1, Y)$ and $(X_1, X_2)$. Specifically, suppose $\hat{Y}|X_S \sim \text{Bernoulli}(\Pr[Y = 1|X_S])$ is a pseudo-label[3], and $\epsilon_0 := \Pr[\hat{Y} = 0|Y = 0]$ and are the conditional probabilities that $\hat{Y}$ and $Y$ agree, given $Y = 0$ and $Y = 1$, respectively. Then, we have $\epsilon_0 + \epsilon_1 > 1$,*

$$\Pr[Y = 1|X_2] = \frac{\Pr[\hat{Y} = 1|X_2] + \epsilon_0 - 1}{\epsilon_0 + \epsilon_1 - 1}, \qquad and \tag{4.1}$$

$$\Pr[Y = 1|X_1, X_2] = \sigma\left(\text{logit}(\Pr[Y=1|X_1]) + \text{logit}(\Pr[Y=1|X_2]) - \text{logit}(\Pr[Y=1])\right). \tag{4.2}$$

Intuitively, suppose we train a model to predict a pseudo-label $\hat{Y}$, generated based on feature $X_1$, from a feature $X_2$. Assuming $X_1$ and $X_2$ are complementary, Eq (4.1) shows how to transform this into a prediction of the true label $Y$, correcting for biases caused by possible disagreement between $\hat{Y}$ and $Y$. Meanwhile, Eq. (4.2) shows how to integrate predictors based on $X_1$ and $X_2$ while accounting for redundancy in the two predictors.

**The role of complementarity.** The assumption that $X_1$ and $X_2$ are complementary plays two separate but equally crucial roles in Theorem 4.4. First, if $X_1$ and $X_2$ only share information about $Y$, then, when we train a model to predict $\hat{Y}$ (which depends only on $X_1$) from $X_2$, the model will only learn to predict information about $Y$ (rather than other relationships between $X_1$ and $X_2$). This insight is key to justifying the bias-correction formula (Eq. (4.1)). Second, by ensuring that the only interaction between $X_1$ and $X_2$ is due to $Y$ itself, complementarity implies that $P_{Y|X_1,X_2}$ is decomposable into $P_{Y|X_1}$ and $P_{Y|X_2}$. Specifically, one can simply add estimates of $P_{Y|X_1}$ and $P_{Y|X_2}$ in logit-space while subtracting a correction term based on the marginal distribution of $Y$ (see Eq. (4.2)).

**The role of informativeness.** It is intuitive that informativeness ($X_1 \not\perp\!\!\!\perp Y$) is necessary; for the pseudo-labels to be useful, $X_1$ must help predict $Y$. More surprisingly, informativeness is *sufficient* for Theorem 4.4, i.e., *any* dependence between $X_1$ and $Y$ allows us to fully learn the relationship between $X_2$ and $Y$. This gives an affirmative answer to our question, *Can the student outperform the teacher?*, from Section 3.2. This is not to say that a strong relationship between $X_1$ and $Y$ is not helpful; while informativeness is equivalent to $\epsilon_0 + \epsilon_1 > 1$ (see Lemma A.2 in Appendix A.1), a weak relationship corresponds to $\epsilon_0 + \epsilon_1 \approx 1$, making the bias-correction 4.1 unstable. Notably, this only affects the (unlabeled) *sample complexity* of learning $P_{Y|X_2}$, not *consistency* (Theorem 4.5).

Appendix A.2 provides further discussion of Theorem 4.4, including its relationship with existing work on learning from noisy labels and possible applications beyond domain adaptation.

### 4.2 A provably consistent algorithm for unsupervised test-domain adaptation

To see why Theorem 4.4 is useful for test-domain adaptation, observe that stability of $X_1$ implies that the conditional distribution $P_{Y|X_1}$ is the same in the training and test domains. Hence, $P_{Y|X_1}$ can be learned using labeled data. Meanwhile, the joint distribution $P_{X_1,X_2}$ in the test domain can be learned using only *unlabeled* test-domain data. Theorem 4.4 thus implies that we can learn $P_{Y|X_1,X_2}$ in the test domain using only labeled data from the training domains and unlabeled data from the test domain.

---

[3]Though discrete, our *stochastic* pseudo-labels differ from hard ($\hat{Y} = 1\{\Pr[Y = 1|X_S] > 1/2\}$) or soft pseudo-labels often used in practice [19, 35, 48]. By capturing irreducible error in $Y$, stochastic pseudo-labels ensure $\Pr[Y|X_2]$ is well-calibrated, allowing us to combine $\Pr[Y|X_1]$ and $\Pr[Y|X_2]$ in Eq. (4.2).

---

**Algorithm 1:** Bias-corrected unsupervised domain adaptation procedure.

**Input:** Regression function $\eta_S(x_S) = \Pr[Y = 1 | X_S = x_S]$, subroutine `regressor`, $n$ unlabeled samples $\{(X_{S,i}, X_{U,i})\}_{i=1}^n$ from the test domain

**Output:** Estimate $\hat{\eta}_n : \mathcal{X}_S \times \mathcal{X}_U \to [0, 1]$ of $\Pr[Y = 1 | X_S = x_S, X_U = x_U]$

**1 for** $i \in [n]$ **do** // generate pseudolabels

**2** $\quad$ Sample $\hat{Y}_i \sim \text{Bernoulli}(\eta_S(X_{S,i}))$

**3** $\hat{\eta}_{U,n} \leftarrow \texttt{regressor}\big(\{(X_{U,i}, \hat{Y}_i)\}_{i=1}^n\big)$

**4** $n_1 \leftarrow \sum_{i=1}^n \hat{Y}_i$; $\hat{\beta}_{1,n} \leftarrow \text{logit}\big(\frac{n_1}{n}\big)$

**5** $(\hat{\epsilon}_{0,n}, \hat{\epsilon}_{1,n}) \leftarrow \Big(\frac{1}{n-n_1}\sum_{i=1}^n (1 - \hat{Y}_i)(1 - \eta_S(X_{S,i})), \quad \frac{1}{n_1}\sum_{i=1}^n \hat{Y}_i \eta_S(X_{S,i})\Big)$

**6 return** $(\hat{\eta}_n(x_S, x_U) \mapsto$

$\quad \sigma\Big(\text{logit}(\eta_S(x_S)) + \text{logit}\Big(\frac{\min\{\hat{\epsilon}_{1,n}, \max\{1 - \hat{\epsilon}_{0,n}, \hat{\eta}_{U,n}(x_U)\}\} + \hat{\epsilon}_{0,n} - 1}{\hat{\epsilon}_{0,n} + \hat{\epsilon}_{1,n} - 1}\Big) - \hat{\beta}_{1,n}\Big)$

---

Based on this reasoning, Alg. 1 presents our proposed unsupervised test-domain adaptation method. Intuitively, given a stable soft-classifier $\eta_S$, Algorithm 1 simply implements a finite-sample version of the bias-correction and combination equations (Eqs. (4.1) and 4.2) in Theorem 4.4. Algorithm 1 also comes with the following guarantee:

**Theorem 4.5** (Consistency Guarantee, Informal). *Assume (i) $X_S$ is stable, (ii) $X_S$ and $X_U$ are complementary, and (iii) $X_S$ is informative of $Y$ in the test domain. If $\hat{\eta}_{U,n} \to \Pr[\hat{Y} = 1 | X_U]$ as $n \to \infty$, then $\hat{\eta}_n \to \Pr[Y = 1 | X_S, X_U]$.*

In words, as the amount of unlabeled data from the test domain increases, if the regressor on Line 3 of Algorithm 1 is able to learn to predict the pseudo-label $\hat{Y}$, then the test-domain classifier output by Algorithm 1 will learn to predict the true label $Y$ in the test domain. Convergence in Theorem 4.5 occurs $P_{X_S, X_U}$-almost everywhere, both weakly (in prob.) and strongly (a.s.), depending on the mode of convergence of $\hat{\eta}_{U,n}$. Due to space constraints, formal statements and proofs are in Appendix B.

## 5 Algorithm: Stable Feature Boosting

We now use our theoretical insights from § 4 to pick up where we left off in § 3.2, ultimately arriving at a practical algorithm for harnessing unstable features without labels. We start by describing the training-domain algorithm, where our goal is to learn stable and complementary features, and then describe the test-domain adaptation algorithm, where our goal is to correctly adapt the unstable classifier $h_U^e$ using the stable predictions (or pseudo-labels).

**Recap and learning goals.** In Eq. (3.1) of § 3.1 we described a joint predictor $f^e(X) = f_S(X) + f_U^e(X)$ which can reliably boost the performance of the unstable predictor $f_S$—so long as we have labels in the test domain to update the unstable or domain-specific classifier $h_U^e$. In § 3.2, we ran into some problems when trying to update $h_U^e$ without labels, and ended the section with a number of questions about when it's possible to use the stable predictions of $f_S$ to update $h_U^e$. In § 4, we provided concrete answers to these questions, proving that informativeness ($f_S$ carries some information about $Y$) and complementarity (the stable and unstable features are conditionally independent given $Y$) suffice for learning the optimal $h_U^e$ from $f_S$'s predictions (asymptotically). Moreover, § 4 showed that, if we can indeed learn informative stable features $X_S$ and complementary features $X_C$, then we can employ the bias-corrected adaptation algorithm of Alg. 1 (or Alg. 2 for the multi-class case) to update $h_U^e$. Thus, our training-time goal is now to extract $X_S$ and $X_C$ from the observed $X$, such that we can harness $X_C$ in the test domain. More specifically, we have the following learning goals:

1. $f_S(X)$ is a stable, well-calibrated predictor with good performance.[4]

2. In a given domain $e$, $f_U^e(X)$ boosts the performance of $f_S(X)$ using complementary features.

---

[4]While Theorem 4.4 only assumes the stable feature is informative, as discussed in Section 4.1, a more accurate stable predictor improves sample efficiency of SFB.

Table 2: OOD accuracies. Mean and standard errors are over 100, 5, 5 seeds (`Synthetic`, `Camelyon17`, `PACS`).

| Algorithm | Synthetic | Camelyon17 | PACS | | | |
| | - | - | P | A | C | S |
|---|---|---|---|---|---|---|
| ERM | $9.9 \pm 0.1$ | $90.2 \pm 1.1$ | $93.0 \pm 0.7$ | $79.3 \pm 0.5$ | $74.3 \pm 0.7$ | $65.4 \pm 1.5$ |
| IRM | $74.9 \pm 0.1$ | $90.2 \pm 1.1$ | $93.3 \pm 0.3$ | $78.7 \pm 0.7$ | $75.4 \pm 1.5$ | $65.6 \pm 2.5$ |
| ACTIR | $74.8 \pm 0.4$ | $77.7 \pm 1.7^\dagger$ | $94.8 \pm 0.1$ | $\mathbf{82.5 \pm 0.4}$ | $76.6 \pm 0.6$ | $62.1 \pm 1.3$ |
| SFB w/o adapt | $74.7 \pm 1.2$ | $89.8 \pm 1.2$ | $93.7 \pm 0.6$ | $78.1 \pm 1.1$ | $73.7 \pm 0.6$ | $69.7 \pm 2.3$ |
| SFB w. adapt | $\mathbf{89.2 \pm 2.9}$ | $\mathbf{90.3 \pm 0.7}$ | $\mathbf{95.8 \pm 0.6}$ | $80.4 \pm 1.3$ | $\mathbf{76.6 \pm 0.6}$ | $\mathbf{71.8 \pm 2.0}$ |

**Objective function.** To achieve the above learning goals, we propose the following objective:

$$\min_{\Phi, h_S, h_U^e} \sum_{e \in \mathcal{E}_{\text{tr}}} R_e(\sigma \circ h_S \circ \Phi_S) + R_e(\sigma \circ ((h_S \circ \Phi_S) + (h_U^e \circ \Phi_U))) \tag{5.1}$$

$$+ \lambda_S \cdot P_{\text{Stab}}(\Phi_S, h_S) + \lambda_c \cdot P_{\text{Comp}}(\Phi_S(X), \Phi_U(X)) \tag{5.2}$$

Here, $P_{\text{Stability}}(\Phi_S, h_S)$ is a penalty encouraging stability of $\Phi_S(X)$ (i.e., $Y \perp\!\!\!\perp E | \Phi_S(X)$), while $P_{\text{Comp}}(\Phi_S(x_i), \Phi_U(x_i))$ is a penalty encouraging complementarity of $\Phi_S(X)$ and $\Phi_U(X)$ (i.e., $\Phi_S(X) \perp\!\!\!\perp \Phi_S(X) | Y$). Several approaches have been proposed for enforcing stability [43, 2, 15, 47, 58, 39, 65], e.g., IRM [2], while complementarity can be enforced by a generic conditional-dependence penalty, e.g., the conditional Hilbert-Schmidt Independence Criterion [21, HSIC] or cheaper proxy methods like that of Jiang and Veitch [30, §3.1]. $\lambda_S \in [0, \infty)$ and $\lambda_C \in [0, \infty)$ are regularization hyperparameters. In principle, an additional hyperparameter $\gamma \in [0, 1]$ could control the relative weighting of stable and joint risks, i.e., $\gamma R_e(h_S \circ \Phi_S)$ and $(1 - \gamma) R_e((h_S \circ \Phi_S) + (h_U \circ \Phi_U))$. However, in practice, we found this to be unnecessary.

**Post-hoc calibration.** Finally, as discussed in Section 4.2, correctly combining the stable and unstable predictions at adaptation time requires them to be properly calibrated. Thus, after optimizing the objective (5.2), we also suggest applying a standard post-processing step that improves the calibration of the stable classifier $h_S \circ \Phi_S$, e.g., simple temperature scaling [24].

**Adapting without labels.** Armed with a stable predictor $f_S = h_S \circ \Phi_S$ and complementary features $\Phi_U(X)$, our goal is now to adapt the unstable classifier $h_U^e$ in the test domain to safely harness (or make optimal use of) $\Phi_U(X)$. To do so, we'll make use of the bias-corrected adaptation algorithm of Alg. 1 (or Alg. 2 for the multi-class case) which takes as input the stable classifier $h_S$ and unlabelled test-domain dataset $\{\Phi_S(x_i), \Phi_U(x_i)\}_{i=1}^{n_e^T}$. This adaptation procedure returns the adapted joint classifier $\hat{f}^{e_T}$ (the logit of $\hat{\eta}_n$ in Line 6 of Alg 1) finally used for prediction in the test domain.

## 6 Experiments

We now evaluate the performance of our algorithm on synthetic and real-world datasets requiring out-of-distribution generalization. Fig. 4 depicts samples from the datasets considered, while Appendix G gives further experimental details. Code will be made available upon acceptance.

**Synthetic dataset.** We first consider an anti-causal synthetic dataset based on that of [30, §6.1] where data is generated according to the following structural equations: $Y \leftarrow \text{Rad}(0.5)$, $X_S \leftarrow Y \cdot \text{Rad}(0.75)$, and $X_U \leftarrow Y \cdot \text{Rad}(\beta^e)$, where the input $X = (X_S, X_U)$ and $\text{Rad}(\beta)$ means that a random variable is $-1$ with probability $1 - \beta$ and $+1$ with probability $\beta$. Following [30, §6.1], we create two training domains with $\beta_e \in \{0.95, 0.7\}$, one validation domain with $\beta_e = 0.6$ and one test domain with $\beta_e = 0.1$ The idea here is that, during training, prediction based on the stable $X_S$ results in lower accuracy (75%) than prediction based on the unstable $X_U$ (82.5%). Thus, models optimizing for prediction accuracy only—and not stability—will use $X_U$ and ultimately end up with only 10% in the test domain. Importantly, while the stable predictor achieves 75% accuracy in the test domain, performance can be improved to 90% if $X_U$ can be used correctly.

Following [30], we use a simple 3-layer network and choose hyperparameters using the validation-domain performance: see Appendix G for further details. As shown in Table 2, ERM performs poorly as it uses the unstable feature $X_S$, while IRM [2], ACTIR [30] and our SFB algorithm all do well by using only the stable feature $X_S$. Critically, only our SFB is capable of harnessing $X_U$ in the test domain *without labels*, leading to a near-optimal boost in performance. In Appendix F.1, we also consider a synthetic dataset where our conditional independence assumption $X_S \perp\!\!\!\perp X_U | Y$ does not hold.

**ColorMNIST.** We now consider the `ColorMNIST` dataset of Arjovsky et al. [2], described in § 1 and depicted in Fig. 1 (left). Experimentally, we follow the setup of Eastwood et al. [14, §6.1], including a simple 3-layer network: see Appendix G for further implementation details.

Table 3: CMNIST test accuracies.

| Algorithm | Test Acc. |
|---|---|
| ERM | $27.9 \pm 1.5$ |
| IRM | $69.7 \pm 0.9$ |
| V-REx | $71.6 \pm 0.5$ |
| EQRM | $71.4 \pm 0.4$ |
| SFB (Ours) w/o adapt. | $70.6 \pm 1.8$ |
| SFB (Ours) w. adapt. | $88.1 \pm 1.8$ |
| Oracle w/o adapt. | $72.1 \pm 0.7$ |
| Oracle w. adapt. | $89.9 \pm 0.1$ |

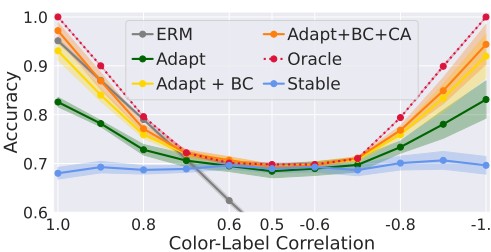

Figure 2: CMNIST results. Oracle: ERM trained on labelled test-domain data. All other curves (but ERM) refer to our algorithm. 'Stable': unadapted, 'BC': bias-corrected, and 'CA': calibrated.

Table 3 shows that: (i) SFB learns a stable predictor with performance comparable to other invariant-prediction methods like IRM [2], V-REx [34] and EQRM [14]; and (ii) only SFB is capable of harnessing the spurious color features in the test domain *without labels*, leading to a near-optimal boost in performance. Note that "Oracle w/o adapt." refers to an ERM model trained on grayscale images, while "Oracle w. adapt" refers to an ERM model trained on labelled test-domain data. In addition, Fig. 2 shows that: (i) both bias-correction (BC) and post-hoc calibration (CA) improve adaptation performance; and (ii) without labels, our SFB algorithm can harness the spurious color feature near-optimally in test domains of varying color-label correlation—the original goal we set out to achieve, depicted in Fig. 1. Further results and ablations are provided in Appendix F.2.

**PACS.** We now consider PACS [36]—a 7-class image-classification dataset consisting of 4 domains: photos (P), art (A), cartoons (C) and sketches (S), with examples shown in Fig. 4 of Appendix E. For each domain, we test model performances after training on the other three domains. Following [23, 30], we choose hyperparameters using leave-one-domain-out cross-validation.

Table 2 shows that our SFB algorithm's stable (i.e., without-adaptation) performance is comparable to that of the other invariance-seeking methods: IRM and ACTIR. One exception is the sketch domain (S), the most severe shift based on performance drop, where our stable predictor performs best. Another exception is that ACTIR's stable predictor performs better on domains A and C. Most notable, however, is: (i) the consistent boost in performance that SFB gets from unsupervised adaptation; and (ii) the fact that SFB performs best or joint-best on 3 of the 4 domains. Together, these results indicate that SFB can be useful on real-world datasets where it is unclear whether or not our conditional-independence assumption holds.

**Camelyon17.** Finally, we consider the Camelyon17 [5] dataset from the WILDS benchmark [32], a medical dataset with histopathology images from 5 hospitals which use different staining and imaging techniques (see Fig. 4 of Appendix E). The goal is to determine whether or not a given image contains tumour tissue, making it a binary classification task. We follow the train-validation-test split of WILDS, using 3 domains for training and 1 each for validation and testing. Following Jiang and Veitch [30], we use an ImageNet-pretrained ResNet18. See Appendix G.3 for further implementation details.

Table 2 shows mixed results. On the one hand, adapting gives SFB a small performance boost and reduces the variance across random seeds. On the other hand, the adapted performance is on par with both IRM and the simpler ERM method. In line with [23], we found that a properly-tuned ERM model can be difficult to beat on real-world datasets, particularly when they don't contain severe distribution shift. While we conducted this proper tuning for ERM, IRM and SFB (see Appendix G.3), doing so for ACTIR was non-trivial. We thus report the result from their paper [30, Tab. 1], which is likely lower due to hyperparameter selection (they report ≈70% accuracy for ERM and IRM).

## 7 Discussion

This work demonstrated, both theoretically and practically, how to adapt spurious but informative features to new test domains using only a stable, complementary training signal. Our proposed Stable Feature Boosting algorithm can provide significant performance gains compared to only using stable features or using unadapted spurious features, without requiring any true labels in the test domain. In theory, the most significant limitation of SFB is its assumption of complementarity (i.e., conditional independence of spurious features and stable features, given the label). Importantly, our experimental results suggest that SFB may be robust to violations of complementarity in practice; on real-world datasets such as PACS or Camelyon17, where there is no reason to believe complementarity holds, SFB performs at least as well or better than unadapted methods such as ERM and IRM.

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

# Appendices

## Table of Contents

## A    Proof and further discussion of Theorem 4.4

### A.1    Proof of Theorem 4.4

In this section, we prove our main results regarding the marginal generalization problem presented in Section 4, namely Theorem 4.4. For the reader's convenience, we restate Theorem 4.4 here:

**Theorem 4.4** (Marginal generalization with for binary labels and complementary features). *Consider three random variables $X_1$, $X_2$, and $Y$, where*

*1. $Y$ is binary ($\{0,1\}$-valued),*
*2. $X_1$ and $X_2$ are complementary features for $Y$ (i.e., $X_1 \perp\!\!\!\perp X_2 | Y$), and*
*3. $X_1$ is informative of $Y$ ($X_1 \not\perp\!\!\!\perp Y$).*

*Then, the joint distribution of $(X_1, X_2, Y)$ can be written in terms of the joint distributions of $(X_1, Y)$ and $(X_1, X_2)$. Specifically, if $\hat{Y}|X_S \sim \text{Bernoulli}(\Pr[Y = 1|X_S])$ is pseudo-label and*

$$\epsilon_0 := \Pr[\hat{Y} = 0|Y = 0] \quad \text{and} \quad \epsilon_1 := \Pr[\hat{Y} = 1|Y = 1] \tag{A.1}$$

*are the conditional probabilities that $\hat{Y}$ and $Y$ agree, given $Y = 0$ and $Y = 1$, respectively, then,*

*1. $\epsilon_0 + \epsilon_1 > 1$,*
*2. $\Pr[Y = 1|X_2] = \dfrac{\Pr[\hat{Y} = 1|X_2] + \epsilon_0 - 1}{\epsilon_0 + \epsilon_1 - 1}$, and*
*3. $\Pr[Y = 1|X_1, X_2] = \sigma\left(\text{logit}(\Pr[Y = 1|X_1]) + \text{logit}(\Pr[Y = 1|X_2]) - \text{logit}(\Pr[Y = 1])\right)$.*

Before proving Theorem 4.4, we provide some examples demonstrating that the complementarity and informativeness assumptions in Theorem 4.4 cannot be dropped.

**Example A.1.** Suppose $X_1$ and $X_2$ have independent $\text{Bernoulli}(1/2)$ distributions. Then, $X_1$ is informative of both of the binary variables $Y_1 = X_1 X_2$ and $Y_2 = X_1(1 - X_2)$ and both have identical conditional distributions given $X_1$, but $Y_1$ and $Y_2$ have different conditional distributions given $X_2$:

$$\Pr[Y_1 = 1|X_2 = 0] = 0 \neq 1/2 = \Pr[Y_2 = 1|X_2 = 0].$$

Thus, the complementarity condition cannot be omitted.

On the other hand, $X_1$ and $X_2$ are complementary for both $Y_3 = X_2$ and an independent $Y_4 \sim \text{Bernoulli}(1/2)$ and both $Y_3$ and $Y_4$ both have identical conditional distributions given $X_1$, but $Y_1$ and $Y_2$ have different conditional distributions given $X_2$:

$$\Pr[Y_3 = 1|X_2 = 1] = 1/2 \neq 1 = \Pr[Y_4 = 1|X_2 = 1].$$

Thus, the informativeness condition cannot be omitted.

Before proving Theorem 4.4, we prove Lemma A.2, which allows us to safely divide by the quantity $\epsilon_0 + \epsilon_1 - 1$ in the formula for $\Pr[Y = 1|X_2]$, under the condition that $X_1$ is informative of $Y$.

**Lemma A.2.** *In the setting of Theorem 4.4, let $\epsilon_0$ and $\epsilon_1$ be the class-wise pseudo-label accuracies defined in as in Eq. (A.1). Then, $\epsilon_0 + \epsilon_1 = 1$ if and only if $X_1$ and $Y$ are independent.*

Note that the entire result also holds, with almost identical proof, in the multi-environment setting of Sections 3 and 5, conditioned on a particular environment $E$.

*Proof.* We first prove the forwards implication. Suppose $\epsilon_0 + \epsilon_1 = 1$. If $\Pr[Y = 1] \in \{0, 1\}$, then $X_1$ and $Y$ are trivially independent, so we may assume $\Pr[Y = 1] \in (0, 1)$. Then,

$$\begin{aligned}
\mathbb{E}[\hat{Y}] &= \epsilon_1 \Pr[Y = 1] + (1 - \epsilon_0)(1 - \Pr[Y = 1]) && \text{(Law of Total Expectation)} \\
&= (\epsilon_0 + \epsilon_1 - 1)\Pr[Y = 1] + 1 - \epsilon_0 \\
&= 1 - \epsilon_0 && (\epsilon_0 + \epsilon_1 = 1) \\
&= \mathbb{E}[\hat{Y}|Y = 0]. && \text{(Definition of } \epsilon_0)
\end{aligned}$$

633 Since $Y$ is binary and $\Pr[Y = 1] \in (0, 1)$, it follows that $\mathbb{E}[\hat{Y}] = \mathbb{E}[\hat{Y}|Y = 0] = \mathbb{E}[\hat{Y}|Y = 1]$; i.e.,
634 $\mathbb{E}[\hat{Y}|Y] \perp\!\!\!\perp Y$. Since $\hat{Y}$ is binary, its distribution is specified entirely by its mean, and so $\hat{Y} \perp\!\!\!\perp Y$. It
635 follows that the covariance between $\hat{Y}$ and $Y$ is 0:

$$
\begin{aligned}
0 &= \mathbb{E}[(Y - \mathbb{E}[Y])(\hat{Y} - \mathbb{E}[\hat{Y}])] \\
&= \mathbb{E}[\mathbb{E}[(Y - \mathbb{E}[Y])(\hat{Y} - \mathbb{E}[\hat{Y}])|X_1]] && \text{(Law of Total Expectation)} \\
&= \mathbb{E}[\mathbb{E}[Y - \mathbb{E}[Y]|X_1]\,\mathbb{E}[\hat{Y} - \mathbb{E}[\hat{Y}]|X_1]] && (Y \perp\!\!\!\perp \hat{Y}|X_1) \\
&= \mathbb{E}[(\mathbb{E}[Y - \mathbb{E}[Y]|X_1])^2],
\end{aligned}
$$

636 where the final equality holds because $\hat{Y}$ and $Y$ have identical conditional distributions given $X_1$.
637 Since the $\mathcal{L}_2$ norm of a random variable is 0 if and only if the variable is 0 almost surely, it follows
638 that, $P_{X_1}$-almost surely,

$$
0 = \mathbb{E}[Y - \mathbb{E}[Y]|X_1] = \mathbb{E}[Y|X_1] - \mathbb{E}[Y],
$$

639 so that $\mathbb{E}[Y|X_1] \perp\!\!\!\perp X_1$. Since $Y$ is binary, its distribution is specified entirely by its mean, and so
640 $Y \perp\!\!\!\perp X_1$, proving the forwards implication.

641 To prove the reverse implication, suppose $X_1$ and $Y$ are independent. Then $\hat{Y}$ and $Y$ are also
642 independent. Hence,

$$
\epsilon_1 = \mathbb{E}[\hat{Y}|Y = 1] = \mathbb{E}[\hat{Y}|Y = 0] = 1 - \epsilon_0,
$$

643 so that $\epsilon_0 + \epsilon_1 = 1$. $\qquad\square$

644 We now use Lemma A.2 to prove Theorem 4.4:

645 *Proof.* To begin, note that $\hat{Y}$ has the same conditional distribution given $X_1$ as $Y$ (i.e., $P_{\hat{Y}|X_1} = P_{Y|X_1}$
646 and that $\hat{Y}$ is conditionally independent of $Y$ given $X_1$ ($\hat{Y} \perp\!\!\!\perp Y|X_1$). Then, since

$$
\Pr[\hat{Y} = 1] = \mathbb{E}[\Pr[Y = 1|X_1]] = \Pr[Y = 1], \tag{A.2}
$$

647 we have

$$
\begin{aligned}
\epsilon_1 = \Pr[\hat{Y} = 1|Y = 1] &= \frac{\Pr\left[Y = 1, \hat{Y} = 1\right]}{\Pr[Y = 1]} && \text{(Definition of } \epsilon_1) \\
&= \frac{\Pr\left[Y = 1, \hat{Y} = 1\right]}{\Pr[\hat{Y} = 1]} && \text{(Eq. (A.2))} \\
&= \frac{\mathbb{E}_{X_1}[\Pr\left[Y = 1, \hat{Y} = 1|X_1\right]]}{\mathbb{E}_{X_1}[\Pr[\hat{Y} = 1|X_1]]} && \text{(Law of Total Expectation)} \\
&= \frac{\mathbb{E}_{X_1}[\Pr[Y = 1|X_1]\Pr[\hat{Y} = 1|X_1]]}{\mathbb{E}_{X_1}[\Pr[\hat{Y} = 1|X_1]]} && (\hat{Y} \perp\!\!\!\perp Y|X_1) \\
&= \frac{\mathbb{E}_{X_1}\left[(\Pr[Y = 1|X_1])^2\right]}{\mathbb{E}_{X_1}[\Pr[Y = 1|X_1]]} && (P_{\hat{Y}|X_1} = P_{Y|X_1})
\end{aligned}
$$

648 entirely in terms of the conditional distribution $P_{Y|X_1}$ and the marginal distribution $P_{X_1}$. Similarly,
649 $\epsilon_0$ can be written as $\epsilon_0 = \frac{\mathbb{E}_{X_1}\left[(\Pr[Y=0|X_1])^2\right]}{\mathbb{E}_{X_1}[\Pr[Y=0|X_1]]}$. Meanwhile, by the law of total expectation, and
650 the assumption that $X_1$ (and hence $\hat{Y}$) is conditionally independent of $X_2$ given $Y$, the conditional
651 distribution $P_{\hat{Y}|X_2}$ of $\hat{Y}$ given $X_2$ can be written as

$$
\begin{aligned}
&\Pr[\hat{Y} = 1|X_2] \\
&= \Pr[\hat{Y} = 1|Y = 0, X_2]\Pr[Y = 0|X_2] + \Pr[\hat{Y} = 1|Y = 1, X_2]\Pr[Y = 1|X_2] \\
&= \Pr[\hat{Y} = 1|Y = 0]\Pr[Y = 0|X_2] + \Pr[\hat{Y} = 1|Y = 1]\Pr[Y = 1|X_2] \\
&= (1 - \epsilon_0)(1 - \Pr[Y = 1|X_2]) + \epsilon_1\Pr[Y = 1|X_2 = X_2] \\
&= (\epsilon_0 + \epsilon_1 - 1)\Pr[Y = 1|X_2] + 1 - \epsilon_0.
\end{aligned}
$$

By Lemma A.2, the assumption $X_1 \not\perp\!\!\!\perp Y$ implies $\epsilon_0 + \epsilon_1 \neq 1$. Hence, re-arranging the above equality gives us the conditional distribution $P_{Y|X_2}$ of $Y$ given $X_2$ purely in terms of the conditional $P_{Y|X_1}$ and $P_{X_1,X_2}$:

$$\Pr[Y = 1|X_2 = X_2] = \frac{\Pr[\hat{Y} = 1|X_2 = X_2] + \epsilon_0 - 1}{\epsilon_0 + \epsilon_1 - 1}.$$

It remains now to write the conditional distribution $P_{Y|X_1,X_2}$ in terms of the conditional distributions $P_{Y|X_1}$ and $P_{Y|X_2}$ and the marginal $P_Y$. Note that

$$
\begin{aligned}
\frac{\Pr[Y = 1|X_1, X_2]}{\Pr[Y = 0|X_1, X_2]} &= \frac{\Pr[X_1, X_2|Y = 1]\Pr[Y = 1]}{\Pr[X_1, X_2|Y = 0]\Pr[Y = 0]} && \text{(Bayes' Rule)} \\
&= \frac{\Pr[X_1|Y = 1]\Pr[X_2|Y = 1]\Pr[Y = 1]}{\Pr[X_1|Y = 0]\Pr[X_2|Y = 0]\Pr[Y = 0]} && \text{(Complementarity)} \\
&= \frac{\Pr[Y = 1|X_1]\Pr[Y = 1|X_2]\Pr[Y = 0]}{\Pr[Y = 0|X_1]\Pr[Y = 0|X_2]\Pr[Y = 1]}. && \text{(Bayes' Rule)}
\end{aligned}
$$

It follows that the logit of $\Pr[Y = 1|X_1, X_2]$ can be written as the sum of a term depending only on $X_1$, a term depending only on $X_2$, and a constant term:

$$
\begin{aligned}
\text{logit}\left(\Pr[Y = 1|X_1, X_2]\right) &= \log\frac{\Pr[Y = 1|X_1, X_2]}{1 - \Pr[Y = 1|X_1, X_2]} \\
&= \log\frac{\Pr[Y = 1|X_1, X_2]}{\Pr[Y = 0|X_1, X_2]} \\
&= \log\frac{\Pr[Y = 1|X_1]}{\Pr[Y = 0|X_1]} + \log\frac{\Pr[Y = 1|X_2]}{\Pr[Y = 0|X_2]} - \log\frac{\Pr[Y = 1]}{\Pr[Y = 0]} \\
&= \text{logit}\left(\Pr[Y = 1|X_1]\right) + \text{logit}\left(\Pr[Y = 1|X_2]\right) - \text{logit}\left(\Pr[Y = 1]\right).
\end{aligned}
$$

Since the sigmoid $\sigma$ is the inverse of logit,

$$\Pr[Y = 1|X_1, X_2] = \sigma\left(\text{logit}\left(\Pr[Y = 1|X_1]\right) + \text{logit}\left(\Pr[Y = 1|X_2]\right) - \text{logit}\left(\Pr[Y = 1]\right)\right),$$

which, by Eq. (4.1), can be written in terms of the conditional distribution $P_{Y|X_1}$ and the joint distribution $P_{X_1,X_2}$. $\qquad\square$

## A.2 Further discussion of Theorem 4.4

**Connections to learning from noisy labels.** Theorem 4.4 leverages two theoretical insights about the special structure of pseudo-labels that complement results in the literature on learning from noisy labels. First, Blanchard et al. [8] showed that learning from noisy labels is possible if and only if the total noise level is below the critical threshold $\epsilon_0 + \epsilon_1 > 1$; in the case of learning from pseudo-labels, we show (see Lemma A.2 in Appendix A.1) that this is satisfied if and only if $X_S$ is informative of $Y$ (i.e., $Y \not\perp\!\!\!\perp X_S$). Second, methods for learning under label noise commonly assume knowledge of $\epsilon_0$ and $\epsilon_1$ [42], which is unrealistic in many applications; however, for pseudo-labels sampled from a known conditional probability distribution $P_{Y|X_S}$, one can express these noise levels we show (as part of Theorem 4.4) that the class-conditional noise levels can be easily estimated.

**Possible applications of Theorem 4.4 beyond domain adaptation** The reason we wrote Theorem 4.4 in the more general setting of the marginal problem rather than in the specific context of domain adaptation is that we envision possible applications to a number of problems besides domain adaptation. For example, suppose that, after learning a calibrated machine learning model $M_1$ using a feature $X_1$, we observe an additional feature $X_2$. In the case that $X_1$ and $X_2$ are complementary, Theorem 4.4

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

$$
\begin{aligned}
\mathbb{E}[\hat{Y}|X_2] &= \sum_{y\in\mathcal{Y}} \mathbb{E}[\hat{Y}|Y=y, X_2]\Pr[Y=y|X_2] &&\text{(Law of Total Expectation)} \\
&= \sum_{y\in\mathcal{Y}} \mathbb{E}[\hat{Y}|Y=y]\Pr[Y=y|X_2] &&\text{(Complementary)} \\
&= \epsilon\,\mathbb{E}[Y|X_2]; &&\text{(Definition of } \epsilon)
\end{aligned}
$$

in particular, when $\epsilon$ is invertible,

$$\mathbb{E}[Y|X_2] = \epsilon^{-1}\,\mathbb{E}[\hat{Y}|X_2],$$

giving a multiclass equivalent of Eq. (4.1) in Theorem 4.4. We also have

$$\epsilon_{y,y'} = \Pr[\hat{Y} = y|Y = y'] = \frac{\Pr[\hat{Y} = y, Y = y']}{\Pr[Y = y']} \qquad = \frac{\mathbb{E}\left[\Pr[\hat{Y} = y, Y = y'|X_1]\right]}{\mathbb{E}\left[\Pr[Y = y'|X_1]\right]}$$

$$= \frac{\mathbb{E}\left[\Pr[\hat{Y} = y|X_1]\Pr[Y = y'|X_1]\right]}{\mathbb{E}\left[\Pr[Y = y'|X_1]\right]}$$

$$= \frac{\mathbb{E}\left[\eta_{1,y}(X_1)\eta_{1,y'}(X_1)\right]}{\mathbb{E}\left[\eta_{1,y'}(X_1)\right]},$$

suggesting the estimate

$$\hat{\epsilon}_{y,y'} = \frac{\sum_{i=1}^{n} \hat{\eta}_{S,y}(X_{S,i})\hat{\eta}_{S,y'}(X_{S,i})}{\sum_{i=1}^{n} \hat{\eta}_{S,y'}(X_{S,i})} = \sum_{i=1}^{n} \hat{\eta}_{S,y}(X_{S,i}) \frac{\hat{\eta}_{S,y'}(X_{S,i})}{\sum_{i=1}^{n} \hat{\eta}_{S,y'}(X_{S,i})}$$

of each $\epsilon_{y,y'}$, or, in matrix notation,

$$\hat{\epsilon} = \eta_S^{\mathsf{T}}(X_S)\,\mathrm{Normalize}(\eta_S(X_S)),$$

where $\mathrm{Normalize}(X)$ scales each column of $X$ to sum to 1. This gives us an multiclass equivalent of Line 4 in Alg. 1.

The multiclass versions of Eq. (4.2) and Line 6 of Alg. 1 are slightly less straightforward. Specifically, whereas, in the binary case, we used the fact that $\Pr[X_S, X_U|Y \neq 1] = \Pr[X_S, X_U|Y = 0] = \Pr[X_S|Y = 0]\Pr[X_U|Y = 0] = \Pr[X_S|Y \neq 1]\Pr[X_U|Y \neq 1]$ (by complementarity), in the multiclass case, we do not have $\Pr[X_S, X_U|Y \neq 1] = \Pr[X_S|Y \neq 1]\Pr[X_U|Y \neq 1]$. However, following similar reasoning as in the proof of Theorem 4.4, we have

$$\frac{\Pr[Y = y|X_S, X_U, E]}{\Pr[Y \neq y|X_S, X_U, E]} = \frac{\Pr[Y = y|X_S, X_U, E]}{\sum_{y' \neq y}\Pr[Y = y'|X_S, X_U, E]}$$

$$= \frac{\Pr[X_S, X_U|Y = y, E]\Pr[Y = y|E]}{\sum_{y' \neq y}\Pr[Y \neq y|X_S, X_U, E]\Pr[Y = y'|E]} \qquad \text{(Bayes' Rule)}$$

$$= \frac{\Pr[X_S|Y = y, E]\Pr[X_U|Y = y, E]\Pr[Y = y|E]}{\sum_{y' \neq y}\Pr[X_S|Y = y', E]\Pr[X_U|Y = y', E]\Pr[Y = y'|E]} \qquad (X_S \per\!\!\!\perp X_U|Y)$$

$$= \frac{\Pr[Y = y|X_S, E]\Pr[Y = y|X_U, E]}{\sum_{y' \neq y}\Pr[Y = y'|X_S, E]\Pr[Y = y'|X_U, E] \cdot \frac{\Pr[Y=y|E]}{\Pr[Y=y'|E]}}. \qquad \text{(Bayes' Rule)}$$

Hence,

$$\mathrm{logit}(\Pr[Y = y|X_S, X_U, E]) = \log\left(\frac{\Pr[Y = y|X_S, E]\Pr[Y = y|X_U, E]}{\sum_{y' \neq y}\Pr[Y = y'|X_S, E]\Pr[Y = y'|X_U, E] \cdot \frac{\Pr[Y=y|E]}{\Pr[Y=y'|E]}}\right)$$

$$= \log\left(\frac{C_y}{\sum_{y' \neq y} C_{y'}}\right) = \log\left(\frac{\frac{C_y}{\|C\|_1}}{\sum_{y' \neq y}\frac{C_{y'}}{\|C\|_1}}\right) = \mathrm{logit}\left(\frac{C_y}{\|C\|_1}\right),$$

for $C \in \mathbb{R}^{\mathcal{Y}}$ defined by

$$C_y = \frac{\eta_{S,y}(X_S)\eta_{U,y}(X_U)}{\Pr[Y = y]} \quad \text{for each } y \in \mathcal{Y}.$$

In particular, applying the sigmoid function to each side, we have

$$\Pr[Y|X_S, X_U] = \frac{C}{\|C\|_1}.$$

We can estimate $C_y$ by

$$\hat{C}_y = \frac{\eta_{S,y}(X_S)\eta_{U,y}(X_U)}{\frac{1}{n}\sum_{i=1}^{n}\eta_{S,y}(X_{S,i})}.$$

In matrix notation, this is

$$\hat{C} = \frac{\eta_S(X_S)\circ\eta_U(X_U)}{\frac{1}{n}\sum_{i=1}^{n}\eta_S(X_{S,i})},$$

where $\circ$ denotes element-wise multiplication. Putting these derivations together gives us our multiclass version of Alg. 1, presented in Alg. 2, where $\Delta^{\mathcal{Y}} = \{z \in [0,1]^K : \sum_{y\in\mathcal{Y}} z_y = 1\}$ denotes the standard probability simplex over $\mathcal{Y}$.

---

**Algorithm 2:** Multiclass bias-corrected unsupervised domain adaptation procedure.

---

**Input:** Regression function $\eta_S : \mathcal{X} \to \Delta^{\mathcal{Y}}$, subroutine `regressor`, $n$ unlabeled samples $\{(X_{S,i}, X_{U,i})\}_{i=1}^{n}$ from the test domain

**Output:** Estimate $\hat{\eta}_n : \mathcal{X}_S \times \mathcal{X}_U \to \Delta^{\mathcal{Y}}$ of regression function $\eta_y(x_S, x_U) = \Pr[Y = y | X_S = x_S, X_U = x_U]$

1 **for** $i \in [n]$ **do** // generate pseudolabels
2 $\quad$ Sample $\hat{Y}_i \sim \text{Categorical}(\eta_S(X_{S,i}))$ $\qquad$ // $\hat{Y} \in \{0,1\}^{n\times K}$ is one-hot encoded
3 $\tilde{\eta}_{U,n} \leftarrow \text{regressor}\big(\{(X_{U,i}, \hat{Y}_i)\}_{i=1}^{n}\big)$ $\qquad$ // regress pseudolabels over $X_U$
4 $\hat{\epsilon} \leftarrow \eta_S^{\intercal}(X_S)\,\text{Normalize}(\eta_S^{\intercal}(X_S))$ $\qquad$ // Estimate $\epsilon_{y,y'} = \Pr[\hat{Y} = y | Y = y]$
5 $\hat{\eta}_{U,n} \leftarrow \big(x_U \mapsto \max\{0, \min\{1, \epsilon^{-1}\tilde{\eta}_{U,n}(x_U)\}, \}\big)$ $\qquad$ // Unstable predictor
6 **for** $y \in [K]$ **do**
7 $\quad$ $C_y \leftarrow \left((x_S, x_U) \mapsto \frac{\eta_{S,y}(x_S)\circ\hat{\eta}_{U,n,y}(x_U)}{\frac{1}{n}\sum_{i=1}^{n}\eta_{S,y}(X_{S,i})}\right)$
8 $\hat{\eta}_{S,U,n} \leftarrow \left((x_S, x_U) \mapsto \frac{C(x_S,x_U)}{\|C(x_S,x_U)\|_1}\right)$ $\qquad$ // Joint predictor
9 **return** $(\hat{\eta}_{U,n}, \hat{\eta}_{S,U,n})$

---

# D Supplementary Results

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

 four datasets: `Synthetic`, `ColorMNIST`, `PACS` and `Camelyon17`. While the first two offer controlled settings with a severe spurious-correlation shift, the latter two offer real-world distribution shifts. Below, Fig. 4 depicts samples from the three image datasets.

**Dataset**         **Domains**

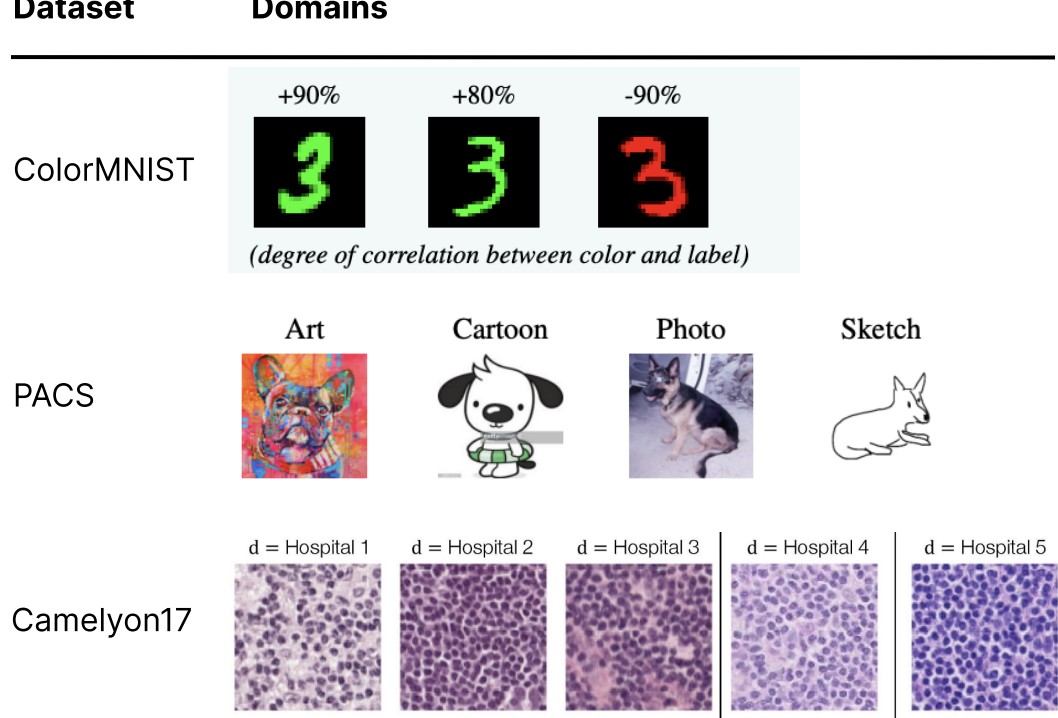

Figure 4: Examples from `ColorMNIST` [2], `PACS` [36] and `Camelyon17` [5]. Figure and examples based on Gulrajani and Lopez-Paz [23, Table 3] and Koh et al. [32, Figure 4]. For `ColorMNIST`, we follow the standard approach [2] and use the first two domains for training and the final one for testing. For `PACS` [36], we follow the standard approach [23] and use each domain in turn for testing, using the remaining three domains for training. For `Camelyon17` [5], we follow WILDS [32] and use the first three domains for training, the fourth for validation, and the fifth for testing.

## F    Further Experiments

This appendix provides further experiments which supplement those in the main text. In particular, it provides: (i) experiments on a synthetic dataset where our assumption of complementarity (i.e., conditionally-independent unstable features) does not hold (Appendix F.1); and (ii) ablations on the `ColorMNIST` dataset showing the effects of bias correction and post-hoc calibration (Appendix F.2).

### F.1    Synthetic dataset

As depicted in Fig. 1 (right), our SFB approach assumes that the harnessed unstable features $X_C \subseteq X_U$ are conditionally independent of the stable features $X_S$. If this assumption is violated, then adaptation can fail as SFB is not guaranteed to learn an asymptotically-optimal predictor in the test domain.

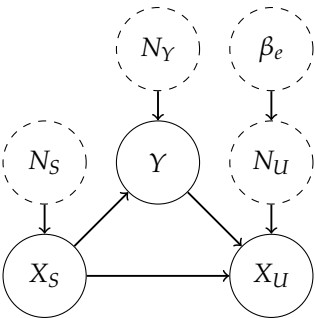

Figure 5: Causal DAG behind the synthetic dataset of Appendix F.1. Dashed circles indicate latent/unobserved variables, while solid circles indicate observed variables.

To investigate the adaptation performance of SFB when this assumption is violated, we conduct experiments on a synthetic cause-effect dataset in which there is a direct dependence between $X_S$ and $X_U$. In particular, similar to Jiang and Veitch [30, Appendix B], we generate synthetic data according to the following structural equations (illustrated graphically in Fig. 5):

$$X_S \leftarrow N_S, \text{with } N_S \leftarrow \text{Bern}(0.5);$$
$$Y \leftarrow \text{XOR}(X_S, N_Y), \text{with } N_Y \leftarrow \text{Bern}(0.75);$$
$$X_U \leftarrow \text{XOR}(\text{XOR}(Y, N_U), X_S), \text{with } N_U \leftarrow \text{Bern}(\beta_e).$$

Here, the input $X = (X_S, X_U)$ and $\text{Bern}(\beta)$ means that a random variable is 1 with probability $\beta$ and 0 with probability $1 - \beta$. Following Jiang and Veitch [30, Appendix B], we create two training domains with $\beta_e \in \{0.95, 0.8\}$, one validation domain with $\beta_e = 0.2$, and one test domain with $\beta_e = 0.1$. Like the anti-causal synthetic dataset of § 6, the idea is that prediction based on the stable $X_S$ results in lower accuracy (75%) than prediction based on the unstable $X_U$. Thus, models optimizing for prediction accuracy only—and not stability—will use $X_U$ and ultimately end up with only 10% accuracy in the test domain. In addition, while the stable predictor achieves 75% accuracy in the test domain, performance can be improved to 90% if $X_U$ can be used correctly. However, unlike the anti-causal synthetic dataset of § 6, the stable $X_S$ and unstable $X_U$ features are not conditionally independent, i.e., $X_U \not\perp X_S | Y$, since $X_S$ directly influences $X_U$. We use the same experimental setup as for the anti-causal synthetic dataset in § 6: see Appendix G.4 for further details.

Looking at Table 4 we see that: (i) ACTIR has poor stable/invariant performance as its notion of stability relies on the now-violated conditional-independence assumption; (ii) IRM has good stable/invariant performance as its notion of stability does not rely on conditional independence; (iii) SFB has good stable/invariant performance as its notion of stability does not rely on conditional independence (IRM's stability penalty is used); and (iv) surprisingly, SFB has near-optimal adapted performance despite the conditional-independence assumption being violated. One explanation for (iv) is that the conditional-independence assumption is only weakly violated in the test domain. Another is that conditional independence isn't necessary for SFB and some weaker, yet-to-be-determined condition suffices.

Table 4: Test-domain accuracies on a synthetic cause-effect dataset with a *direct* dependence between $X_S$ and $X_U$, meaning $X_U \not\perp X_S | Y$. Means and standard errors are over 100 seeds.

| Algorithm | Accuracy |
|---|---|
| ERM | $11.57 \pm 0.71$ |
| IRM | $69.61 \pm 1.26$ |
| ACTIR | $43.51 \pm 2.63$ |
| SFB w/o adapt | $74.89 \pm 3.64$ |
| SFB w. adapt | $\mathbf{88.56 \pm 1.38}$ |

 **F.2   ColorMNIST**

We now provide ablations on the `ColorMNIST` dataset to illustrate the effectiveness of the different components of SFB. In particular, we focus on bias correction and calibration, while also showing how multiple rounds of pseudo-labelling can improve performance in practice.

**Bias correction.**   To adapt the unstable classifier in the test domain, SFB employs the bias-corrected adaptation algorithm of Alg. 1 (or Alg. 2 for the multi-class case) which corrects for biases caused by possible disagreements between the stable-predictor pseudo-labels $\hat{Y}$ and the true label $Y$. In this (sub)section, we investigate the performance of SFB with and without bias correction (BC).

**Calibration.**   As discussed in § 4.2, correctly combining the stable and unstable predictions post-adaptation requires them to be properly calibrated. In particular, it requires the stable predictor $f_S$ to be calibrated with respect to the true labels $Y$ and the unstable predictor $f_U$ to be calibrated with respect to the pseudo-labels $\hat{Y}$. In this (sub)section, we investigate the performance of SFB with and without post-hoc calibration (in particular, simple temperature scaling [24]). More specifically, we investigate the effect of calibrating the stable predictor (CS) and calibrating the unstable predictor (CU).

**Multiple rounds of pseudo-labelling.**   While SFB learns the optimal unstable classifier $h_U^e$ in the test domain *given enough unlabelled data*, § 4.1 discussed how more accurate pseudo-labels $\hat{Y}$ improve the sample efficiency of SFB. In particular, in a restricted-sample setting, more accurate pseudo-labels result in an unstable classifier $h_U^e$ which better harnesses $X_U$ in the test domain. With this in mind, note that, after adapting, we expect the joint predictions of SFB to be more accurate than its stable-only predictions. This raises the question: can we use these improved predictions to form more accurate pseudo-labels, and, in turn, an unstable classifier $h_U^e$ that leads to even better performance? Furthermore, can we repeat this process, using multiple rounds of pseudo-labelling to refine our pseudo-labels and ultimately $h_U^e$? While this multi-round approach loses the asymptotic guarantees of § 4.2, we found it to work quite well in practice. In this (sub)section, we thus investigate the performance of SFB with and without multiple rounds of pseudo-labelling (PL rounds).

Table 5: SFB ablations on `ColorMNIST`. Means and standard errors are over 3 random seeds. *BC:* bias correction. *CS:* post-hoc calibration of the stable classifier. *CU:* post-hoc calibration of the unstable classifier. *PL Rounds:* Number of pseudo-labelling rounds used. *GT adapt:* adapting using true labels in the test domain.

| Model | Bias Correction | Calibration Stable | Calibration Unstable | PL Rounds | Test Acc. |
|---|---|---|---|---|---|
| SFB w/o adapt | | | | 1 | $70.6 \pm 1.8$ |
| SFB with adapt | | | | 1 | $78.0 \pm 2.9$ |
| +BC | ✓ | | | 1 | $83.4 \pm 2.8$ |
| +CS | | ✓ | | 1 | $80.6 \pm 3.4$ |
| +CU | | | ✓ | 1 | $76.6 \pm 2.4$ |
| +BC+CS+CU | ✓ | ✓ | ✓ | 1 | $84.4 \pm 2.2$ |
| +BC+CS | ✓ | ✓ | | 1 | $84.9 \pm 2.6$ |
| +BC+CS | ✓ | ✓ | | 2 | $87.4 \pm 1.9$ |
| +BC+CS | ✓ | ✓ | | 3 | $88.1 \pm 1.8$ |
| +BC+CS | ✓ | ✓ | | 4 | $88.6 \pm 1.3$ |
| +BC+CS | ✓ | ✓ | | 5 | $88.7 \pm 1.3$ |
| SFB with GT adapt | ✓ | ✓ | | 1 | $89.0 \pm 0.3$ |

**Results.**   Table 5 reports the ablations of SFB on `ColorMNIST`. Here we see that: (i) bias correction significantly boosts performance (+BC); (ii) calibrating the stable predictor also boosts performance without (+CS) and with (+BC+CS) bias correction, with the latter leading to the best performance; (iii) calibrating the unstable predictor (with respect to the pseudo-labels) slightly hurts performance without (+CU) and with (+BC+CS+CU) bias correction and stable-predictor calibration; (iv) multiple rounds of pseudo-labelling boosts performance, while also reducing the performance variation across random seeds; (v) using bias correction, stable-predictor calibration and 5 rounds of pseudo-labelling

results in near-optimal adaptation performance, as indicated by the similar performance of SFB when using true labels $Y$ to adapt $h_U^e$ (denoted "SFB with GT adapt" in Table 5).

# G Implementation Details

Below we provide further implementation details for each of the experiments/datasets considered in this work. Code for reproducing all experimental results will be made available upon acceptance.

## G.1 ColorMNIST

**Training details.** We follow the setup of Eastwood et al. [14, §6.1] and build on their open-source code[5]. In particular, we use the original `MNIST` training set to create training and validation sets for each domain, and the original `MNIST` test set for the test sets of each domain. For all methods, we use a 2-hidden-layer MLP with 390 hidden units, the Adam optimizer, a learning rate of 0.0001 with cosine scheduling, and dropout with $p = 0.2$. In addition, we use full batches (size 25000), 400 steps for ERM pertaining (which directly corresponds to the delicate penalty "annealing" or warm-up periods used by penalty-based methods on `ColorMNIST` [2, 34, 14]), and 600 total steps. We sweep over stability-penalty weights in $\{50, 100, 500, 1000, 5000\}$ for IRM, VREx and SFB and $\alpha$'s in $1 - \{e^{-100}, e^{-250}, e^{-500}, e^{-750}, e^{-1000}\}$ for EQRM. As the stable (shape) and unstable (color) features are conditionally independent given the label, we fix SFB's conditional-independence penalty weight $\lambda_C = 0$. As is the standard for `ColorMNIST`, we use a test-domain validation set to select the best settings (after the total number of steps), and then report the mean and standard error over 10 random seeds on a test-domain test set. As in previous works, the hyperparameter ranges of all methods are selected by peeking at test-domain performance. While far from ideal, this is quite difficult to avoid with `ColorMNIST` and highlights a core problem with hyperparameter selection in DG—as discussed by many previous works [2, 34, 23, 64, 14].

**Adaptation details.** For SFB's unsupervised adaptation in the test domain, we use a batch size of 2048 and employ the bias correction of Alg. 1. In addition, we calibrate the stable predictor using post-hoc temperature scaling, choosing the temperature to minimize the expected calibration error (ECE, [24]) across the two training domains. Again using the two training domains for hyperparameter selection, we sweep over adaptation learning rates in $\{0.1, 0.01\}$, choose the best adaptation step in $[5, 20]$ (via early stopping), and sweep over the number of pseudo-labelling rounds in $[1, 5]$. Finally, we report the mean and standard error over 3 random seeds for adaptation.

## G.2 PACS

We follow the experimental setup of Jiang and Veitch [30, Section 6.4] and build on their open-source implementation[6]. This means using an ImageNet-pretrained ResNet-18, the Adam optimizer with a learning rate of $10^{-4}$, and, following [23], choosing hyperparameters using leave-one-domain-out cross-validation. This is akin to K-fold cross-validation except with domains, meaning that we train 3 models—each time leaving out 1 of the 3 training domains for validation—and then select hyperparameters based on the best average performance across the held-out validation domains. Finally, we use the selected hyperparameters to retrain the model using all 3 training domains.

For SFB, we sweep over $\lambda_S$ in $\{0.01, 0.1, 1, 5, 10, 20\}$, $\lambda_C$ in $\{0.01, 0.1, 1\}$, and learning rates in $\{10^{-4}, 50^{-4}\}$. For SFB's unsupervised adaptation, we employ the multi-class bias correction of Alg. 2 and calibrate the stable predictor using post-hoc temperature scaling, choosing the temperature to minimize the expected calibration error (ECE, [24]) across the three training domains. In addition, we use the Adam optimizer with an adaptation learning rate of 0.01, choosing the number of adaptation steps in $[1, 20]$ (via early stopping) using the training domains. Finally, we report the mean and standard error over 3 random seeds.

## G.3 Camelyon17

We follow the experimental setup of Jiang and Veitch [30, Section 6.3] and build on their open-source implementation[7]. This means using an ImageNet-pretrained ResNet-18, the Adam optimizer, and, following [32], choosing hyperparameters using the validation domain (hospital 4). In contrast to

---

[5]https://github.com/cianeastwood/qrm/tree/main/CMNIST
[6]https://github.com/ybjiaang/ACTIR.
[7]See Footenote 6.

[30], we use a learning rate of $10^{-5}$ for all methods, rather than $10^{-4}$, and employ early stopping using the validation domain. We found this to significantly improve all methods. E.g., the baselines of ERM and IRM improve by approximately 20 percentage points, jumping from $\approx 70\%$ to $\approx 90\%$.

For SFB, we sweep over $\lambda_S$ in $\{0.01, 0.1, 1, 5, 10, 20\}$ and $\lambda_C$ in $\{0.01, 0.1, 1\}$. For SFB's unsupervised adaptation, we employ the bias correction of Alg. 1 and calibrate the stable predictor using post-hoc temperature scaling, choosing the temperature to minimize the expected calibration error (ECE, [24]) on the validation domain. In addition, we use the Adam optimizer with an adaptation learning rate of 0.01, choosing the number of adaptation steps in $[1, 20]$ (via early stopping) using the validation domain. Finally, we report the mean and standard error over 3 random seeds.

### G.4 Synthetic

Following Jiang and Veitch [30], we use a simple three-layer network with 8 units in each hidden layer and the Adam optimizer, choosing hyperparameters using the validation domain.

For SFB, we sweep over $\lambda_S$ in $\{0.01, 0.1, 1, 5, 10, 20\}$ and $\lambda_C$ in $\{0.01, 0.1, 1\}$. For SFB's unsupervised adaptation, we employ the bias correction of Alg. 1 and calibrate the stable predictor using post-hoc temperature scaling, choosing the temperature to minimize the expected calibration error (ECE, [24]) on the validation domain. In addition, we use the Adam optimizer with an adaptation learning rate of 0.01, choosing the number of adaptation steps in $[1, 20]$ (via early stopping) using the validation domain. Finally, we report the mean and standard error over 100 random seeds.

## H    Further Related Work

**Using spurious or unstable features without labels.**    Bui et al. [12] exploit-domain specific or unstable features with a meta-learning approach. However, they use the unstable features *in the same way* in the test domain, which, by their very definition, can lead to degraded performance. In contrast, we seek a *robust* approach to safely harness the unstable features in the test domain, as summarised in Table 1. Sun et al. [54] share the goal of exploiting spurious or unstable features to go "beyond invariance". However, their approach requires labels for the spurious features at training time and only applies to label shifts. In contrast, we do not require labels for the spurious features and are not restricted to label shifts.

**Self-learning via pseudo-labelling.**    In the source-free and test-time domain adaptation literature, adapting to the test domain using a model's own pseudo-labels is a common approach [35, 38, 61, 29]— see Rusak et al. [48] for a recent review. In contrast to these approaches, we use one model to provide the pseudo-labels (the stable model) and the other to use/adapt to the pseudo-labels (the unstable model). In addition, while the majority of this pseudo-labelling work is purely empirical, we provide theoretical justification and guarantees for our SFB approach.

## I    Limitations

In our view, the most significant limitation of this work is the assumption of complementarity (i.e., that the spurious features are conditionally independent of the stable features, given the label). Complementarity is implicit in the causal generative models assumed by existing related work [45, 60, 30], and, as Example A.1 in Appendix A.1 demonstrates, is cannot simply be dropped from our theoretical motivation. In the related context of co-training, this condition was initially assumed and then weakened in subsequent work [10, 4, 1, 62]; similarly, we hope future work will identify weaker conditions that are sufficient for SFB to succeed. On the other hand, our experimental results on the synthetic dataset of Appendix F.1, as well as the real datasets of `PACS` and `Camelyon17`, suggest that SFB may be robust to violations of complementarity—perhaps mirroring the surprisingly good practical performance of methods such as naive Bayes classification which are justified under similar assumptions [44].