# OpenReview forum: "Spuriosity Didn’t Kill the Classifier: Using Invariant Predictions to Harness Spurious Features"
_NeurIPS.cc/2023/Conference — NeurIPS 2023 poster_

### Official Review · Reviewer_5BRF · 2023-06-30

**Soundness:** 4 excellent
**Presentation:** 3 good
**Contribution:** 3 good
**Rating:** 7
**Confidence:** 3

**Summary:**

The authors present a method for training models that are robust to out-of-distribution data by focusing on stable features. Stable features are those whose conditional relationship with the label do not change across domains. The benefit of their approach is demonstrated 1 synthetic dataset and 3 "real-world" datasets.


**Strengths:**

Spurious correlations is a significant problem in many applications, which erodes the power of predictive models and shortens their usable life. The method proposed by the authors helps to mitigate these issues.

The paper is well written and easy to follow. The proposed algorithm appears easy to use and computationally scalable.


**Weaknesses:**

This work is closely related to Puli et al. (2021). I think the authors should consider comparing against NURD as well.

Puli, Aahlad, et al. "Out-of-distribution generalization in the presence of nuisance-induced spurious correlations." arXiv preprint arXiv:2107.00520 (2021).

Does line 243 have a typo? What is \epsilon_1?

Algorithm 1 (listing) could use some text describing each step.

In Table 3, please bold the best non-oracle accuracy.


**Questions:**

How can one know whether two feature sets are complementary? Are there any tests for that?

What if the stable features cannot be fully isolated? For example, in Puli et al. the authors need only grab the border of images to define the spurious variables but that leaves the center of the image as a mixture of spurious and non-spurious content.


**Limitations:**

The authors have adequately addressed limitations.

---

> ### Author Rebuttal · Authors · 2023-08-09
>
> We sincerely thank the reviewer for their comments and suggestions.
>
> > “This work is closely related to Puli et al. (2021). I think the authors should consider comparing against NURD as well.”
>
> Thank you for pointing out the relevant paper of Puli et al. (2021), which we will cite in the revision. While both their method (NuRD) and ours (SFB) seek to improve OOD generalization, the two approaches are complementary. Specifically, the goal of NuRD is to _remove_ domain-specific information from the predictor, _thereby encouraging stability_. Meanwhile, _SFB builds on top of stability by using it to leverage complementary, domain-specific information_. For example, the representation learned by NuRD could serve as the stable representation/feature $X_S$ used by SFB.
>
> As a concrete example of how these approaches differ, in our CMNIST experiment (Table 3 of the submission), NuRD (treating color as the nuisance variable) might achieve the 70% test performance of IRM, but NuRD cannot achieve the 90% test performance of SFB, because NuRD _discards_ the unstable but highly informative color feature, whereas SFB _leverages_ color.
>
> > “Does line 243 have a typo? What is $\epsilon_1$?”
>
> Yes, this was a typo; Line 243 should read, “$\epsilon_0 := \Pr[\hat Y = 0|Y = 0]$ and $\epsilon_1 := \Pr[\hat Y = 1|Y = 1]$”. We have fixed this.
>
> > “Algorithm 1 (listing) could use some text describing each step.”
>
> We agree and have included a clearer version of Algorithm 1 in the general-response PDF above.
>
> > “In Table 3, please bold the best non-oracle accuracy.”
>
> Good point, we have now done so.
>
> > “How can one know whether two feature sets are complementary? Are there any tests for that?”
>
> If $Y$ takes $K$ values $\{1,...,K\}$, then complementarity is equivalent to $X_S \perp X_U | Y = k$ for all $k \in \{1,...,K\}$. Thus, testing complementarity reduces to performing $K$ tests of independence between $X_S$ and $X_U$ (once using the data from each class). There are many statistical tests for independence, such as $\chi^2$ tests, kernel- or nearest-neighbor-based tests, etc., which make different assumptions about the distributions of $X_S$ and $X_U$ (discrete vs. continuous, parametric vs. nonparametric, etc.).
>
> > “What if the stable features cannot be fully isolated? For example, in Puli et al. the authors need only grab the border of images to define the spurious variables but that leaves the center of the image as a mixture of spurious and non-spurious content.”
>
> Indeed, as we discuss in the general response, isolating stable features is a challenging and active area of research [1--6]. While the theory supporting SFB assumes $X_S$ is a stable feature, it is agnostic to how $X_S$ is actually learned/extracted/isolated. For example, $X_S$ could be the “uncorrelating representation” learned by NuRD [6], or that learned by IRM [1], VREx [2], CLOvE [3], QRM [4], Fish [5], etc. To illustrate this point, we have added ablations on CMNIST where SFB uses different stability penalties, namely IRM, VREx, and CLOvE (see the general response PDF).

---

> > ### Comment · Reviewer_5BRF · 2023-08-11
> > **I have read the rebuttal**
> >
> > I've read the authors' rebuttal. I am content to let my score stand (at this time).

---

### Official Review · Reviewer_f1Zc · 2023-07-01

**Soundness:** 3 good
**Presentation:** 2 fair
**Contribution:** 3 good
**Rating:** 6
**Confidence:** 4

**Summary:**

This work considers unsupervised domain adaptation, where a learner observes labeled data from multiple source domains and unlabeled data from a target domain, and needs to produce a classifier that works well for data from the target domain. In contrast to the domain generalization setting where no data is available from the target domain, here, the learner can be less conservative and use features that are spurious (unstable or domain-specific) but help improve performance on the target domain. The main challenge is to do this with only unlabeled examples from the target domain.

The main result of this work (Theorem 4.4) specifies a sufficient condition under which it is possible to use domain-specific features to boost the performance of a conservative classifier that relies on stable features only. Given stable features $X_S$ (i.e., $Y | X_S$ is invariant across domains) and unstable features $X_U$, such that $X_U  \perp X_S \mid Y$ (complementarity) and $X_S$ is informative about $Y$, they express $p_\text{target}(Y | X_S, X_U)$ in terms of $p_\text{target}(X_S, X_U)$ and $p(Y | X_S)$, both of which can be estimated in the unsupervised domain adaptation setting. Essentially, it reduces to predicting pseudo-labels $\widehat{Y} \sim p(Y|X_S)$ from the unstable features $X_U$ and combining it with $p(Y|X_S)$ to construct a classifier that uses both $X_S$ and $X_U$.

Besides this the authors propose a representation learning method that separates stable and unstable features, such that they also satisfy the complementarity assumption. They evaluate their method on a few synthetic datasets, CMNIST, PACS, and Camelyon-17. The results indicate improvement compared to domain generalization methods (i.e., without an adaptation step).

**Strengths:**

**Strength #1: Significance.**  Unsupervised domain adaption is an important and relevant setting for the NeurIPS community. The main result (Theorem 4.4) is a solid contribution to the list of conditions under which some form of out-of-domain generalization (with or with adaptation) becomes possible. I like how the problem of label shift that is a challenge for many methods, is handled naturally in the proposed approach.

**Strength #2: Originality.** As far as I know, the main result (Theorem 4.4) is novel. The fact that in population-level analysis even slight informativeness of $X_S$ for $Y$ allows recovering the true conditional distribution $P(Y|X_S,X_U)$ is not trivial.

**Weaknesses:**

**Weakness #1: Clarity & Presentation.** The clarity of the presentation can be improved. The notation is not consistent in many cases. Many important discussions/details are presented in appendices, while the main text space is not used effectively. Please see more detailed comments below.

**Weakness #2: Limitations.**
1. In Theorem 4.4, it is assumed that we have access to stable $X_S$ and unstable features $X_U$. This assumption is quite strong, as one can argue that finding stable and unstable features is one of the key steps in out-of-domain generalization. To my best knowledge there are no methods that extract and separate stable and unstable features well.
2. In Theorem 4.4, it is also assumed that we have access to the Bayes probabilities $P(Y|X_1)$, where $X_1$ are the stable features. In practice this is not easy to achieve.
3. The proposed algorithm to extract stable and unstable complementary features has many hyper-parameters. It is unclear to me how sensitive is the method to these hyper-parameters and how easy it is to tune them.
4. The proposed method requires enforcing stability and complementarity, which is challenging. Not only it is hard to enforce these conditions on source domains (e.g., limitations of conditional HSIC), but the same OOD generalization concern arises -- do these conditions enforced on source domains carry over to the target domain? For this reason, the work will benefit significantly from evaluations targeted to reveal how well the proposed representation learning approach achieves stability and complementarity (rather than evaluating only target domain performance after adaptation).

**Questions:**

- Lines 112 - 116: It should specified what is meant by *relationship* between a feature $X$ and the label $Y$ (i.e., the conditional distribution of $Y|X$). I understand that stable and unstable features are defined later, but at this point of the text reader does not know it and can only guess.
- Line 115: I think it should be more exact what $X_S$ and $X_U$ are. As I understand from eq. (3.1) they are not subsets of the components of $X$, but rather extracted features: i.e., $X_S = \Phi_S(X)$ and $X_U = \Phi_U(X)$.
- Lines 126-127: This sentence is not rigorous enough.
- Eq (3.1): It should be $f^e_U(X)$. Also, in (3.1) and (3.2) it should be $h^e_U$ instead of $h^e$.
- Line 131: $f^s(X)$ -> $f_S(X)$.
- Lines 137-143: A separate letter should be used to denote the new environment (using $e$ for everything is confusing). Also, the usage of $h_U$ or $h_U^e$ is inconsistent. Why is there a sum over training environments in eq. (3.3)? Shouldn't it be just optimizing for the domain-specific classifier on the target data given already trained $\Phi_S, \Phi_U,$ and $h_S$?
- Line 163: It should be $\sigma \circ h_U \circ \Phi_U$.
- Definition 4.3. It should be $X_S \perp Y \mid E$.
- Theorem 4.4: It is worth to mention in the statement that $\epsilon_0$ and $\epsilon_1$ can be expressed in terms of $P(Y|X_1)$ and $P(X_1)$.
- Lines 295-305 (recap and learning goals): I suggest to remove this part. Instead you can bring some parts from the appendix to the main text (e.g., Appendices A.2, H or I).
- Line 309, objective function: Do I understand correctly that during this first phase you have one domain-specific classifier $h^e_U$ per domain? If so, is the point to make sure that you learn meaningful $\Phi_U$, rather than a constant function? In that case one can argue that if there is a new kind of spurious features available for the target domain, that might not be picked in this first phase of training.
- Do the results in the main text use multiple rounds of pseudo-labeling?
- Are there no suitable unsupervised domain adaptation methods to compare the proposed approach with?

**Limitations:**

The authors listed assumption of complementarity in Theorem 4.4 as the main limitation of their work. To me this assumption itself is not a big limitation, as it is satisfied in many cases. The main limitation is finding a representation of data that separates stable and unstable features (see my comments on limitations in the "weaknesses" section).

---

> ### Author Rebuttal · Authors · 2023-08-09
>
> We sincerely thank the reviewer for their detailed review, comments and suggestions.
>
> > “Weakness #1: Clarity & Presentation ...”
>
> We take the reviewer’s point here and acknowledge that there were some notational inconsistencies and that some important material was relegated to the appendix for space reasons. In response, we have:
> - Corrected the typos pointed out by the reviewer.
> - Lines 112 - 116: Added a forward pointer to the formal definitions in Section 4 and been more exact in specifying that $X_S = \Phi_S(X)$ and $X_U = \Phi_U(X)$ are extracted features.
> - Lines 126-127: Added a more precise explanation of $\Phi$ and $h$.
> - Lines 137-143: Used $e^t$ for the new test environment and ensured we are consistent in using $h^e_U$. The reviewer is correct that there should not be a sum over training environments in Eq. (3.3) – this has been removed.
> - Theorem 4.4: Noted that $\epsilon_0$ and $\epsilon_1$ can be expressed in terms of $P(Y|X_1)$ and $P(X_1)$.
> - Lines 295-305 (recap and learning goals): Removed.
> - Limitations: Revised the limitations to include the learning/extraction of $X_S$ and $X_U$ as we agree that this is one of the major challenges. We discuss this point further below.
> - Algorithm: Rewritten Algorithm 1 -- please see the general-response PDF.
>
>
> > “Weakness #2: Limitations”
>
> We now respond to each limitation. In short, we agree with the highlighted limitations and will be sure to discuss them properly in the paper. However, as detailed below, many of these limitations/challenges apply much more widely than just to our method, such as learning a stable predictor, a well-calibrated predictor, or transferring knowledge/functions from source to target domains.
>
> 1. _"In Theorem 4.4, it is assumed that we have access to stable $X_S$ and unstable $X_U$ features. This assumption is quite strong… To my best knowledge there are no methods that extract and separate stable and unstable features well."_
>
> We agree that extracting stable (and unstable) features is one of the key steps in OOD generalization, and very much an open problem (and standalone topic in its own right). We will highlight this open challenge, and note that SFB could benefit from future improvements in invariant/stable-feature learning.
>
> 2. _“In Theorem 4.4, it is also assumed that we have access to the Bayes probabilities $P(Y|X_1)$, where $X_1$ are the stable features. In practice this is not easy to achieve.”_
>
> We agree that, in practice, it is not easy to access Bayes probabilities (i.e., learn a perfectly-calibrated stable predictor). We will highlight this practical difficulty and note that (i) learning well-calibrated predictors is also a topic of active research [22--25] in its own right; and (ii) future improvements in this direction would benefit SFB.
>
> 3. _“It is unclear to me how sensitive is the method to hyper-parameters and how easy it is to tune them.”_
>
> We will conduct a sensitivity analysis for the next/final version of the paper. In particular, we will use a 2D contour/heat map to show SFB’s performance as a function of $\lambda_{\text{Stability}}$ and $\lambda_{\text{Compl.}}$ on synthetic datasets.
>
> 4. _“The proposed method requires enforcing stability and complementarity, which is challenging… do these conditions enforced on source domains carry over to the target domain?”_
>
> We agree that, like prior works, the goal is for conditions enforced on source domains to carry over to target domains. As with any kind of OOD generalization, carrying properties like stability or complementarity from source to target domains depends on the specific assumptions made about the domains: e.g., IRM [1] assumes that the causal mechanism of $Y$ remains fixed, while QRM [4] assumes IID domains. However, we do not believe that there is a better way to assess this transfer than evaluating target-domain performance (pre and post adaptation). For example, on CMNIST and the synthetic dataset, the near-oracle pre- and post-adaptation performances tell us that (i) stability was indeed transferred to the target domain; and (ii) the unstable features extracted from source domains were successfully transferred to (and exploited in) the target domain. Alternatives include:
> - a. **Conducting a sensitivity analysis**: to see how target-domain performance changes as a function of $\lambda_{\text{Stability}}, \lambda_{\text{Compl.}}$, i.e., as a function of better-enforced conditions (on the training data). As mentioned above, we will include this sensitivity analysis in the next/final version of the paper.
> - b. **Reporting the magnitude of the conditions/penalties on source and target domains:** This is generally not done in DG (e.g., for [1–3]) as these values are not interpretable/informative.
> - c. **Something else:** Perhaps the reviewer had some other evaluation in mind?
>
> > “Line 309, objective function: Do I understand correctly that during this first phase you have one domain-specific classifier $h^e_U$ per domain? If so, is the point to make sure that you learn meaningful $\Phi_U$, rather than a constant function? In that case one can argue that if there is a new kind of spurious features available for the target domain, that might not be picked in this first phase of training.”
>
> Yes, this is correct. Our method cannot _extract_ a new spurious feature from $X$ at test time, if it was not already included in $\Phi_U(X)$. However, if a new feature is given to us _directly_ at test time, e.g. patient test results or meta-data to complement imaging results, we can indeed exploit this new feature without labels (so long as it satisfies complementarity). We have added this to our discussion.
>
> > “Do the results in the main text use multiple rounds of pseudo-labeling?”
>
> Only for CMNIST. We will clarify this in the main text.
>
> > “Are there no suitable unsupervised domain adaptation methods to compare the proposed approach with?”
>
> Please see the general response and the first part of our response to Reviewer S7EC.

---

> > ### Comment · Reviewer_f1Zc · 2023-08-16
> > **Reviewer response**
> >
> > Thank you for the detailed response.
> >
> > > b. Reporting the magnitude of the conditions/penalties on source and target domains: This is generally not done in DG (e.g., for [1–3]) as these values are not interpretable/informative.
> > > c. Something else: Perhaps the reviewer had some other evaluation in mind?
> >
> > Given a reliable way of estimating $p(y | \Phi_S)$, one can estimate any distance or divergence between $p(y | \Phi_S)$ of all available domains (source and target) and test whether the average source-source distance is smaller than the average source-target distance.

---

> > > ### Author Response · Authors · 2023-08-22
> > >
> > > We thank the reviewer for their clarification/suggestion on evaluating stability.
> > >
> > > In theory, if we had a large number of training domains to obtain a null distribution for this average distance/divergence, this approach seems quite sensible. In practice, however: (i) real-world DG/DA datasets generally don't have a large number of training domains (e.g., PACS and Camelyon17 each have 3 training domains); and (ii) synthetic datasets are generally designed such that OOD performance already evaluates stability (e.g., on CMNIST ~70% accuracy indicates a stable, shape-based predictor).

---

### Official Review · Reviewer_NDD6 · 2023-07-06

**Soundness:** 4 excellent
**Presentation:** 4 excellent
**Contribution:** 3 good
**Rating:** 7
**Confidence:** 3

**Summary:**

Unlike the common trend in the out-of-distribution generalization literature in the presence of spurious correlation, this paper takes a different approach. They argue that the spurious feature could be utilized to enhance the performance of the classifier. Then they go on to show how unlabelled data in the test domain can be used to train a classifier that adaptively uses the spurious/unstable feature (along with the stable features) to get near optimal prediction performance.

**Strengths:**

**Clarity**: This paper is very well-written and easy to follow with appropriate connections between theory and experiments.

**Significance**:
1. The majority of current literature in the OOD generalization in the presence of the spurious correlation is focused on learning a predictor that only uses the stable feature and thus generalizes across different environments. In doing so they throw away (possibly) easy-to-learn and highly predictive features for the given task since their correlation is likely to change across domains.
2. This work is an important step in the direction of adaptively making use of the spurious feature without any need for extra labels in the test domains.

**Theory**:
1. Gave sufficient conditions under which we can train an adaptive classifier that could use unlabeled data in the test domain to use the spurious/unstable feature.
2. Show how to estimate the prediction probability using an individual classifier that only uses stable or unstable features.

**Experiment**:
Demonstrate that their method achieves near-optimal performance on synthetic datasets where the assumption of their method holds.




**Weaknesses:**

**Experiment**:
1. QRM baseline for the syn, Camelyon, and PACS datasets are not available.
2. Possible experiment on other real-world datasets where we expect the assumption made in the paper to hold: otherwise it is performing almost equivalently on Camelyon17 and PACS.






**Questions:**

**Minor Typos**:
1. Line 210: X_{S} \not\perp Y |E
2. Line 651: 4th line of the equation: Extra bracket after $\epsilon_{1}$ and $Pr[Y=1|X_2]$ instead of $Pr[Y=1|X_2]$.

---

> ### Author Rebuttal · Authors · 2023-08-09
>
> We sincerely thank the reviewer for their comments and suggestions.
>
> > “QRM baseline for the syn, Camelyon, and PACS datasets are not available.”
>
> We are happy to add the QRM baseline for these datasets in the next/final version of the paper.
>
> > “Possible experiment on other real-world datasets where we expect the assumption made in the paper to hold: otherwise it is performing almost equivalently on Camelyon17 and PACS.”
>
> We are also happy to consider other real-world datasets where our assumptions hold. If we find a suitable candidate, we will run experiments for the next/final version. However, **we note that SFB does not perform almost equivalently (to other methods) on PACS**. In particular, SFB performs best or joint-best on 3 of 4 domains, with a 6-percentage-point improvement on domain S and a 1-percentage-point improvement on domain A (representing a ~25% error reduction compared to the next best method).
>
>
> > “Minor Typos”
>
> Thank you for pointing these out – they have been corrected.

---

> > ### Comment · Reviewer_NDD6 · 2023-08-16
> > **Response to Authors**
> >
> > I thank the authors for their response. I stand with my current assessment. Please incorporate the abovementioned changes in the final version of the paper.

---

### Official Review · Reviewer_S7EC · 2023-07-09

**Soundness:** 3 good
**Presentation:** 3 good
**Contribution:** 3 good
**Rating:** 6
**Confidence:** 4

**Summary:**

This paper introduces a method to utilize spurious features when they are assumed to carry complementary information for predictions. A main contribution is to show that with unlabeled test samples, it is possible to learn how to use spurious features in the test domain. Based on the assumption that both stable feature $X_S$ and spurious feature $X_U$ are known, they derived a provably consistent algorithm for unsupervised test-domain adaptation. They further propose a new training objective function to learn $X_S$ and $X_U$ simultaneously when we have multiple training domains, based on the complementarity assumption ($X_S$ ⊥ $X_U$|$Y$). The full algorithm, including training and test domain adaptation, is named Stable Feature Boosting (SFB). The effectiveness of the test adaptation part of SFB is theoretically proved to be consistent. The whole SFB is empirically shown effective on both real and synthetic datasets.

**Strengths:**

- Originality: good. It is a new exploration on how to harness spurious features.
- This paper shows a comprehensive theoretical analysis of the adaptation problem.
- This paper is well written and easy to follow. Experiment settings are well introduced.

**Weaknesses:**

- Incomplete related works and baselines. As this work assumes the access to a set of unlabeled samples from the test domain, its setting is domain adaptation but not domain generalization. As a result, related works on domain adaption should be discussed and compared so that we can judge the contribution of this work. However, only domain generalization methods are compared, which could be unfair as different knowledge are assumed.
- IID accuracies are not shown. It is unclear how the SFB affect the IID performance.
- Missing ablation studies for the new training objective function of SFB.

**Questions:**

- How about the IID performance of SFB?
- What stability penalty is used to define $P_{Stability}$ in the experiments? How the quality of learning $X_S$ may effect the OOD performance of SFB w. adapt?
- How this work can be distinguished from existing domain adaptation methods?

**Limitations:**

The authors have underlined the complementarity assumption as a limitation. Besides, the estimation of $P_{X_S, X_U}$ is challenging in practice when the two variables are high dimensional. Furthermore, this adaptation method is theoretically useful only when $X_S$ is partially predictive.

---

> ### Author Rebuttal · Authors · 2023-08-09
>
> We sincerely thank the reviewer for their comments and suggestions. We hope that we have adequately addressed the concerns below. If the reviewer agrees, we hope that they consider raising their score.
>
> > “Incomplete related works and baselines. As this work assumes the access to a set of unlabeled samples from the test domain, its setting is domain adaptation but not domain generalization. As a result, related works on domain adaption should be discussed and compared so that we can judge the contribution of this work. However, only domain generalization methods are compared, which could be unfair as different knowledge are assumed.”
>
> We agree that, due to the use of unlabeled test-domain samples, our setting is perhaps most accurately described as multi-source, source-free domain adaptation [8-14]. However, we believe that it is more related to the domain generalization literature, for the reasons outlined below. Having said this, we do agree with the reviewer that related works in (unsupervised) domain adaptation should be discussed and compared in more detail, which we address below.
>
> **Domain generalization or domain adaptation?** Distribution shifts can often be categorised as covariate shift (P(x) changes), label shift (P(y) changes) or condition/correlation shift (P(y|x) changes) [19-21]. While a host of unsupervised domain adaptation (UDA) methods exist for addressing covariate and label shift, few exist for addressing correlation shift. One major reason for this is that, prior to our work, it was unclear how to adapt to such correlation shifts without test-domain labels (e.g., the most related DA method, ACTIR [15], to which we compared SFB in the paper, requires test-domain labels and primarily compares to DG methods themselves). As a result, datasets containing correlation shift have mostly been considered in the DG literature [1–6, 16–18], where the dominant approach has been to learn invariant models (without test-domain adaptation). For this reason, we chose to focus on the DG setting and methods, drawing connections to UDA in the related work.
>
> **Comparing to UDA methods: further experiments.** While UDA methods tend not to consider datasets with correlation shift, there are a few exceptions. One such exception is the test-time adaptation method of T3A [9]. This uses common DG datasets (containing correlation shift) yet adapts in the test domain using unlabelled examples. We have thus added a comparison to T3A on CMNIST (see PDF in general response), as well as a simple but effective pseudo-labeling baseline [7,8]. For the next/final version of the paper, we will add these comparisons for all other datasets (Synthetic, PACS, Camelyon17).
>
> **Comparing to UDA methods: further discussion.** To better highlight the above points, we have pulled up material from the appendix on UDA/pseudo-labelling methods and added further clarifications about the failures on existing UDA methods on severe correlation shifts, as well as their lack of theoretical guarantees on such shifts.
>
>
> > “IID accuracies are not shown. It is unclear how the SFB affect the IID performance.”
>
> For CMNIST, in-distribution/IID accuracies are depicted in Figure 2 of the submitted text: the 0.9 and 0.8 color-label correlatiions represent the training domains. The corresponding numerical values are now provided in Table 2 of the general-response PDF (newly added). In the next/final version of the paper, we are happy to provide the in-distribution/IID accuracies for SFB and baseline methods on all datasets (unfortunately, we did not save these values so will have to re-run the experiments to get them). This will include both the stable- and joint- predictor performances.
>
>
> > “Missing ablation studies for the new training objective function of SFB.”
>
> **In Appendix F.2**, we provided ablations on the following components of SFB: bias correction, post-hoc calibration and multiple pseudo-labeling rounds.
>
> **In the above PDF (general response)**, we have added ablations showing the performance of SFB when using different objectives for $P_{\text{Stability}}$ (IRM [1], VREx [2], CLOvE [3]).
>
> **In the next/final version of the paper**, we will also provide ablations for:
> - The $P_{\text{Stability}}$ and $P_{\text{Compl.}}$ components of SFB through sensitivity analyses on their hyperparamters ($\lambda_{\text{Stability}}, \lambda_{\text{Compl.}}$). In particular, we will use 2D contour/heat maps to show performance as a function of these hyperparameters, including the setting where one/both of them are 0.
> - The joint and stable-only risks of SFB in Eq. 5.1.
>
> > “What stability penalty is used to define $P_{\text{stability}}$ in the experiments? How the quality of learning $X_S$ may effect the OOD performance of SFB w. adapt?”
>
> We use IRM for $P_{\text{stability}}$ and have now clarified this in the main text. We have also added ablations showing the performance of SFB with other objectives for $P_{\text{Stability}}$, namely IRM [1], VREx [2], CLOvE [3] (see general response PDF). These additional ablations elucidate how the quality of learning $X_S$ affects the OOD performance of SFB w. adapt, with poorer stable predictors ultimately leading to poorer post-adaptation performance.
>
> > “the estimation of $P_{X_S,X_U}$  is challenging in practice when the two variables are high dimensional.“
>
> Although we described Theorem 4.4 in terms of the more general quantity $P_{X_S,X_U}$, **our pseudo-label approach in fact only requires $P_{\hat Y|X_U}$**, which is much easier to estimate; any method for calibrated classification can be used for this. We have clarified this in Section 4.

---

> > ### Comment · Reviewer_S7EC · 2023-08-15
> > **Post-rebuttal comment**
> >
> > I appreciate the authors' response. The discussions and new results address my concerns well, which move the scores upward. I trust the authors will incorporate these elements and the further ablation results in the final version, as indicated.

---

### Author Rebuttal · Authors · 2023-08-09

We sincerely thank all reviewers for their time and thoughtful feedback.

We were pleased to read that the paper is _“well-written and easy to follow”_ (S7EC, NDD6, 5BRF), has a _“comprehensive theoretical analysis”_ (S7EC), and makes _“an important step in the direction of adaptively making use of the spurious feature without any need for extra labels in the test domains”_ (NDD6).

The main concerns surrounded:
1. Extracting stable $X_S$ and unstable $X_U$ features in practice (f1Zc, 5BRF)
2. Comparing to unsupervised domain adaptation (UDA) baselines (S7EC, f1Zc)

In response, we have:
1. **Added text to our limitations section** on how extracting stable (and unstable) features is very much an active area of research in domain generalization [1–6], and in fact a topic in its own right, with future improvements in this area likely to benefit SFB.
2. **Added comparisons to two UDA baselines** on CMNIST (see attached PDF) and promised to add these baselines for the other datasets in the next version of the paper. In addition, we have **added further discussion of related UDA works**, pointing out that spurious-correlation shifts are typically studied in DG settings rather than UDA settings since, prior to our work, it was not clear how to deal with such shifts (in a principled/guaranteed way) without test-domain labels.

We believe that these clarifications and additional experiments have improved the paper and hope that they have addressed the main concerns of reviewers. We are happy to respond to any further questions/concerns during the discussion phase.

---

We now provide further details on the attached PDF.

### Algorithm 1
We have updated the notation and added comments to make this more readable (as suggested by Reviewer 5BRF).

### Table 1: SFB with different stability penalties
We have added experiments on CMNIST showing the performance of SFB with different stability penalties, namely IRM [1], VREx [2] and CLOvE [3].

### Table 2 & Figure 1: Comparison with UDA baselines
We have added UDA baselines on CMNIST:
- **Pseudo-labeling (PL, [7,8])**: This takes a trained model (e.g., ERM or IRM) and adapts it to unlabeled test-domain data by: (i) predicting; (ii) forming pseudo-labels by taking the maximum class-wise prediction; (iii) updating model weights using the pseudo-labels as if they were true labels; (iv) repeating steps i–iii for t iterations.
- **T3A [9]**: This takes a trained model (e.g., ERM or IRM) and adapts it to unlabeled test-domain data by: (i) predicting; (ii) forming pseudo-labels by taking the maximum class-wise prediction; (iii) forming pseudo-prototype representations for each class (i.e., per-class centroids or mean feature embeddings); (iv) forming new predictions based on the distance to these pseudo-prototypes or centroids.

From Table 2 and Figure 1, we note that:
- SFB performs best on almost all domains.
- When built on ERM, both T3A and PL fail on severe OOD domains (negative color-label correlation). This is no surprise since these methods rely on good initial performance in the test domain, which is not provided by ERM.
- When built on IRM, T3A gives a small but consistent boost on test-domains, particularly when the color-label correlation is strong. However, this boost is far below that of SFB.
- When built on IRM, PL works quite well on test-domains with a very strong color-label correlation (although always below SFB, and with much larger variance). However, on test-domains with a weak or medium color-label correlation, PL actually hurts performance. This is likely because PL does not combine stable and unstable predictions in a principled way (as SFB does; see Eq. (4.2) of the paper), causing issues on domains where a principled prediction-combination is required to achieve good performance.

Note that, to ensure the fairest comparison, we chose UDA methods that can operate in the source-free/test-time setting. Here, there are two distinct phases: (i) training, where methods have access to labeled training-domain data but no test-domain data; (ii) testing/adaptation, where methods have access to unlabeled test-domain data but no training-domain data.

[1] Arjovsky, M., Bottou, L., Gulrajani, I., & Lopez-Paz, D. (2019). Invariant risk minimization. _arXiv preprint arXiv:1907.02893_.

[2] Krueger, D., Caballero, E., Jacobsen, J. H., Zhang, A., Binas, J., Zhang, D., ... & Courville, A. (2021). Out-of-distribution generalization via risk extrapolation (rex). In _International Conference on Machine Learning_ (pp. 5815-5826). PMLR.

[3] Wald, Y., Feder, A., Greenfeld, D., & Shalit, U. (2021). On calibration and out-of-domain generalization. In _Advances in Neural Information Processing Systems_, 34, 2215-2227.

[4] Eastwood, C., Robey, A., Singh, S., Von Kügelgen, J., Hassani, H., Pappas, G. J., & Schölkopf, B. (2022). Probable domain generalization via quantile risk minimization. In _Advances in Neural Information Processing Systems_, 35, 17340-17358.

[5] Shi, Y., Seely, J., Torr, P. H., Siddharth, N., Hannun, A., Usunier, N., & Synnaeve, G. (2021). Gradient matching for domain generalization. _arXiv preprint arXiv:2104.09937_.

[6] Puli, A. M., Zhang, L. H., Oermann, E. K., & Ranganath, R. (2021). Out-of-distribution Generalization in the Presence of Nuisance-Induced Spurious Correlations. In _International Conference on Learning Representations_.

[7] Lee, D. H. (2013). Pseudo-label: The simple and efficient semi-supervised learning method for deep neural networks. In _Workshop on challenges in representation learning, ICML_ (Vol. 3, No. 2, p. 896).

[8] Rusak, E., Schneider, S., Pachitariu, G., Eck, L., Gehler, P. V., Bringmann, O., ... & Bethge, M. (2022). If your data distribution shifts, use self-learning. _Transactions on Machine Learning Research_.

[9] Iwasawa, Y., & Matsuo, Y. (2021). Test-time classifier adjustment module for model-agnostic domain generalization. In _Advances in Neural Information Processing Systems_, 34, 2427-2440.

---

> ### Author Response · Authors · 2023-08-10
> **References (continued)**
>
> [10] Dong, J., Fang, Z., Liu, A., Sun, G., & Liu, T. (2021). Confident anchor-induced multi-source free domain adaptation. In _Advances in Neural Information Processing Systems_, 34, 2848-2860.
>
> [11] Shen, M., Bu, Y., & Wornell, G. (2022). On the benefits of selectivity in pseudo-labeling for unsupervised multi-source-free domain adaptation. _arXiv preprint arXiv:2202.00796_.
>
> [12] Kundu, J. N., Venkat, N., & Babu, R. V. (2021). Universal source-free domain adaptation. In _Proceedings of the IEEE/CVF Conference on Computer Vision and Pattern Recognition_ (pp. 4544-4553).
>
> [13] Liang, J., Hu, D., & Feng, J. (2021). Do we really need to access the source data? source hypothesis transfer for unsupervised domain adaptation. In _International Conference on Machine Learning_ (pp. 6028-6039). PMLR.
>
> [14] Eastwood, C., Mason, I., Williams, C., & Schölkopf, B. (2022). Source-Free Adaptation to Measurement Shift via Bottom-Up Feature Restoration. In _International Conference on Learning Representations_.
>
> [15] Jiang, Y., & Veitch, V. (2022). Invariant and transportable representations for anti-causal domain shifts. In _Advances in Neural Information Processing Systems_, 35, 20782-20794.
>
> [16] Zhang, X., He, Y., Xu, R., Yu, H., Shen, Z., & Cui, P. (2023). Nico++: Towards better benchmarking for domain generalization. In _Proceedings of the IEEE/CVF Conference on Computer Vision and Pattern Recognition_ (pp. 16036-16047).
>
> [17] Wiles, O., Gowal, S., Stimberg, F., Rebuffi, S. A., Ktena, I., Dvijotham, K. D., & Cemgil, A. T. (2022). A Fine-Grained Analysis on Distribution Shift. In _International Conference on Learning Representations_.
>
> [18] Ye, N., Li, K., Bai, H., Yu, R., Hong, L., Zhou, F., ... & Zhu, J. (2022). OOD-Bench: Quantifying and understanding two dimensions of out-of-distribution generalization. In _Proceedings of the IEEE/CVF Conference on Computer Vision and Pattern Recognition_ (pp. 7947-7958).
>
> [19] Quinonero-Candela, J., Sugiyama, M., Schwaighofer, A., & Lawrence, N. D. (Eds.). (2008). Dataset shift in machine learning. _MIT Press_.
>
> [20] Storkey, A. (2009). When training and test sets are different: characterizing learning transfer. In _Dataset Shift in Machine Learning_, 30(3-28), 6.
>
> [21] Moreno-Torres, J. G., Raeder, T., Alaiz-Rodríguez, R., Chawla, N. V., & Herrera, F. (2012). A unifying view on dataset shift in classification. _Pattern Recognition_, 45(1), 521-530.
>
> [22] Guo, C., Pleiss, G., Sun, Y., & Weinberger, K. Q. (2017, July). On the calibration of modern neural networks. In _International Conference on Machine Learning_ (pp. 1321-1330). PMLR.
>
> [23] Lakshminarayanan, B., Pritzel, A., & Blundell, C. (2017). Simple and scalable predictive uncertainty estimation using deep ensembles. In _Advances in Neural Information Processing Systems_, 30.
>
> [24] Kumar, A., Liang, P. S., & Ma, T. (2019). Verified uncertainty calibration. In _Advances in Neural Information Processing Systems_, 32.
>
> [25] Gupta, C., Podkopaev, A., & Ramdas, A. (2020). Distribution-free binary classification: prediction sets, confidence intervals and calibration. In _Advances in Neural Information Processing Systems_, 33, 3711-3723.

---

### Decision · Program_Chairs · 2023-09-21

**Decision:**

Accept (poster)

**Comment:**

The paper shows that spurious features can be harnessed in the test domain without labels, using only invariant-feature pseudo-labels, and propose an algorithm for doing so. Initially reviewers had some concerns about the limitation of the proposed work and comparison with proper baselines. Authors did a good job in addressing these issues during the rebuttal phase. Thus, all reviewers agree the paper is above the bar for being accepted to neurips.